# Dissecting human population variation in single-cell responses to SARS-CoV-2

Yann Aquino[1,2,27], Aurélie Bisiaux[1,27], Zhi Li[1,27], Mary O'Neill[1,27], Javier Mendoza-Revilla[1], Sarah Hélène Merkling[3], Gaspard Kerner[1], Milena Hasan[4], Valentina Libri[4], Vincent Bondet[5], Nikaïa Smith[5], Camille de Cevins[6], Mickaël Ménager[6,7], Francesca Luca[8,9,10], Roger Pique-Regi[8,9], Giovanna Barba-Spaeth[11], Stefano Pietropaoli[11], Olivier Schwartz[12], Geert Leroux-Roels[13], Cheuk-Kwong Lee[14], Kathy Leung[15,16], Joseph T. Wu[15,16], Malik Peiris[17,18,19], Roberto Bruzzone[18,19], Laurent Abel[20,21,22], Jean-Laurent Casanova[20,21,22,23,24], Sophie A. Valkenburg[18,25], Darragh Duffy[5,19], Etienne Patin[1], Maxime Rotival[1,28 ✉] & Lluis Quintana-Murci[1,26,28 ✉]

Humans display substantial interindividual clinical variability after SARS-CoV-2 infection[1–3], the genetic and immunological basis of which has begun to be deciphered[4]. However, the extent and drivers of population differences in immune responses to SARS-CoV-2 remain unclear. Here we report single-cell RNA-sequencing data for peripheral blood mononuclear cells—from 222 healthy donors of diverse ancestries—that were stimulated with SARS-CoV-2 or influenza A virus. We show that SARS-CoV-2 induces weaker, but more heterogeneous, interferon-stimulated gene activity compared with influenza A virus, and a unique pro-inflammatory signature in myeloid cells. Transcriptional responses to viruses display marked population differences, primarily driven by changes in cell abundance including increased lymphoid differentiation associated with latent cytomegalovirus infection. Expression quantitative trait loci and mediation analyses reveal a broad effect of cell composition on population disparities in immune responses, with genetic variants exerting a strong effect on specific loci. Furthermore, we show that natural selection has increased population differences in immune responses, particularly for variants associated with SARS-CoV-2 response in East Asians, and document the cellular and molecular mechanisms by which Neanderthal introgression has altered immune functions, such as the response of myeloid cells to viruses. Finally, colocalization and transcriptome-wide association analyses reveal an overlap between the genetic basis of immune responses to SARS-CoV-2 and COVID-19 severity, providing insights into the factors contributing to current disparities in COVID-19 risk.

A notable feature of the COVID-19 pandemic is the substantial clinical variation among individuals infected with SARS-CoV-2, ranging from asymptomatic infection to fatal disease[1–3]. Risk factors include advanced age[1] as well as male sex[5], comorbidities[6] and host genetics[4,7,8]. Furthermore, variation in innate immunity[9–11]—including inborn errors or neutralizing auto-antibodies against type I interferons[12–14]—contribute to variation in clinical outcome, and epidemiological and genetic data suggest differences between populations[6,7,15,16]. This, together with reports of ancestry-related differences in transcriptional responses to immune challenges[17–19], calls for investigations of the magnitude and drivers of variation in immune responses to SARS-CoV-2 across populations worldwide.

Pathogen-imposed selection pressures have been paramount during human evolution[20]. Human adaptation to RNA viruses, through selective sweeps or archaic admixture, has been identified as a source of population genetic differentiation[18,21,22] and adaptation signals have been reported at coronavirus-interacting proteins in East Asians[23,24].

There is also evidence for links between archaic introgression and immunity[25], with Neanderthal haplotypes associated with COVID-19 severity[26,27]. However, the effects of natural selection and archaic admixture on immune responses to SARS-CoV-2 remain to be investigated.

We addressed these questions by exposing peripheral blood mononuclear cells (PBMCs) from individuals of Central African, West European and East Asian descent to SARS-CoV-2 and, for comparison, to influenza A virus (IAV). By combining single-cell RNA-sequencing (scRNA-seq) with quantitative and population genetics approaches, we delineate environmental and genetic drivers of population differences in immune responses to SARS-CoV-2.

## Single-cell responses to RNA viruses

We characterized transcriptional responses to SARS-CoV-2 and IAV by performing scRNA-seq analysis of PBMCs from 222 SARS-CoV-2-naive donors originating from three geographical locations (Central Africa,

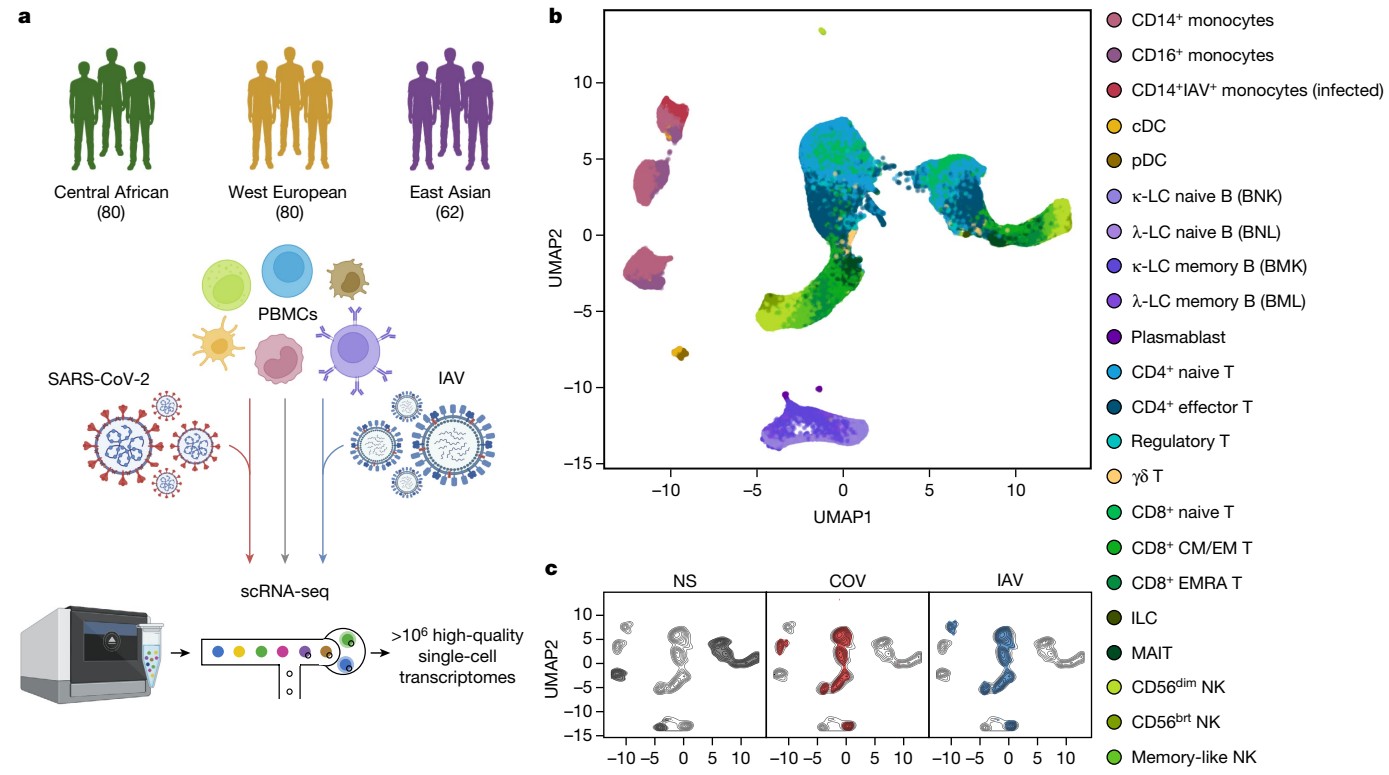

**Fig. 1 | Population single-cell responses to SARS-CoV-2 and IAV. a**, The study design. The diagram was created using BioRender. **b,c**, Uniform manifold approximation and projection (UMAP) embedding of 1,047,824 PBMCs: resting (non-stimulated; NS) or stimulated with SARS-CoV-2 (COV) or IAV for 6 h.

**b**, The colours indicate the 22 cell types inferred. **c**, The distribution of cells in the NS, COV and IAV conditions on UMAP coordinates. The contour plot indicates the overall density of cells, and the coloured areas delineate regions of high cell density in each condition (NS (grey), COV (red) and IAV (blue)).

$n$ = 80 male; West Europe, $n$ = 80 male; East Asia, $n$ = 36 female and 26 male) and with different genetic ancestries (Supplementary Fig. 1 and Supplementary Table 1). PBMCs were treated for 6 h (Supplementary Note 1, Supplementary Fig. 2 and Supplementary Table 2) with a mock-control (non-stimulated), SARS-CoV-2 (ancestral strain, BetaCoV/France/GE1973/2020) or IAV (H1N1/PR/8/1934). We captured over 1 million high-quality single-cell transcriptomes (Fig. 1a, Supplementary Fig. 3 and Supplementary Table 3a). By combining transcriptome-based clusters with cellular indexing of transcriptomes and epitopes by sequencing (CITE-seq; Methods), we defined 22 cell types across myeloid, B, CD4$^+$ T, CD8$^+$ T and natural killer (NK) immune lineages (Fig. 1b, Supplementary Fig. 4 and Supplementary Table 3b–d). After virus exposure, most cell types showed moderate changes in abundance, with the strongest changes observed in the myeloid lineage after IAV treatment (Supplementary Note 2 and Supplementary Table 3e).

After adjusting for technical factors (Methods and Supplementary Fig. 5), we found that lineage identity was the main driver of gene expression variation (around 32%), followed by virus exposure (around 27%) (Fig. 1b,c). Both viruses induced a strong transcriptional response, with 2,914 genes upregulated (false-discovery rate (FDR) < 0.01, log$_2$[FC] > 0.5; out of 12,655 with detectable expression; Supplementary Table 3f). These responses were highly correlated across lineages and featured a strong induction of interferon-stimulated genes (ISGs) (Extended Data Fig. 1a). However, myeloid responses were markedly heterogeneous, with SARS-CoV-2 inducing a transcriptional network enriched in inflammatory-response genes (Gene Ontology (GO): 0006954; fold-enrichment (FE) = 3.4, FDR < 4.9 × 10$^{-8}$; Supplementary Table 3g). For example, *IL1A*, *IL1B* and *CXCL8* were highly and specifically upregulated in response to SARS-CoV-2 (log$_2$[FC] > 2.8, FDR < 2.3 × 10$^{-36}$), consistent with in vitro and in vivo studies[28,29].

To assess interindividual variability in the response to viruses, we summarized each individual's response as a function of their mean ISG expression (Supplementary Table 3h). SARS-CoV-2 induced more variable ISG activity than IAV across lineages[30], with myeloid cells displaying the strongest differences (Levene test, $P$ < 6.2 × 10$^{-6}$; Extended Data Fig. 1b). We determined the contributions of the various interferons (IFNs) to variation of ISG activity using single-molecule arrays (SIMOA) to quantify the levels of secreted IFNα, IFNβ and IFNγ. In the SARS-CoV-2 condition, IFNα accounted for up to 57% of ISG variability (Extended Data Fig. 2a,b), consistent with its determinant role in COVID-19 pathogenesis[13]. *IFNA1-21* transcripts were mostly produced by infected CD14$^+$ monocytes and plasmacytoid dendritic cells (pDCs) after IAV stimulation, whereas pDCs were the only important source of *IFNA1-21* after SARS-CoV-2 stimulation (that is, producing 88% of transcripts; Extended Data Fig. 2c). *IFNA1-21* expression by pDCs was weaker after stimulation with SARS-CoV-2 (log$_2$[FC] = 6.4 versus 12.5 for IAV, Wilcoxon's rank-sum test, $P$ = 1.2 × 10$^{-16}$). Nevertheless, patterns of interindividual variability for ISG activity were notably similar after virus treatment ($r$ = 0.60, Pearson's $P$ < 1.2 × 10$^{-22}$; Extended Data Fig. 2d), indicating that the IFN-driven response is largely shared between SARS-CoV-2 and IAV.

## Cellular heterogeneity across populations

We assessed how immune responses differ across populations by comparing male individuals of African and European ancestry, who were sampled in a single recruitment effort thereby mitigating potential batch effects (Methods). As East Asian donors were recruited independently and present distinct demographic characteristics (Supplementary Table 1), they were excluded from cross-population comparisons. Focusing on cellular proportions, we detected marked population

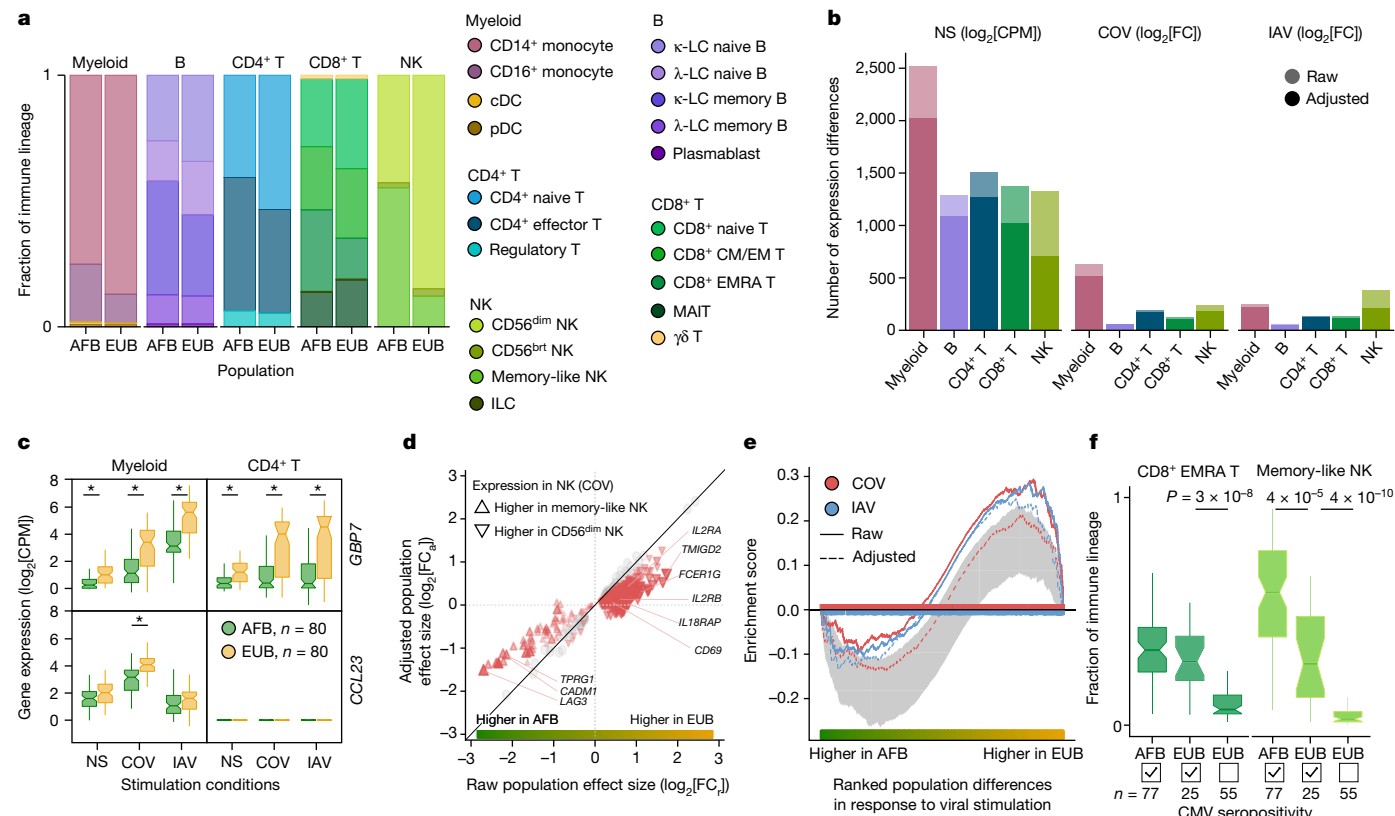

**Fig. 2 | Cellular composition affects the transcriptional responses to viral stimuli. a**, Cell type proportions within each immune lineage in Africans (AFB) and Europeans (EUB). brt, bright. **b**, The number of genes differentially expressed between the African and European groups in the basal state (NS) or in response to SARS-CoV-2 (COV) or IAV, in each immune lineage. Numbers are provided before and after adjustment for cellular composition. **c**, Examples of popDRGs, either shared across cell types and viruses (*GBP7*) or specific to SARS-CoV-2-stimulated myeloid cells (*CCL23*). Statistical analysis was performed using two-sided Student's t-tests with adjustment using the Benjamini–Hochberg method; *$P < 0.001$. Exact $P$ values are provided in Supplementary Table 4b. **d**, The effect of adjusting for cellular composition on genes differentially expressed between populations after exposure to SARS-CoV-2. Adjustment reduces raw population fold-changes ($FC_a$ versus $FC_r$) in the expression of genes that are differentially expressed between memory-like NK cells and

CD56$^{dim}$ NK cells (red triangles; genes with similar expression are shown in grey). **e**, The effect of adjusting for cellular composition on population differences in the response to viral stimulation for genes involved in the positive regulation of cell migration (GO:0030335) in the NK lineage. For each stimulus, gene set enrichment analysis enrichment curves are shown before and after adjusting on the basis of cellular composition. Grey shades indicate the 95% expected range for the enrichment curve when gene labels are permuted at random. **f**, The distribution of CD8$^+$ EMRA T and memory-like NK cell frequencies in Africans and Europeans according to CMV$^{+/-}$ serostatus. $P$ values ($P < 0.01$) calculated using Wilcoxon's two-sided rank-sum tests are shown. For **c** and **f**, the centre line shows the median; the notches show the 95% confidence intervals (CIs) of the median; the box limits show the upper and lower quartiles; and the whiskers show 1.5× interquartile range. The number (*n*) of independent biological samples is indicated where relevant.

differences in lineage composition, particularly for NK cells (Fig. 2a and Supplementary Table 4a). A subset identified as memory-like NK cells[31] constituted 55.2% of the NK compartment in African-descent individuals, but only 12.2% in Europeans (Wilcoxon's rank-sum test, $P < 1.3 × 10^{-18}$; Extended Data Fig. 3a,b and Supplementary Fig. 6). African donors also presented higher proportions of CD16$^+$ monocytes[32] and memory lymphocyte subsets, such as memory B cells, effector CD4$^+$ T cells and effector memory re-expressing CD45RA (EMRA) CD8$^+$ T cells (Wilcoxon's rank-sum test, $P < 4.7 × 10^{-3}$).

Across lineages, we found 3,389 genes displaying population differences in expression in the basal state (popDEGs; FDR < 0.01, |log$_2$[fold change (FC)]| > 0.2) and 898 and 652 displaying differential responses between populations (popDRGs; FDR < 0.01, |log$_2$[FC]| > 0.2) after stimulation with SARS-CoV-2 and IAV, respectively (Fig. 2b and Supplementary Table 4b,c). popDRGs included key immunity regulators, such as the IFN-responsive *GBP7* and the gene coding for the macrophage inflammatory protein MIP-3, *CCL23*, both of which were more strongly upregulated in Europeans (Fig. 2c). The *GBP7* response was common to both viruses and all lineages (log$_2$[FC] > 0.88, Student's t-test, adjusted $P$ ($P_{adj}$) < $1.4 × 10^{-3}$), but that of *CCL23* was specific to SARS-CoV-2-stimulated myeloid cells (log$_2$[FC] = 0.72, Student's t-test,

$P_{adj} = 5.3 × 10^{-4}$). We estimated that population differences in cellular composition accounted for 15–47% of popDEGs and for 7–46% of popDRGs, with the strongest impact on NK cells (Fig. 2b,d and Extended Data Fig. 3c). Variation in cellular composition mediated pathway-level differences in response to viral stimulation between populations (Supplementary Table 4d). For example, in virus-stimulated NK cells, genes involved in the promotion of cell migration, such as *CSF1* or *CXCL10*, were more strongly induced in Europeans (normalized enrichment score > 1.5, gene set enrichment analysis, $P_{adj} < 0.009$). However, the loss of this signal after adjustment for cellular composition (Fig. 2e) indicates that fine-scale cellular heterogeneity drives population differences in immune responses to SARS-CoV-2.

## Repercussions of CMV infection

We next investigated the sources of population differences in cellular composition. We found no strong genetic effects on cellular proportions (Supplementary Note 3 and Supplementary Table 4e), suggesting a predominantly environmental origin to such population differences. As latent cytomegalovirus (CMV) infection alters cellular proportions[33–35] and its prevalence varies across populations[36], we determined

the CMV$^{+/-}$ serostatus of the samples. All but one of the African-descent individuals were CMV$^+$ (99%), versus 31% of Europeans, and CMV$^+$ was associated with higher proportions of memory-like NK and CD8$^+$ EMRA T cells in Europeans (Fig. 2f and Extended Data Fig. 3d). Using mediation analysis, we estimated that CMV serostatus accounts for up to 73% of the differences in the proportion of these cell types between Africans and Europeans; these differences substantially impact the transcriptional response to SARS-CoV-2 (Supplementary Table 4f,g, Supplementary Notes 4 and 5 and Supplementary Fig. 7). However, other than its effects on cellular composition, CMV$^+$ had a limited direct effect on SARS-CoV-2 responses, with only one gene presenting significant expression differences in response to this virus (*ERICH3* in CD8$^+$ T cells, log$_2$FC = 1.7, FDR = 0.007; Supplementary Table 4h). These findings highlight how differing environmental exposures, such as CMV infection, may lead to population differences in the responses to SARS-CoV-2 through changes in the lymphoid composition.

## Genetic basis of the leukocyte response

To assess the effects of human genetic variants on transcriptional variation, we mapped expression quantitative trait loci (eQTLs) jointly in all three populations, focusing on *cis*-regulatory variants. At an FDR of 1%, we identified 1,866–4,323 independent eQTLs per lineage, affecting 5,198 genes (Fig. 3a and Supplementary Table 5a). Among the 9,150 eQTLs detected, 11% were ancestry specific (*n* = 973; Supplementary Note 6), underscoring the importance of including diverse ancestries in genomics research. Increasing the resolution to 22 cell types revealed an additional 3,603 eQTLs (Extended Data Fig. 4a,b and Supplementary Table 5b). We found that 79% of eQTLs were replicated (*P* < 0.01) in at least three cell types, but only 22% were common to all lineages. In total, 812 eQTLs were cell-type-specific, around 45% of which were detected in myeloid cells (Extended Data Fig. 4b), including a pDC-specific eQTL (rs114273142) at *MIR155HG*—hosting a microRNA that promotes sensitivity to type I IFNs[37] (Extended Data Fig. 4c and Supplementary Note 7). Broadly, eQTL effect sizes were more correlated across ontogenetically related cell types (mean correlation within and between lineages of *r* = 0.60 and 0.47, Wilcoxon's rank-sum test, *P* = 6.2 × 10$^{-6}$; Extended Data Fig. 4d).

Focusing on variants that altered responses to viral stimuli (reQTLs), we identified 1,505 reQTLs affecting 1,213 genes (Supplementary Table 5c,d). Supporting the replicability of the results, our IAV reQTLs are enriched in genes that are reported to contain IAV-specific eQTLs[19] (OR > 3.2, Fisher's exact test, *P* < 9.4 × 10$^{-4}$), with more than 98% of replicated eQTLs affecting expression in the same direction (Supplementary Note 8, Supplementary Fig. 8 and Supplementary Table 5e). The correlation of reQTL effect sizes across ontogenetically related cell types was weaker than for eQTLs (*r* = 0.36 and 0.50, respectively, Wilcoxon's rank-sum test, *P* < 5.6 × 10$^{-13}$; Extended Data Fig. 4d). Furthermore, the proportion of virus-dependent reQTLs differed across cell types. In lymphoid cells, only 7.7% of reQTLs differed in effect size between viruses (interaction *P* < 0.01; Fig. 3b,c), whereas 49% of myeloid reQTLs were virus dependent (interaction *P* < 0.01), with 46 and 185 reQTLs displaying specific, stronger effects after SARS-CoV-2 and IAV stimulation, respectively. The strongest SARS-CoV-2 reQTL (rs534191, Student's *t*-test, *P* = 1.96 × 10$^{-16}$ (SARS-CoV-2) and *P* = 0.05 (IAV); Fig. 3d) was identified in myeloid cells at *MMP1*, encoding a biomarker of COVID-19 severity[38]. These analyses reveal that the effects of virus-induced reQTLs are cell-type dependent and highlight the virus specificity of the genetic basis of the myeloid response.

## Ancestry effects on immune responses

To evaluate the contribution of genetic variation to population differences in immune responses, we focused on popDEGs and popDRGs. We found that 11–24% of the genes expressed in each lineage had at least one

eQTL, but this proportion increased up to 56% and 60% for popDEGs and popDRGs that were not explained by cellular heterogeneity, respectively (Fisher's exact test, *P* < 1.4 × 10$^{-6}$; Fig. 3e and Extended Data Fig. 5a). The popDEGs and popDRGs displaying the largest population differences were more likely to be under genetic control and associated with large-effect (r)eQTLs (Extended Data Fig. 5b–d). We used mediation analysis to assess, for each gene, immune lineage and virus, the fraction of population differences explained by genetics (that is, the most significant eQTL) or cellular heterogeneity (Supplementary Table 6 and Supplementary Note 9). Cellular composition had a broad effect on population differences in gene expression and viral responses (explaining 16–62% of differences per lineage and virus, with the strongest effect in NK cells), whereas genetics had a weaker effect (explaining 13–35% of population differences; Fig. 3f and Extended Data Fig. 5e). However, genetics had strong effects on a gene subset (141–433 genes per lineage) for which they accounted for 32–58% of population differences. For example, 81–100% of the differences in *GBP7* expression between Africans and Europeans were explained by a single variant displaying strong population differentiation (rs1142888, derived allele frequency (DAF) = 0.13 and 0.53 in Africans and Europeans, respectively, fixation index ($F_{ST}$) = 0.26, $|\beta_{eQTL}|$ > 1.7 across lineages after stimulation). Thus, population variation in immune responses is driven largely by cellular heterogeneity, but genetic variants with marked allele frequency variation contribute to population differences at specific loci.

## Natural selection and SARS-CoV-2 responses

To investigate the contribution of natural selection to population differences in immune responses, we first searched for overlaps between (r)eQTLs and genome-wide signals of local adaptation, measured by the population branch statistic (PBS)[39]. We identified 1,616 eQTLs (1,215 genes) and 180 reQTLs (166 genes) displaying strong population differentiation (empirical *P* < 0.01), 90 of which were ancestry specific (Supplementary Table 7a and Supplementary Note 6). Among genes harbouring putatively adaptive (r)eQTLs, we found key players in IFN-mediated antiviral immunity, such as *DHX58* and *TRIM14* in Africans, *ISG20*, *IFIT5*, *BST2* and *IFITM2-3* in Europeans, and *IFI44L* and *IFITM2* in East Asians.

We then used CLUES[40] to identify rapid changes in (r)eQTL frequency over the last 2,000 generations (that is, 56,000 years) in each population (Supplementary Fig. 9 and Supplementary Table 7b). We found signals of rapid adaptation (maximum $|Z|$ > 3) targeting the same (*IFITM2*, *IFIT5*) or different (*ISG20*, *IFITM3*, *TRIM14*) eQTLs at highly differentiated genes, suggesting repeated adaptations targeting IFN-mediated antiviral immunity (Supplementary Note 10, Supplementary Table 7c and Supplementary Fig. 10). We determined whether selection had altered gene expression in specific cell types or in response to SARS-CoV-2 or IAV by testing for increased population differentiation (PBS) at (r)eQTLs within each cell type, relative to random single-nucleotide polymorphisms (SNPs) matched for allele frequency, linkage disequilibrium (LD) and distance to the nearest gene. In the basal state, eQTLs were more strongly differentiated in Europeans, the strongest signal observed for γδ T cells (Extended Data Fig. 6a). Among popDEGs for which genetics mediates more than 50% of the differences between Africans and Europeans, 34% presented signals of rapid adaptation in Europeans (versus 21% in Africans, Fisher's exact test, *P* = 7.7 × 10$^{-6}$). For example, population differences at *GBP7* have been driven by a frequency increase, over the last 782–1,272 generations, of the rs1142888-G allele in Europeans (maximum $|Z|$ > 4.3, Extended Data Fig. 6b).

Focusing on responses to viruses, SARS-CoV-2 reQTLs displayed increased population differentiation in East Asians (FE = 1.24, one-sided resampling, *P* < 2 × 10$^{-4}$; Extended Data Fig. 6c) and were enriched in East-Asian-specific variants (OR > 4.2, Fisher's exact test, *P* < 2.3 × 10$^{-6}$; Supplementary Note 6 and Supplementary Table 7d). Furthermore, among SARS-CoV-2-specific reQTLs, 28 reQTLs (5.3%) displayed

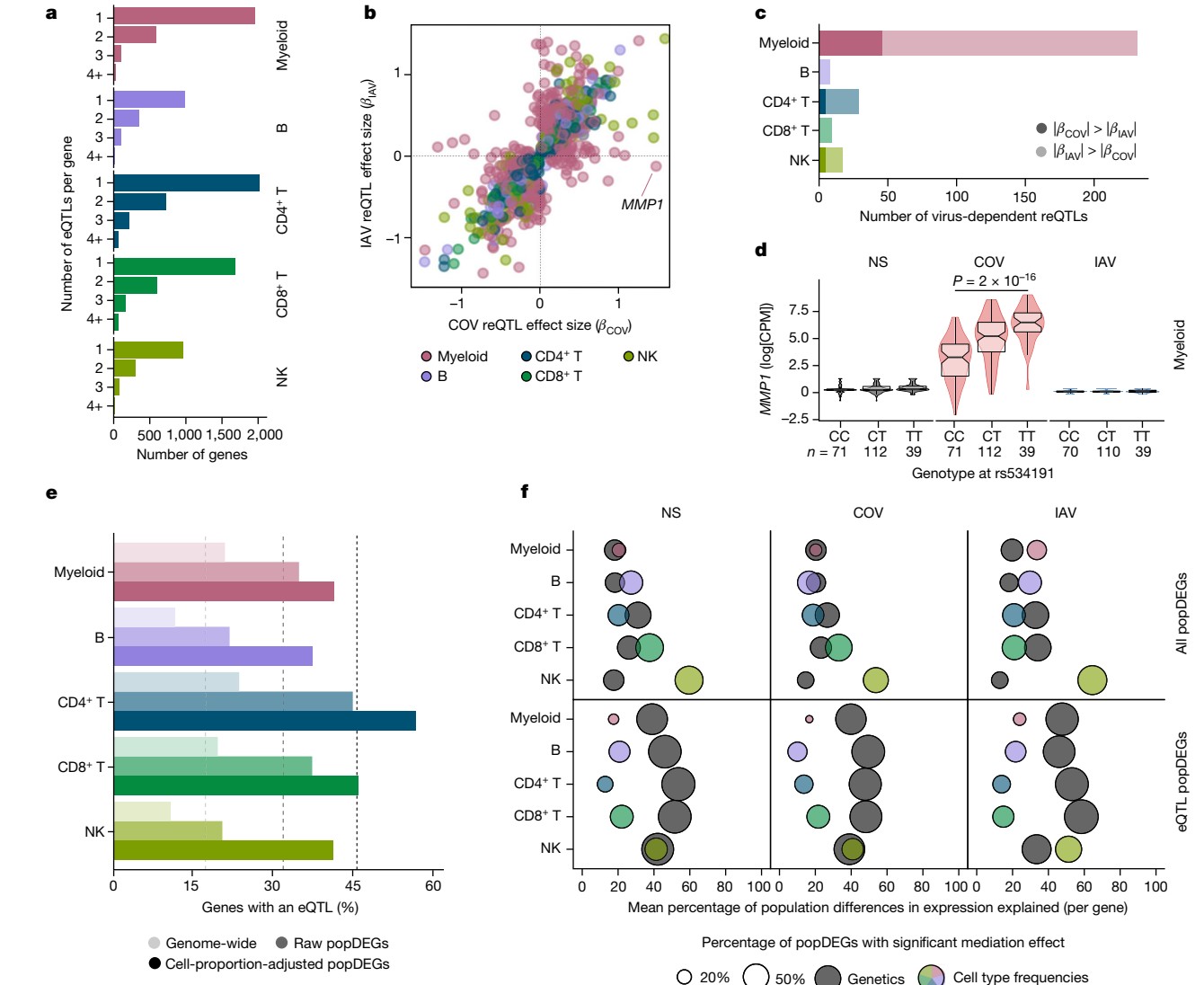

**Fig. 3 | Genetic basis of immune responses to RNA viruses. a**, The number of eQTLs detected per gene within each immune lineage. **b**, Comparison of reQTL effect sizes ($\beta$) between SARS-CoV-2- and IAV-stimulated cells. Each dot represents a specific reQTL (that is, SNP, gene and lineage) and its colour indicates the lineage in which it was detected. **c**, The number of virus-dependent reQTLs (two-sided Student's *t*-test nominal interaction, *P* < 0.01) in each immune lineage, coloured according to the lineage and the stimulus for which the reQTL has the largest effect size. **d**, Example of a SARS-CoV-2-specific reQTL at *MMP1*. *P* values (*P* < 0.01) calculated using Student's two-sided *t*-tests are shown. The centre line shows the median; the notches show the 95% CIs of the median; the box limits show the upper and lower quartiles; the whiskers show 1.5× interquartile range;

and the points show outliers. **e**, Enrichment in eQTLs among genes that are differentially expressed between populations (popDEGs). For each immune lineage, the bars indicate the percentage of genes with a significant eQTL, at the genome-wide scale and among popDEGs, before or after adjustment for cellular composition. **f**, For each lineage and stimulus, the *x* axis indicates the mean contribution of either genetics (that is, the most significant eQTL per gene in each lineage and stimulus) or cellular composition to population differences in expression, across all popDEGs (top) or popDEGs associated with an eQTL (bottom). The size of the dots reflects the percentage of genes with a significant mediated effect at an FDR of 1% (Supplementary Table 6). The number (*n*) of independent biological samples is indicated where relevant.

signals of adaptation in East Asians starting 770–970 generations ago (around 25,000 years)—a timeframe associated with genetic adaptation at SARS-CoV-2-interacting proteins[23] (OR relative to other populations = 2.6, Fisher's exact test, *P* = 7.3 × 10⁻⁴; Fig. 4a and Extended Data Fig. 7a–c). An example is the immune mediator *LILRB1*, which has a SARS-CoV-2-specific reQTL (rs4806787) in pDCs (Extended Data Fig. 7d). However, the selection events making the largest contribution to the differentiation of SARS-CoV-2 responses in East Asia (top 5% PBS) began before this period (more than 970 generations ago, OR = 1.94, Fisher's exact test, *P* = 0.019; Fig. 4b). For example, the rs1028396-T allele (80% frequency in East Asia versus 16–25% elsewhere), associated with a weaker response of *SIRPA* to SARS-CoV-2 in CD14⁺ monocytes,

presents a selection signal beginning more than 45,000 years ago (Fig. 4b and Extended Data Fig. 7e). SIRPα inhibits infection by endocytic viruses, including SARS-CoV-2[41]. These results suggest recurrent genetic adaptation targeting antiviral immunity over the last 50,000 years, contributing to present-day population differences in immune responses to SARS-CoV-2.

## Neanderthal heritage on immune functions

We investigated the effects of Neanderthal introgression on immune responses to viruses by defining 100,345 'archaic' variants (aSNPs) and testing for biased eQTL representation among aSNPs relative to

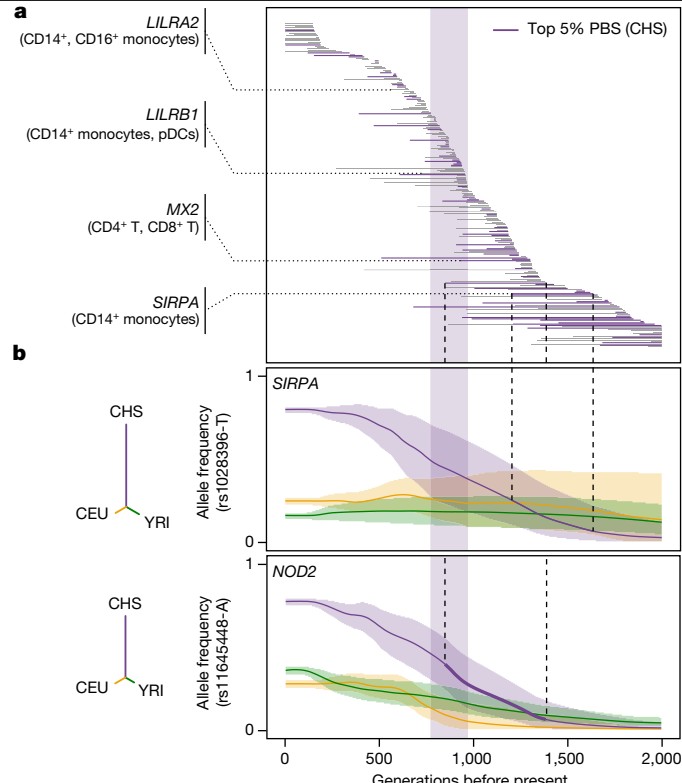

**Fig. 4 | Natural selection effects on population differentiation of immune responses. a**, Estimated periods of selection, over the past 2,000 generations, for 245 SARS-CoV-2 reQTLs with significant signals of rapid adaptation in East Asians (CHS) (maximum $|Z| > 3$). Each horizontal line represents a variant, sorted in descending order of time to onset of selection. The area shaded in purple highlights the period (770–970 generations ago) associated with genetic adaptation at host coronavirus-interacting proteins in East Asians[23]. Several immunity-related genes are highlighted. **b**, Allele frequency trajectories of two SARS-CoV-2 reQTLs (rs1028396 at *SIRPA* and rs11645448 at *NOD2*) in Africans (YRI, green), Europeans (CEU, yellow) and East Asians (CHS, purple). The full lines indicate the maximum a posteriori estimate of allele frequency at each epoch and shaded areas indicate the 95% CIs. The dendrograms show the estimated unrooted population phylogeny for each eQTL based on PBS (that is, the branch length between each pair of populations is proportional to $-\log_{10}[1 - F_{ST}]$).

random, matched SNPs (Methods). We found that archaic haplotypes were 1.4–1.5 times more likely to alter gene expression in the basal state (one-sided permutation test, $P = 3 \times 10^{-4}$) and after stimulation with SARS-CoV-2 or IAV (one-sided permutation test, $P = 9 \times 10^{-4}$ and $3 \times 10^{-3}$, respectively) in Europeans, and this trend was only marginally significant in East Asians after viral stimulation (FE > 1.2, one-sided permutation test, $P < 2 \times 10^{-2}$; Extended Data Fig. 8a and Supplementary Table 8a–c). Enrichment was strongest in SARS-CoV-2-stimulated CD16+ monocytes from Europeans, suggesting that archaic haplotypes altering myeloid responses have been preferentially retained in their genomes. Archaic haplotypes with eQTLs are generally present at higher frequencies compared with archaic haplotypes without eQTLs ($\Delta f$(introgressed allele) >3.2%, Student's *t*-test, $P_{adj} < 8 \times 10^{-3}$; Extended Data Fig. 8b and Supplementary Table 8d,e), even after adjustment for minor allele frequency (MAF) to ensure similar power for eQTL detection, supporting the adaptive nature of Neanderthal regulatory alleles.

To characterize the functional consequences of archaic introgression at the cell-type level, we focused on introgressed eQTLs for which the archaic allele was found at its highest frequency in Eurasians (that is, 5% most frequent). These included known adaptively introgressed variants at *OAS1-3* or *PNMA1* in Europeans and *TLR1*, *FANCA* or *IL10RA* in East Asians[18,42–46], for which we delineated the cellular and molecular effects

(Extended Data Figs. 8c and 9a and Supplementary Table 8f). Yet, we identified previously unreported signals of Neanderthal introgression affecting immunity phenotypes. For example, an introgressed reQTL (rs58964929-A, 38% of Europeans versus 22% of East Asians) decreases *UBE2F* responses to SARS-CoV-2 and IAV in monocytes (Extended Data Fig. 9b). UBE2F is involved in neddylation, a post-translational modification that is required for the nuclear translocation of IRF7 by myeloid cells after RNA virus infection and, therefore, for the induction of type I IFN responses[47]. Likewise, an introgressed eQTL (rs11119346-T, 43% in East Asians versus less than 3% in Europeans) downregulates *TRAF3IP3*— a negative regulator of the cytosolic RNA-induced IFN response[48]— in IAV-infected monocytes, thereby favouring IFN release after viral infection (Extended Data Fig. 9c,d). We also identified a 35.5 kb Neanderthal haplotype reaching 61% frequency in East Asians (versus 24% in Europeans, tagged by rs9520848-C allele) that is associated with higher basal expression of the cytokine gene *TNFSF13B* by MAIT cells (Extended Data Fig. 9e,f). Collectively, these results reveal how archaic introgression has altered immune functions in present-day Eurasians at the molecular and cellular level.

## Contribution of eQTLs to COVID-19 risk

We investigated the contributions of genetic variants altering responses to SARS-CoV-2 ex vivo to COVID-19 risk in vivo by determining whether (r)eQTLs were more strongly associated with COVID-19 GWAS hits[8] than random, matched SNPs (Methods). We observed an enrichment in eQTLs at loci associated with susceptibility (reported cases) and severity (hospitalized or critical cases) (FE = 4.1 and FE > 3.8, respectively, one-sided resampling, $P < 10^{-4}$), and a specific enrichment in reQTLs at severity loci (FE > 3.7, one-sided resampling, $P < 3 \times 10^{-3}$; Fig. 5a). This trend was observed across most cell lineages (Extended Data Fig. 10a). Colocalization analyses identified 40 genes at which there was a high probability of (r)eQTL colocalization with COVID-19 hits (posterior probability that both traits are linked to the same SNP ($PP_{H4}$) > 0.8) and transcriptome-wide association studies (TWASs) linked predicted gene expression with COVID-19 risk for 30 of these genes (FDR$_{TWAS}$ < 0.01; Supplementary Table 9a). These included direct regulators of innate immunity, such as *IFNAR2* in non-stimulated CD4+ T cells, *IRF1* in non-stimulated NK and CD8+ T cells, *OAS1* in lymphoid cells stimulated with SARS-CoV-2 and IAV, and *OAS3* in SARS-CoV-2-exposed CD16+ monocytes (Fig. 5b and Extended Data Fig. 10b,c). These results support a contribution of immunity-related (r)eQTLs to COVID-19 risk.

Focusing on the evolutionary factors affecting COVID-19 risk, we identified 20 eQTLs that (1) colocalized with COVID-19 hits ($PP_{H4}$ > 0.8) and (2) presented positive selection signals (top 1% PBS, $n = 13$ eQTLs) or evidence of archaic introgression ($n = 7$ eQTLs), 14 of which regulate genes of which the expression is correlated with COVID-19 susceptibility and/or severity (FDR$_{TWAS}$ < 0.01) (Fig. 6). For example, two variants in high LD at *DR1* (rs569414 and rs1559828, $r^2$ > 0.73) displayed extremely high levels of population differentiation, probably due to selection outside Africa (DAF = 0.13 in Africa versus higher than 0.62 in Eurasia; Extended Data Fig. 10d). DR1 suppresses type I IFN responses[49] and the selected alleles, which decrease COVID-19 severity, reduce *DR1* expression in most immune cells (Fig. 6). Likewise, an approximately 39 kb Neanderthal haplotype, spanning the *MUC20* locus in Eurasians, contains the rs2177336-T allele that increases *MUC20* expression in SARS-CoV-2-stimulated cells, particularly for CD4+ T cells, and decreases COVID-19 susceptibility (Fig. 6). Together, these results reveal how past selection or Neanderthal introgression have impacted immune responses that contribute to present-day disparities in COVID-19 risk.

## Discussion

Here we show that cell type composition is a major driver of population differences in immune responses to SARS-CoV-2. The higher proportions

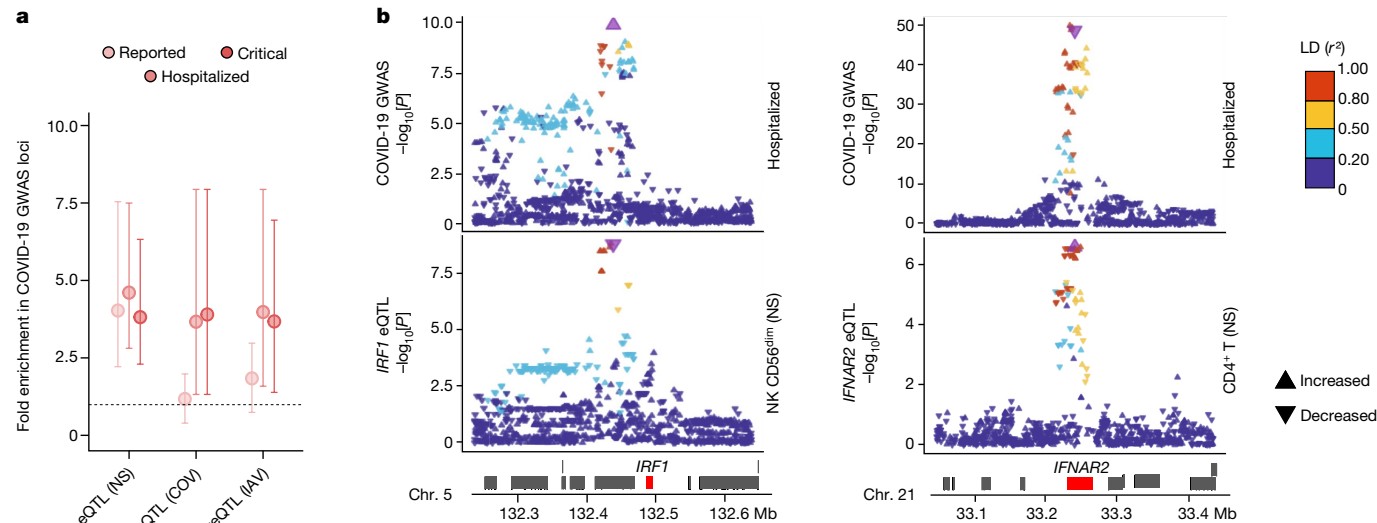

**Fig. 5 | eQTLs and reQTLs contribute to COVID-19 risk. a**, Enrichment in GWAS loci associated with COVID-19 susceptibility and severity at eQTLs and reQTLs. Data are the mean and 2.5th–97.5th percentiles (95% CIs) of fold enrichments observed over $n = 10,000$ resamplings. **b**, Colocalization of *IRF1* and *IFNAR2* eQTLs with COVID-19 severity loci. Top, the $-\log_{10}[P]$ profiles (two-sided Student's *t*-tests) for association with COVID-19-related hospitalization.

Bottom, the $-\log_{10}[P]$ profiles for association with expression in non-stimulated CD56$^{dim}$ NK cells (*IRF1*) and CD4$^+$ T cells (*IFNAR2*). The colour code reflects the degree of LD ($r^2$) with the consensus SNP identified by colocalization analyses (purple). For each SNP, the direction of the arrow indicates the direction of the effect. Chr., chromosome.

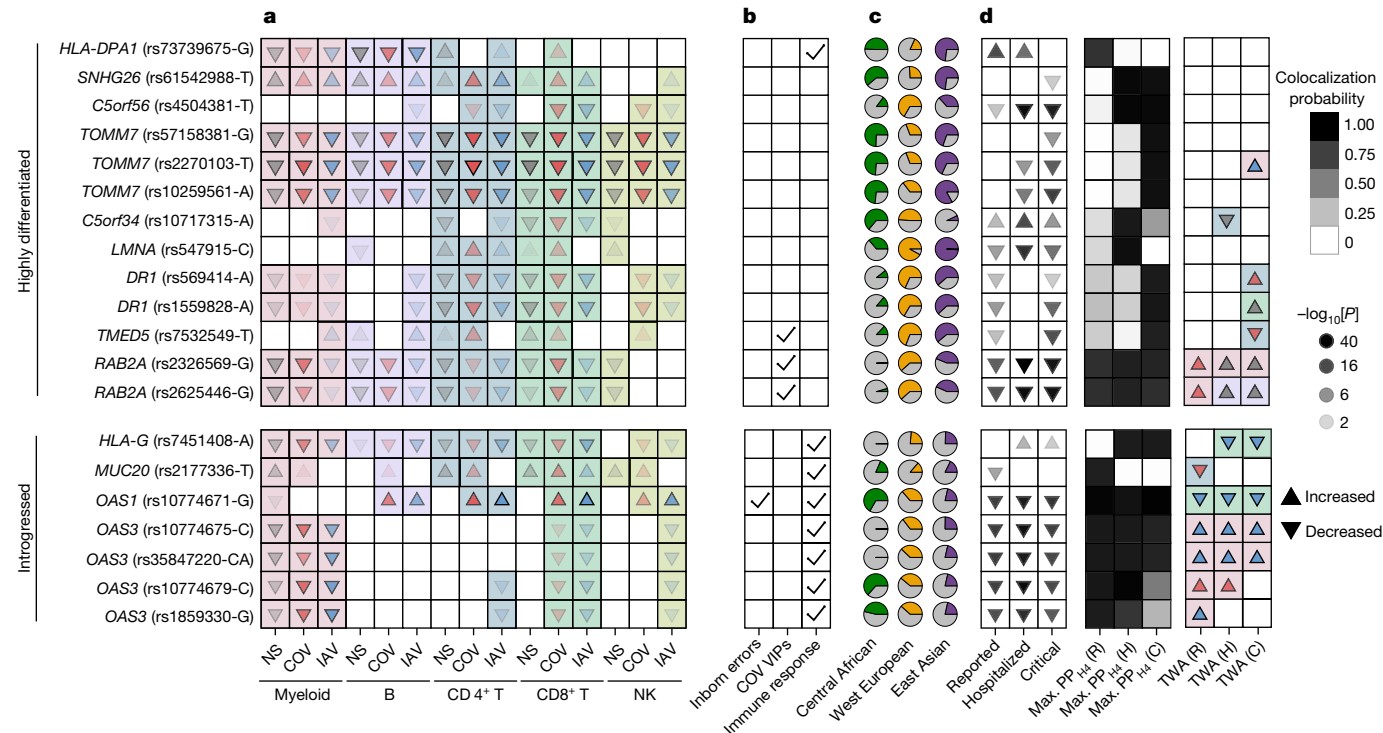

**Fig. 6 | Adaptation and archaic introgression at COVID-19-associated (r) eQTLs. a–d**, Features of (r)eQTLs colocalizing with COVID-19 risk loci (PP$_{H4}$ > 0.8) and presenting either strong population differentiation (top 1% PBS genome-wide) or evidence of Neanderthal introgression. **a**, Effects of the target allele on gene expression across immune lineages and stimulation conditions. **b**, Clinical and functional annotations of associated genes. **c**, Present-day population frequencies of the target allele. **d**, The effects of the target allele on COVID-19 risk (infection, hospitalization and critical state), colocalization probability and the lineage and condition in which gene expression most likely affects COVID-19 risk as detected by transcriptome-wide association (TWA) analyses. For expression or COVID-19 associations, the arrows indicate increases/decreases in expression or disease risk with each copy of the target allele, and the opacity reflects the strength of association (two-sided Student's

*t*-test $-\log_{10}[P]$). For the TWA analysis, the arrows indicate the effect of an increase in gene expression on the risk of COVID-19. In **a** and **d**, the arrow colours indicate stimulation conditions (non-stimulated (grey), SARS-CoV-2-stimulated (red), IAV-stimulated (blue)) and the background colour indicates the lineage (myeloid (pink), B (purple), CD4$^+$ T (blue), CD8$^+$ T (green), NK (light green)). For each eQTL, the target allele is defined as (1) the derived allele for highly differentiated eQTLs or (2) the allele that segregates with the archaic haplotype for introgressed eQTLs. When the ancestral state is unknown, the minor allele is used as a proxy for the derived allele. Note that, in some cases (for example, *OAS1*), the introgressed allele can be present in Africa, which is attributed to the reintroduction in Eurasia of an ancient allele by Neanderthals[46]. C, critical; H, hospitalized; R, reported.

of memory cells in lymphoid lineages from individuals of African descent, along with their association with CMV infection, highlight how previous environmental exposures can contribute to population disparities in cellular activation states. Neglecting socioenvironmental factors that covary with ancestry may therefore inflate the estimated effects of genetic ancestry on phenotypic variation. One such factor is CMV, affecting leukocyte responses to SARS-CoV-2, but the impact of other exposures on population variation in immune responses remains to be determined. Common genetic variants can also contribute to immune response variation, but their effects primarily apply to a subset of genes showing strong population differentiation. This is illustrated by the rs1142888-G allele, which accounts for the greater than 2.8-fold higher levels of *GBP7* expression in response to viral stimulation in Europeans compared with in Africans. The higher frequency of this allele in Europe probably results from selection occurring 21,900–35,600 years ago. GBP7 facilitates IAV replication by suppressing innate immunity[50], but also regulates host defence to intracellular bacteria such as *Listeria monocytogenes* and *Mycobacterium tuberculosis*[51], providing a plausible mechanism for positive selection at this locus.

This study also shows that natural selection and Neanderthal introgression contributed to differentiate present-day immune responses to SARS-CoV-2. We found traces of selection targeting SARS-CoV-2-specific reQTLs around 25,000 years ago in the ancestors of East Asians, coinciding with the proposed timing of an epidemic that affected the evolution of host coronavirus-interacting proteins[23,24]. However, there is little overlap between alleles selected during this period and variants underlying COVID-19 risk, suggesting changes in the genetic basis of infectious diseases over time, possibly due to the evolution of viruses themselves. Nevertheless, we identified cases (for example, *DR1*, *OAS1-3*, *TOMM7*, *MUC20*) in which selection or archaic introgression contributed to changes in both SARS-CoV-2 immune responses and COVID-19 outcome. Studies based on ancestry-aware polygenic risk scores from cross-population GWAS will be required to establish a formal link between past adaptation and present-day population differences in COVID-19 risk.

Finally, the genetic dissection of variation in transcriptional responses to SARS-CoV-2 provides mechanistic insights into the effects of alleles that are associated with COVID-19 risk. Variants of *IRF1*, *IFNAR2* and *DR1* associated with lower COVID-19 severity increase type I IFN signalling in lymphoid cells by upregulating *IRF1* and *IFNAR2* or down-regulating *DR1*, attesting to the importance of efficient IFN signalling for a favourable clinical outcome[4,12–14]. Another example is *MUC20*, at which we identified a Neanderthal-introgressed eQTL that increases *MUC20* expression in SARS-CoV-2-stimulated CD4+ T cells and decreases COVID-19 susceptibility. Given the role of mucins in forming a barrier against infection in the respiratory tract, the high *MUC20* expression in ciliated epithelial cells from the bronchus[52] and the detection of the *MUC20* eQTL in pulmonary tissue (Supplementary Note 11), we suggest that the greater resistance to infection conferred by the Neanderthal haplotype may result from a similar effect on *MUC20* expression in the respiratory tract.

We note two main limitations of our results. First, our samples mostly originate from male individuals, so the impact of sex on immune variation was not addressed. Sex has a widespread yet moderate effect on both transcriptional responses to microbial threats[53] and the genetic regulation of gene expression[54], supporting the transferability of our main conclusions. Nonetheless, examining sex-balanced cohorts will enable the characterization of possible sex-specific differences at the population scale. Second, given the sample sizes and cell counts needed to accurately define population variation in immune activity, we focused on a single system (PBMCs) and selected viral strains. Although PBMCs constitute a valuable model to characterize peripheral immune activation by SARS-CoV-2[9,10], they provide an incomplete representation of the pulmonary epithelium—the primary infection site for respiratory viruses. However, we found that 38% of the eQTLs identified in this

study are also detected in lung tissue[55], rising to 72% for eQTLs shared across immune lineages (Supplementary Note 11 and Supplementary Table 9b). Further studies are needed to examine the transferability of our findings to other cell types and to investigate how diverse viral strains affect the dynamics of host responses to SARS-CoV-2.

Overall, our results highlight the value of single-cell approaches in capturing the full diversity of peripheral immune responses to RNA viruses, particularly SARS-CoV-2, and provide insights into environmental, genetic and evolutionary drivers of immune response variation across individuals and populations.

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

¹Human Evolutionary Genetics Unit, Institut Pasteur, Université Paris Cité, CNRS UMR2000, Paris, France. ²Collège Doctoral, Sorbonne Université, Paris, France. ³Insect-Virus Interactions Unit, Institut Pasteur, Université Paris Cité, CNRS UMR2000, Paris, France. ⁴Cytometry and Biomarkers UTechS, Institut Pasteur, Université Paris Cité, Paris, France. ⁵Translational Immunology Unit, Institut Pasteur, Université Paris Cité, Paris, France. ⁶Université Paris Cité, Imagine Institute, Laboratory of Inflammatory Responses and Transcriptomic Networks in Diseases, Atip-Avenir Team, INSERM UMR1163, Paris, France. ⁷Labtech Single-Cell@Imagine, Imagine Institute, INSERM UMR1163, Paris, France. ⁸Center for Molecular Medicine and Genetics, Wayne State University, Detroit, MI, USA. ⁹Department of Obstetrics and Gynecology, Wayne State University, Detroit, MI, USA. ¹⁰Department of Biology, University of Rome Tor Vergata, Rome, Italy. ¹¹Structural Virology Unit, Institut Pasteur, Université Paris Cité, CNRS UMR3569, Paris, France. ¹²Virus and Immunity Unit, Institut Pasteur, Université Paris Cité, CNRS UMR3569, Paris, France. ¹³Ghent University and University Hospital, Ghent, Belgium. ¹⁴Hong Kong Red Cross Blood Transfusion Service, Hospital Authority, Hong Kong SAR, China. ¹⁵WHO Collaborating Centre for Infectious Disease Epidemiology and Control, School of Public Health, Li Ka Shing Faculty of Medicine, The University of Hong Kong, Hong Kong SAR, China. ¹⁶Laboratory of Data Discovery for Health (D24H), Hong Kong Science Park, Hong Kong SAR, China. ¹⁷Division of Public Health Laboratory Sciences, School of Public Health, Li Ka Shing Faculty of Medicine, The University of Hong Kong, Hong Kong SAR, China. ¹⁸HKU-Pasteur Research Pole, School of Public Health, The University of Hong Kong, Hong Kong SAR, China. ¹⁹Centre for Immunology and Infection, Hong Kong Science Park, Hong Kong SAR, China. ²⁰St Giles Laboratory of Human Genetics of Infectious Diseases, The Rockefeller University, New York, NY, USA. ²¹Laboratory of Human Genetics of Infectious Diseases, INSERM UMR1163, Necker Hospital for Sick Children, Paris, France. ²²Université Paris Cité, Imagine Institute, Paris, France. ²³Department of Pediatrics, Necker Hospital for Sick Children, Paris, France. ²⁴Howard Hughes Medical Institute, New York, NY, USA. ²⁵Department of Microbiology and Immunology, Peter Doherty Institute for Infection and Immunity, University of Melbourne, Melbourne, Victoria, Australia. ²⁶Chair Human Genomics and Evolution, Collège de France, Paris, France. ²⁷These authors contributed equally: Yann Aquino, Aurélie Bisiaux, Zhi Li, Mary O'Neill. ²⁸These authors jointly supervised this work: Maxime Rotival, Lluis Quintana-Murci. ✉e-mail: maxime.rotival@pasteur.fr; quintana@pasteur.fr

## Methods

### Sample collection

The individuals of self-reported African (AFB) and European (EUB) descent studied are part of the EVOIMMUNOPOP cohort[18]. In brief, 390 healthy male donors (188 AFB and 202 EUB) were recruited between 2012 and 2013 in Ghent (Belgium), thus, before the COVID-19 pandemic. Blood was obtained from the healthy volunteers, and the PBMC fraction was isolated and frozen. Inclusion in the current study was restricted to 80 nominally healthy individuals of each ancestry, aged between 19 and 50 years at the time of sample collection. Donors of African descent originated from West Central Africa, with >90% being born in either Cameroon or the Democratic Republic of Congo. For this study, 71 additional individuals of East Asian descent (ASH) were included, of whom 62 were retained after quality control (see the 'scRNA-seq library preparation and data processing' section). ASH individuals were recruited at the School of Public Health, University of Hong Kong, and were included in a community-based sero-epidemiological COVID-19 study (research protocol number JTW 2020.02). Inclusion for the study described here was restricted to nominally healthy ASH individuals (30 men and 41 women) aged between 19 and 65 years of age and seronegative for SARS-CoV-2. Samples were collected at the Red Cross Blood Transfusion Service (Hong Kong) where the PBMC fraction was isolated and frozen. Target sample sizes were determined to ensure >80% power for the detection of eQTLs with R2 higher than 0.2, at a $P < 5 \times 10^{-9}$ threshold.

In this study, we refer to individuals of Central African (AFB), West European (EUB) and East Asian (ASH) ancestries to describe individuals who are genetically similar (that is, lowest $F_{ST}$ values) to populations from West-Central Africa, Western Europe and East Asia, using the 1000 Genomes (1KG) Project[56] data as a reference (Supplementary Fig. 1a). Notably, the AFB, EUB and ASH samples present no evidence of recent genetic admixture with populations originating from another continent, besides two AFB donors who respectively present 22% of Near Eastern- and 25% of European-ancestries. Such a moderate level of admixture in fewer than 1% of individuals is unlikely to have any significant impact on the results presented.

All of the samples were collected after written informed consent was obtained from the donors, and the study was approved by the ethics committee of Ghent University (B670201214647), the Institutional Review Board of the University of Hong Kong (UW 20-132), and the relevant French authorities (CPP, CCITRS and CNIL). This study was also monitored by the Ethics Board of Institut Pasteur (EVOIMMUNOPOP-281297).

### Genome-wide DNA genotyping

The AFB and EUB individuals were previously genotyped at 4,301,332 SNPs, using the Omni5 Quad BeadChip (Illumina) with processing as previously described[18]. The additional 71 ASH donors were genotyped separately at 4,327,108 SNPs using the Infinium Omni5-4 v.1.2 BeadChip (Illumina). We updated SNP identifiers based on Illumina annotation files (https://support.illumina.com/content/dam/illumina-support/documents/downloads/productfiles/humanomni5-4/v1-2/infinium-omni5-4-v1-2-a1-b144-rsids.zip) and called the genotypes of all ASH individuals jointly on GenomeStudio (v.2011.1; https://www.illumina.com/techniques/microarrays/array-data-analysis-experimental-design/genomestudio.html). We then removed SNPs with (1) no 'rs' identifiers or with no assigned chromosome or genomic position ($n = 14,637$); (2) duplicated identifiers ($n = 5,059$); or (3) a call rate of <95% ($n = 10,622$). We then used the 1KG Project Phase 3 data[56] as a reference for merging the ASH genotyping data with that of AFB and EUB individuals and detecting SNPs misaligned between the three genotype datasets. Before merging, we removed SNPs that (1) were absent from either the Omni5 or 1KG datasets ($n = 469,535$); (2) were transversions ($n = 138,410$); (3) had incompatible alleles between datasets, before and after allele flipping ($n = 1,250$); and (4) had allele frequency differences of more

than 20% between the AFB and Luhya from Webuye, Kenya (LWK) and Yoruba from Ibadan, Nigeria (YRI), or between the EUB and Utah residents with Northern and Western European ancestries (CEU) and British individuals from England and Scotland (GBR), or between the ASH and Southern Han Chinese (CHS) ($n = 777$). Once the data had been merged, we performed principal component analysis (PCA) using PLINK (v.1.9)[57] and ensured that the three study populations (that is, AFB, EUB and ASH) overlapped with the corresponding 1KG populations, to exclude batch effects between genotyping platforms (Supplementary Fig. 1a). The final genotyping dataset included 3,723,840 SNPs.

### Haplotype phasing and imputation

After merging genotypes from AFB, EUB and ASH donors, we filtered genotypes for duplicates with bcftools norm --rm-dup all (v.1.16)[58] and lifted all genotypes over to the human genome assembly GRCh38 with GATK's (v.4.1.2.0) LiftoverVcf using the RECOVER_SWAPPED_ALT_REF=TRUE option[59]. We then filtered out duplicated variants again before phasing genotypes with SHAPEIT4 (v.4.2.1)[60] and imputing missing variants with Beagle5.1 (v.18May20.d20)[61], treating each chromosome separately. For both phasing and imputation, we used the genotypes of 2,504 unrelated individuals from the 1KG Project Phase 3 data as a reference (downloaded from http://ftp.1000genomes.ebi.ac.uk/vol1/ftp/release/20130502 and lifted over to GRCh38) and downloaded genetic maps from the GitHub pages of the associated software (that is, SHAPEIT4 for phasing and *Beagle5.1* for imputation). A third round of duplicate filtering was performed after phasing and before imputation using Beagle5.1 (v.18May20.d20)[61]. Phasing was performed using the setting --pbwt-depth=8 and imputation was performed assuming an effective population size ($N_e$) of 20,000. The quality of imputation was assessed by cross-validation; specifically, we performed 100 independent rounds of imputation excluding 1% of the variants and compared the imputed allelic dosage with the observed genotypes for these variants (Supplementary Fig. 1b,c). The results obtained confirmed that imputation quality was satisfactory, with 98% of common variants (that is, MAF > 5%) having an $r^2 > 0.8$ for the correlation between observed and imputed genotypes (>95% concordance for 96% of common variants). After imputation, variants with a MAF < 1% or with a low predicted quality of imputation (that is, DR2 < 0.9) were excluded, yielding a final dataset of 13,691,029 SNPs for downstream analyses.

### Viruses used in this study

To evaluate population differences in the immune responses to SARS-CoV-2, we chose the viral strain that circulated in France at the time of our experiments (Autumn 2020). This reference strain (BetaCoV/France/GE1973/2020) was supplied by the National Reference Centre for Respiratory Viruses hosted by Institut Pasteur and headed by S. van der Werf. The human sample from which the strain was isolated was provided by L. Andreoletti from the Robert Debré Hospital. To characterize the distinctive features of SARS-CoV-2 responses, we included in our study the IAV as a reference comparison of another respiratory RNA virus. Specifically, we chose the PR8 strain (IAV, PR/8, H1N1/1934) based on our previous experience with this virus, its availability in the laboratory and its ability to trigger IFN responses in healthy human donors[53,62]. The PR8 strain used was purchased from Charles River Laboratories (3X051116) and provided in ready-to-use aliquots that were stored at −80 °C.

### SARS-CoV-2 stock production

To produce SARS-CoV-2, we used African green monkey kidney Vero E6 cells that were tested for mycoplasma contamination and maintained at 37 °C in 5% $CO_2$ in Dulbecco's minimum essential medium (DMEM) (Sigma-Aldrich) supplemented with 10% fetal bovine serum (FBS, Dutscher) and 1% penicillin–streptomycin (Gibco, Thermo Fisher Scientific). Vero E6 cells were plated at 80% confluence in 150 $cm^2$ flasks

and infected with SARS-CoV-2 at a multiplicity of infection (MOI) of 0.01 in DMEM supplemented with 2% FBS and 1% penicillin–streptomycin. After 1 h, the inoculum was removed and replaced with DMEM supplemented with 10% FBS and 1% penicillin–streptomycin, and cells were incubated for 72 h at 37 °C in 5% $CO_2$. The cell culture supernatant was collected and centrifuged for 10 min at 3,000 rpm to remove cellular debris, and polyethylene glycol (PEG; PEG8000, Sigma-Aldrich) precipitation was performed to concentrate the viral suspension. In brief, 1 l of viral stock was incubated with 250 ml of 40% PEG solution (that is, 8% PEG final) overnight at 4 °C. The suspension was centrifuged at 10,000$g$ for 30 min at 4 °C and the resulting pellet was resuspended in 100 ml of RPMI medium (Gibco, Thermo Fisher Scientific) supplemented with 10% FBS (hereafter R10) and viral aliquots were stored at −80 °C. SARS-CoV-2 viral titres were determined using a focus-forming unit assay as previously described[63]. In brief, Vero E6 cells were plated in a 96-well plate with $2 \times 10^4$ cells per well. The cellular monolayer was infected with serial dilutions (1:10) of viral stock and overlaid with a semi-solid 1.5% carboxymethylcellulose (Sigma-Aldrich) and 1× MEM medium for 36 h at 37 °C. Cells were then fixed with 4% paraformaldehyde (Sigma-Aldrich), and permeabilized with 1× phosphate-buffered saline, 0.5% Triton X-100 (Sigma-Aldrich). Infectious foci were stained with a human anti-SARS-CoV-2 spike antibody (H2-162, Hugo Mouquet's laboratory, Institut Pasteur) and the corresponding HRP-conjugated secondary antibody (Sigma-Aldrich). Foci were stained using a 3,3'-diaminobenzidine staining solution (DAB, Sigma-Aldrich) and counted using the BioSpot suite of the C.T.L. ImmunoSpot S6 Image Analyzer.

## In vitro peripheral blood mononuclear cell stimulation

We performed scRNA-seq analysis of SARS-CoV-2-, IAV- and mock-stimulated (referred to as the non-stimulated condition) PBMCs from healthy donors (80 AFB, 80 EUB and 71 ASH) in 16 experimental runs. We first performed a kinetic experiment (run 1) on samples from 4 AFB and 4 EUB individuals stimulated for 0, 6 and 24 h to validate our in vitro model across different timepoints (Supplementary Note 1, Supplementary Fig. 2 and Supplementary Table 2). The 6 h timepoint was identified as the optimal timepoint for the analysis (Supplementary Note 1). We then processed the rest of the cohort, over runs 2 to 15. Finally, we reprocessed some samples (run 16) to assess the technical variability in our setting and to increase in silico cell counts (see the 'scRNA-seq library preparation and data processing' section). Ancestry-related batch effects were minimized by scheduling sample processing to ensure a balanced distribution of AFB, EUB and ASH donors within each run. Donors used for feach run were randomly selected within each population.

For each run, cryopreserved PBMCs were thawed in a 37 °C water bath, transferred to 25 ml of R10 medium (that is, RPMI 1640 supplemented with 10% heat-inactivated FBS) at 37 °C, and centrifuged at 300$g$ for 10 min at room temperature. Cells were counted, resuspended at $2 \times 10^6$ cells per ml in warm R10 in 25 cm$^2$ flasks, and rested overnight (that is, 14 h) at 37 °C. The next morning, PBMCs were washed and resuspended at a density of $3.3 \times 10^6$ cells per ml in R10; 120 μl of a suspension containing $4 \times 10^5$ cells from each sample was then plated in a 96-well untreated plate (Greiner Bio-One) for each of the three sets of stimulation conditions. We added 80 μl of either R10 (non-stimulated), SARS-CoV-2 or IAV stock (corresponding to $4 \times 10^5$ focus-forming units diluted in R10) to the cells, so as to achieve a multiplicity of infection (MOI) of 1 and an optimal PBMC concentration of $2 \times 10^6$ cells per ml. Cells were incubated at 37 °C for 0, 6 or 24 h for the kinetic experiment (run 1), and for 6 h for all subsequent runs (runs 2 to 16), in a biosafety level 3 (BSL-3) facility at Institut Pasteur, Paris. The plates were then centrifuged at 300$g$ for 10 min and supernatants were stored at −20 °C until use (see 'Supernatant cytokine assays' section). All of the samples from the same run were resuspended in Dulbecco's phosphate-buffered saline (Gibco), supplemented with 0.04% bovine serum albumin (BSA, Miltenyi Biotec), and multiplexed in eight pools

according to a pre-established study design (Supplementary Fig. 3a and Supplementary Table 3a). The cells from each pool were counted using the Cell Countess II automated cell counter (Thermo Fisher Scientific) and the cell density was adjusted to 1,000 viable cells per μl of 0.04% BSA in phosphate-buffered saline. When performing stimulations, researchers were blinded to the genotype and environmental exposures of the individual.

## scRNA-seq library preparation and data processing

We generated scRNA-seq cDNA libraries using the Chromium Controller (10x Genomics) according to the manufacturer's instructions for the Chromium Single Cell 3' Library and Gel Bead Kits (v.3.1). Library quality and concentration were assessed using the Agilent 2100 Bioanalyzer and a Qubit fluorometer (Thermo Fisher Scientific). The final products were processed for high-throughput sequencing on a HiSeqX platform (Illumina).

Paired-end sequencing reads from each of the 133 scRNA-seq cDNA libraries (13 libraries from the kinetic experiment and 120 from the population-level study) were independently mapped onto the concatenated human (GRCh38), SARS-CoV-2 (hCoV-19/France/GE1973/2020) and IAV (A/Puerto Rico/8/1934(H1N1)) genome sequences using the STARsolo aligner (v.2.7.8a)[64] (Supplementary Fig. 3b). We obtained a mean of 10,785 cell-containing droplets per library, and each droplet was assigned to its sample of origin with Demuxlet (v.0.1)[65], based on the genotyping data available for each individual. Singlet/doublet calls were compared with the output of Freemuxlet (v.0.1)[65] to ensure good agreement (Supplementary Fig. 3c–e). We loaded feature-barcode matrices for all cell-containing droplets identified as singlets by Demuxlet in each scRNA-seq library onto a SingleCellExperiment (v.1.14.1) object[66]. Data from barcodes associated with low-quality or dying cells were removed with a hard threshold-based filtering strategy based on three metrics: cells with fewer than 1,500 total unique molecular identifier (UMI) counts, 500 detected features or a mitochondrial gene content exceeding 20% were removed from each sequencing library (Supplementary Fig. 3f). We also discarded samples from nine ASH donors from whom fewer than 500 cells were obtained in at least one condition (Supplementary Fig. 3g).

We then log-normalized raw UMI counts with a unit pseudocount and library size factors (that is, the number of reads associated with each barcode) were calculated with quickClusters and computeSumFactors from the scran package (v.1.20.1)[67]. We then calculated the mean and variance of log-transformed counts for each gene and broke the variance down into a biological and a technical component with the fitTrendPoisson and modelGeneVarByPoisson functions of scran. This approach assumes that technical noise is Poisson-distributed and simulates Poisson-distributed data to derive the mean-variance relationship expected in the absence of biological variation. Excess variance relative to the null hypothesis is considered to correspond to the biological variance. We retained only those genes for which the biological variance component was positive with an FDR below 1%. We used this filtered feature set and the technical variance component modelled from the data to run PCA with denoisePCA from scran, thus discarding later components more likely to capture technical noise. Doublets (that is, barcodes assigned to cells from different individuals captured in the same droplet) are likely to be in close neighbourhoods when projected onto a subspace of the data of lower dimensionality. We therefore used a $k$-nearest neighbours approach to discard cryptic doublets (that is, barcodes associated to different cells from the same individual captured in the same droplet). Barcodes identified as singlets by Demuxlet but having over 5 out of 25 doublet nearest neighbours in the PCA space were reassigned as doublets and excluded from further analyses (Supplementary Fig. 3h).

After data preprocessing, we performed a second round of UMI count normalization, feature selection and dimensionality reduction on the cleaned data to avoid bias due to the presence of low-quality cells and

cryptic doublets. Differences in sequencing depth were equalized between batches (that is, sequencing libraries) using multiBatchNorm from batchelor (v.1.8.1) to scale library size factors according to the ratio of mean counts between batches[68] (Supplementary Fig. 3i). We accounted for the different mean-variance trends in each batch, by applying modelGeneVarByPoisson separately for each sequencing library, and then combining the results for all batches with combineVar from scran[67]. We then bound all 133 separate preprocessed feature-barcode matrices into a single merged SingleCellExperiment object, log-normalized UMI counts according to the scaled size factors and selected genes with mean log-expression values over 0.01 or a biological variance compartment exceeding 0.001 (Supplementary Fig. 3j). On the basis of this set of highly variable genes and the variance decomposition, we then performed PCA on the whole dataset using denoisePCA, and then used Harmony (v.0.1.0) on the PCs to adjust for library effects[69].

### Clustering and cell type assignment

We performed cluster-based cell type identification in each stimulation condition, according to a four-step procedure. We first performed low-resolution (res. parameter = 0.8) shared nearest-neighbour graph-based ($k$ = 25) clustering using FindClusters from Seurat (v.4.1.1) with assignment to one of three meta-clusters (that is, myeloid, B lymphoid and T/NK lymphoid) on the basis of the transcriptional profiles of the cells for canonical markers (for example, *CD3E-F*, *CD14*, *FCGR3A*, *MS4A1*) (Supplementary Fig. 4a,b). We next performed a second round of clustering at higher resolution (res. parameter = 3) within each meta-cluster and stimulation condition (Supplementary Fig. 4c). We systematically tested for differential expression between each cluster and the other clusters of the same meta-cluster and stimulation condition. This made it possible to define unbiased markers ($|\log_2FC| \neq 0$, FDR < 0.01) for each cluster (Supplementary Fig. 4d). We then used these expression profiles of these genes to assign each cluster manually to one of 22 different cell types (Supplementary Fig. 4e), which, for some analyses, were collapsed into five major immune lineages. This step was performed in parallel by three investigators to consolidate consensus assignments. We also used cellular indexing of transcriptomes and epitopes by sequencing (CITE-seq) data, generated for a subset of cells (2% of the whole dataset), to validate our assignments and redefine clusters presenting ambiguous transcriptional profiles (for example, memory-like NK cells; Supplementary Fig. 4f).

By calling cell types from high-resolution, homogeneous clusters, assigned independently for each lineage and stimulation condition (that is, non-stimulated, SARS-CoV-2, and IAV), we were able to preserve much of the diversity in our dataset, while avoiding potential confounding effects due to the stimulation conditions. However, some clusters were characterized by markers associated with different cell types. Most of these clusters corresponded to mixtures of similar cell types (for example, the expression of *CD3E*, *CD8A*, *NKG7* and *CD16* suggested a mixture of cytotoxic CD8+ T and NK cells) and were consistent with the known cell hierarchy. Other, less frequent clusters expressed a combination of markers usually associated with lineages originating from different progenitors (for example, *CD3E* and *CD19*, associated with T and B lymphocytes, respectively). These clusters were considered to be incoherent and were discarded. In the fourth and final step of our procedure, we used linear discriminant analysis to resolve within each condition the mixtures that were consistent with the established cell hierarchy, to obtain a final cell assignment (Supplementary Fig. 4g,h). For clusters of mixed identity AB, we built a training dataset from 10,000 observations sampled from the set of cells called as A or B, preserving the corresponding frequencies of these cells in the whole dataset. We then used a model trained on these data to predict the specific cellular identities within the mixed cluster.

Cell abundance from each immune lineage/cell type was compared between non-stimulated and SARS-CoV-2-/IAV-stimulated conditions,

using Wilcoxon's signed-rank test matching on the individual. FDR was calculated across all conditions and lineages with the Benjamini–Hochberg procedure (p.adjust function with the 'fdr' method). Viral stimulation had a moderate effect on the estimated cell proportions and, although significant differences were detected, the total number of cells per cell type was generally consistent across conditions (Supplementary Note 2 and Supplementary Table 3e).

### Cellular indexing of transcriptomes and epitopes by sequencing

To confirm the identity of specific cell types expressing ambiguous markers at the RNA level, during the last experimental run (run 16), half the cells from each experimental condition were used to perform CITE-seq, according to the manufacturer's instructions (10x Genomics). PBMCs were washed, resuspended in chilled 1% BSA in phosphate-buffered saline and incubated with human TruStain FcX blocking solution (BioLegend) for 10 min at 4 °C. Cells were then stained with a cocktail of TotalSeq-B antibodies (BioLegend) previously centrifuged at 14,000*g* for 10 min (Supplementary Table 3b). The cells were incubated for 30 min at 4 °C in the dark and were then washed three times. Cell density was then adjusted to 1,000 viable cells per µl in 1% BSA in phosphate-buffered saline. We generated scRNA-seq libraries and cell protein libraries (L131–L134) with the Chromium Single Cell 3′ Reagent Kit (v.3.1), using the Feature Barcoding technology for Cell Surface Proteins (10x Genomics).

### Supernatant cytokine assays

Before protein analysis, sample supernatants were treated in the BSL-3 facility to inactivate the viruses, according to a published protocol for SARS-CoV[70], which we validated for SARS-CoV-2. In brief, all of the samples were treated with 1% (v/v) Triton X-100 (Sigma-Aldrich) for 2 h at room temperature, which effectively inactivated both SARS-CoV-2 and IAV. The protein concentration was then determined with a commercial Luminex multi-analyte assay (Biotechne, R&D Systems) and the SIMOA Homebrew assay (Quanterix). For the Luminex assay, we used the XL Performance Kit according to the manufacturer's instructions, and proteins were determined using the Bioplex 200 (Bio-Rad) system. Furthermore, IFNα, IFNγ (duplex) and IFNβ (single-plex) protein concentrations were quantified in SIMOA digital ELISA tests developed as Quanterix Homebrews according to the manufacturer's instructions (https://portal.quanterix.com/). The SIMOA IFNα assay was developed with two autoantibodies specific for IFNα isolated and cloned (Evitria) from two patients with autoimmune polyglandular syndrome type 1 (also known as autoimmune polyendocrinopathy-candidiasis-ectodermal dystrophy)[71] and covered by patent application WO2013/098419. These antibodies can be used for the quantification of all IFNα subtypes with a similar sensitivity. The 8H1 antibody clone was used to coat paramagnetic beads at a concentration of 0.3 mg ml$^{-1}$ for use as a capture antibody. The 12H5 antibody was biotinylated (biotin/antibody ratio = 30:1) and used as the detector antibody, at a concentration of 0.3 µg ml$^{-1}$. The SBG enzyme for detection was used at a concentration of 150 pM. Recombinant IFNα17/αI (PBL Assay Science) was used as calibrator. For the IFNγ assay, the MD-1 antibody clone (BioLegend) was used to coat paramagnetic beads at a concentration of 0.3 mg ml$^{-1}$ for use as a capture antibody. The MAB285 antibody clone (R&D Systems) was biotinylated (biotin/antibody ratio = 40:1) and used as the detector antibody at a concentration of 0.3 µg ml$^{-1}$. The SBG enzyme used for detection was used at a concentration of 150 pM. Recombinant IFNγ protein (PBL Assay Science) was used as a calibrator. For the IFNβ assay, the 710322-9 IgG1, kappa, mouse monoclonal antibody (PBL Assay Science) was used to coat paramagnetic beads at a concentration of 0.3 mg ml$^{-1}$, for use as a capture antibody. The 710323-9 IgG1 kappa mouse monoclonal antibody was biotinylated (biotin/antibody ratio = 40:1) and used as the detector antibody at a concentration of 1 µg ml$^{-1}$. The SBG enzyme for detection was used at a concentration of 50 pM. Recombinant IFNβ protein (PBL Assay Science) was used

as a calibrator. The limit of detection of these assays was 0.8 fg ml$^{-1}$ for IFNα, 20 fg ml$^{-1}$ for IFNγ and 0.2 pg ml$^{-1}$ for IFNβ, considering the dilution factor of 10.

## Flow cytometry

Frozen PBMCs from three AFB (CMV$^+$) and six EUB (three CMV$^+$, three CMV$^-$) donors were thawed and allowed to rest overnight, as previously described. For each donor, $10^6$ cells were resuspended in phosphate-buffered saline supplemented with 2% FBS and incubated with human Fc blocking solution (BD Biosciences) for 10 min at 4 °C. Cells were then stained with the following antibodies for 30 min at 4 °C: CD3 VioGreen (BW264/56, Miltenyi Biotec), CD14 V500 (M5E2, BD Biosciences), CD57 Pacific Blue (HNK-1, BioLegend), NKp46 PE (9E2/NKp46, BD Biosciences), CD16 PerCP-Cy5.5 (3G8, BD Biosciences), CD56 APC-Vio770 (REA196, Miltenyi Biotec), NKG2A FITC (REA110, Miltenyi Biotec) and NKG2C APC (REA205, Miltenyi Biotec). The cells were then washed and acquired on the MACSQuant cytometer (Miltenyi Biotec), and the data were analysed using FlowJo (v.10.7.1)[72].

## Quantification of batch effects and replicability

Once all the samples had been processed, we used the kBET metric (v.0.99.6)[73] to assess the intensity of batch effects and to quantify the relative effects of technical and biological variation on cell clustering. This made it possible to confirm that the variation across libraries, and across experimental runs, remained limited relative to the variation across individuals or across conditions (Supplementary Fig. 5a). We used technical replicates to assess the replicability of our observations across independent stimulations. Agreement was good between the cell proportions and the ISG activity scores inferred across independent runs ($r > 0.82$, $P < 7.6 \times 10^{-13}$) (Supplementary Fig. 5b,c).

## Pseudobulk estimation, normalization and batch correction

Individual variation in gene expression was quantified at two resolutions: five major immune lineages and 22 cell types. We aggregated raw UMI counts from all high-quality single-cell transcriptomes ($n = 1,047,824$) into bulk expression estimates by summing gene expression values across all cells assigned to the same lineage/cell type and sample (that is, individual and stimulation conditions) using the aggregateAcrossCells function of scuttle (v.1.2.1)[74]. We then normalized the raw aggregated UMI counts by library size, generating 3,330 lineage-wise (222 donors × 3 sets of conditions × 5 lineages) and 14,652 cell type-wise (666 samples × 22 cell types) pseudobulk counts-per-million (CPM) values, for all genes in our dataset. CPM values were then log$_2$-transformed, with an offset of 1 to prevent non-finite values and to stabilize variation for weakly expressed genes. Genes with a mean CPM < 1 across all conditions and lineages/cell types were considered to be non-expressed and were discarded from further analyses, leading to a final set of 12,667 genes at the lineage level (12,672 genes when increasing granularity to 22 cell types), including 12 viral transcripts. To quantify the experimental variation induced by the experimental run, library preparation and sequencing, and to remove unwanted batch effects, we first used the lmer function of the lme4 package (v.1.1-27.1)[75] to fit a linear model of the following form in each stimulation condition and for each lineage/cell type:

$$\log(1 + \text{CPM}_i) = \alpha + \text{IID}_i + \text{LIB}_i + \text{RUN}_i + \text{FLOW}_i + \varepsilon_i \qquad (1)$$

where CPM$_i$ is the gene expression in sample $i$ (that is, one replicate of a given individual and set of experimental conditions); $\alpha$ is the intercept; $\text{IID}_i \sim \mathcal{N}(0, \sigma^2_{\text{IND}})$ captures the effect of the corresponding individual on gene expression; $\text{LIB}_i \sim \mathcal{N}(0, \sigma^2_{\text{LIB}})$ captures the effect of 10x Genomics library preparation; $\text{RUN}_i \sim \mathcal{N}(0, \sigma^2_{\text{RUN}})$ captures the effect of the experimental run; $\text{FLOW}_i \sim \mathcal{N}(0, \sigma^2_{\text{Flowcell}})$ captures the effect of the sequencing flow cell; and $\varepsilon_i$ captures residual variation between samples. We then subtracted the estimated value of the library, experimental

run and flow cell effects (as provided by the ranef function) from the transformed CPMs of each sample, to obtain batch-corrected CPM values. Finally, we averaged the batch-corrected CPM values obtained across different replicates for the same individual and set of stimulation conditions, to obtain final estimates of gene expression.

For each cell type and stimulation condition, an inverse-normal rank-transformation was applied to the log$_2$[CPM] of each gene, before testing for differences in gene expression between populations and mapping eQTL. Within each lineage and set of stimulation conditions, we ranked, for each gene, the pseudobulk expression values of all individuals, assigning ranks at random for ties, and replaced each observation with the corresponding quantile from a normal distribution with the same mean and s.d. as the original expression data. This inverse-normal rank-transformation rendered downstream analyses robust to zero-inflation in the data and outlier values, while maintaining the rank-transformed values on the same scale as the original data.

## Variance explained by lineage identity and viral exposure

We used CAR scores[76] to quantify the fraction of gene expression variance that is explained by variation across immune lineages and stimulation conditions. First, we built per-gene linear models regressing pseudobulk expression levels on two sets of dummy variables, encoding both lineage identity and stimulation condition. Specifically, we used a model of the form:

$$\text{Expr}_{ils} = \alpha + \sum_{l=2}^{5} \beta_l I_{\{\text{lineage}=l\}} + \sum_{l=2}^{5} \sum_{s=2,3} \gamma_{ls} I_{\{\text{lineage}=l \text{ and stim}=s\}} + \varepsilon_{ils} \qquad (2)$$

Where Expr$_{ils}$ is the log-transformed expression of the target gene for donor $i$, in lineage $l$ and in the condition of stimulation $s$; $\alpha$ is the intercept measuring the mean expression of the reference lineage (CD4$^+$ T cells) in the non-stimulated state; $\beta_l$ are parameters that capture the mean difference (log-fold change) in expression between lineage $l$ and the reference lineage; $I$ is an indicator variable equal to 1 when the subscript condition is met, and 0 otherwise; $\gamma_{ls}$ are parameters that capture the mean log-fold change in expression of lineage $l$ in response to stimulus $s$; and $\varepsilon_{ils}$ are normally distributed residuals. We then ran the carscore function from care R package (v.1.1.11)[76] on each model, setting $\lambda = 0$ (that is, no shrinkage), to obtain the CAR score associated with each parameter. In brief, care decorrelates a set of predictors using a Mahalanobis whitening transformation and computes CAR scores as marginal correlations between these decorrelated predictors and the response variable. This enables direct estimation of the contribution of each predictor to the variance of the response variable as the square of its CAR score. The variance explained by cellular identity (lineage) and stimulation is then computed as:

$$\text{Var}_{\text{lineage}} = \sum_{l=2}^{5} \text{CAR}(\beta_l)^2 \text{ and } \text{Var}_{\text{stim}} = \sum_{l=2}^{5} \sum_{s=1,2} \text{CAR}(\gamma_{ls})^2 \qquad (3)$$

## ISG activity calculation

ISGs strongly respond to both viruses across all lineages/cell types. We therefore evaluated each donor's ISG expression level in the basal state or after stimulation with either SARS-CoV-2 or IAV by constructing an ISG activity score. For the human genes in our filtered gene set ($n = 12,655$), we defined as ISGs ($n = 174$) those genes included in the union of GSEA's hallmark (https://www.gsea-msigdb.org/gsea/msigdb/genesets.jsp?collection=H) IFNα response and IFNγ response gene sets, but excluded those from the inflammatory response set. We then used AddModuleScore from Seurat (v.4.1.1)[77] to measure ISG activity as the mean pseudobulk expression level of ISGs in each sample minus the mean expression for one hundred randomly selected non-ISGs matched for mean magnitude of expression. In all analyses, ISG activity

scores were adjusted for cell mortality of the sample by fitting a model of the form:

$$\text{ISG}_i = \alpha + \beta_p\,\text{Population}_i + \beta_m\,\text{CellMortality}_i + \varepsilon_i \qquad (4)$$

and subtracting the effect of cell mortality from the raw ISG scores. In this model, $\text{ISG}_i$ denotes the ISG activity score of individual $i$; $\alpha$ is the intercept, $\text{Population}_i$ and $\text{CellMortality}_i$ are variables reflecting the donor's population and the cell mortality of the sample; $\beta_p$ and $\beta_m$ are parameters capturing the effect of the population and cell mortality on ISG activity; and $\varepsilon_i$ are normally distributed residuals. The difference in variance of ISG activity between SARS-CoV-2 and IAV was assessed using Levene's test. For comparisons with SIMOA-estimated IFN levels, the carscore function from the care R package[76] was used to model ISG activity as a function of levels of IFNα, IFNβ and IFNγ, adjusting for population, age, sex and cell mortality. The percentage of ISG variance attributable to each IFN (α, β, or γ) was estimated as the square of the resulting CAR scores.

## Testing for differences in lineage/cell type abundance between populations

We compared immune cell abundance between donors of African and European ancestries by contrasting the average number and percentage of cells assigned to each lineage/cell type between donors from both populations. To assess the statistical significance of population differences in cell type frequency, we first corrected cellular frequencies for the confounding effects of age, cell mortality and total number of cells in each sample (that is, donor × condition) using a linear model of the form

$$\begin{aligned}\text{CellularFrequency}_{cis} = {} & \alpha + \beta_p\,\text{Population}_i + \beta_a\,\text{Age}_i \\ & + \beta_m\,\text{CellMortality}_i + \beta_c\,\text{NCells}_{is} + \varepsilon_i\end{aligned} \qquad (5)$$

and subtracting the effect of these three covariates from the raw cell frequencies. In this model, $\text{CellularFrequency}_{cis}$ denotes the frequency/number of the lineage/cell type $c$ under consideration in individual $i$ and condition $s$; $\alpha$ is the intercept; $\text{Population}_i$, $\text{Age}_i$, $\text{CellMortality}_i$ and $\text{NCells}_{is}$ are variables reflecting the donor's age and population, the cell mortality of the sample and the total number of cells recovered in condition $s$; $\beta_p$, $\beta_a$, $\beta_m$ and $\beta_c$ are parameters capturing the effect of these covariates on cellular composition; and $\varepsilon_i$ are normally distributed residuals. The adjusted cell frequencies were then compared between populations using Wilcoxon's rank-sum tests.

## Mapping the genetic determinants of immune cell composition

We performed genome-wide association studies of the proportions of each immune lineage/cell type in the different stimulation conditions. In brief, we used PLINK (v.1.9)[78] to estimate at each locus the additive effect of each copy of the alternate allele on two quantitative traits: (1) the number of cells of each lineage relative to total number of cells in the sample and (2) the number of cells of each cell type relative to the lineage under consideration. In total, we performed 79 GWASs: one for each of the 27 immune classes (5 lineages and 22 cell types), in each of the 3 experimental conditions (except for the IAV-infected CD14+ monocytes, which are only present in the IAV condition). In each GWAS, we modelled cell type frequencies across individuals as

$$\begin{aligned}\text{CellularFrequency}_{cis} = {} & \alpha + \beta\,\text{SNP}_i + \beta_p\,\text{Population}_i + \beta_s\,\text{Sex}_i + \beta_a\,\text{Age}_i \\ & + \beta_m\,\text{CellMortality}_i + \beta_c\,\text{NCells}_{is} + \varepsilon_i\end{aligned} \qquad (6)$$

where the $\text{CellularFrequency}_{cis}$ is the rank-transformed percentage of lineage/cell type $c$ in the sample (that is, among cells from donor $i$ in condition $s$); $\text{SNP}i$ is the number of alternative alleles of donor $i$ for the target SNP; $\text{Population}_i$, $\text{Sex}_i$ and $\text{Age}_i$ are variables reflecting the donor's characteristics (population of origin, genetic sex and age);

$\text{CellMortality}_i$ and $\text{NCells}_{is}$ are variables reflecting technical parameters (respectively, the percentage of dead cells after thawing the cryopreserved PBMCs and the count of high-quality cells in the sample); $\beta$, $\beta_p$, $\beta_s$, $\beta_a$, $\beta_m$ and $\beta_c$ are parameters capturing the effect of these variables on cellular composition; and $\varepsilon_i$ are normally distributed residuals. For each SNP, we used Bonferroni correction to adjust for the number cell types and the condition tested and considered $P_{\text{adj}} < 5 \times 10^{-8}$ as genome-wide significant. Winner's curse-adjusted $Z$-score and $R^2$ were computed using FDR inverse quantile transformation[79].

## Effect of CMV infection on cell composition

We determined the CMV serostatus of AFB ($n = 78$), EUB ($n = 80$) and ASH ($n = 49$) donors with a human anti-IgG CMV ELISA kit (Abcam) on plasma samples, according to the manufacturer's instructions. We assessed the contribution of CMV infection to differences in cellular composition between Africans and Europeans using mediation analysis. Specifically, we used the mediate function of the mediation package of R (v.4.5.0)[80] to model the frequency of each cell type, as a function of population, CMV serostatus and covariates:

$$\text{CellularFrequency}_i = \alpha + \beta\,\text{CMV}_i + \delta\,I_i^{\text{EUB}} + Z_i^T \cdot \boldsymbol{\gamma} + \varepsilon_i \qquad (7)$$

$$\text{logit}(\text{Prob}(\text{CMV}_i = 1)) = \alpha' + \delta' I_i^{\text{EUB}} + Z_i^T \cdot \boldsymbol{\gamma}' \qquad (8)$$

where $\text{CellularFrequency}_i$ corresponds to the basal state frequency of the cell type under consideration; $\alpha$ and $\alpha'$ are two intercepts; $\beta$ is the effect of the CMV serostatus ($\text{CMV}_i$) on cellular proportions; $\delta$ and $\delta'$ are the (direct) effect of population (captured through the indicator variable $I_i^{\text{EUB}}$) on cell type frequency and CMV serostatus; $\boldsymbol{\gamma}$ and $\boldsymbol{\gamma}'$ capture the confounding effect of covariates (that is, age and cell mortality) on both gene expression and CMV serostatus; and $\varepsilon_i$ are normally distributed residuals. Under this model, we implicitly assumed that the effect of CMV serostatus is the same across populations. Although this assumption cannot be tested due to the lack of CMV⁻ individuals in the African group, we used an interaction test to evaluate whether the effect of CMV serostatus on cell composition is similar between European and East Asian donors (Supplementary Note 4 and Supplementary Fig. 7). To do so, we defined the following model, with the same notations as before

$$\text{CellularFrequency}_i = \alpha + \beta\,\text{CMV}_i + \delta\,I_i^{\text{ASH}} + \theta\,\text{CMV}_i I_i^{\text{ASH}} + Z_i^T \cdot \boldsymbol{\gamma} + \varepsilon_i \quad (9)$$

and performed a Student's $t$-test for the null hypothesis that the effect of CMV is the same in Europeans and East Asians ($\mathcal{H}_0 : \theta = 0$).

## Modelling population effects on the variation of gene expression

To estimate population effects on gene expression while mitigating any potential batch effect relating to sample processing, we first focused exclusively on AFB and EUB individuals, as all these individuals were recruited during the same sampling campaign and their PBMCs were processed at the same time, with the same experimental procedure[18]. For each immune lineage, cell type, stimulation condition and gene, we then built a separate linear model of the form:

$$\text{Expr}_i = \alpha + \beta_r I_i^{\text{EUB}} + Z_i^T \cdot \boldsymbol{\gamma} + \varepsilon_i \qquad (10)$$

where $\text{Expr}_i$ is the rank-transformed gene expression (log-normalized CPM) for individual $i$ in the lineage/cell type and condition under consideration; $I_i^{\text{EUB}}$ is an indicator variable equal to 1 for European-ancestries individuals and 0 otherwise; and $Z_i$ represents the set of core covariates of the sample that includes the individual's age and cellular mortality (that is, the proportion of dying cells in each thawed vial, as a proxy of sample quality). Moreover, $\varepsilon_i$ are the normally distributed residuals and $\alpha, \beta_r, \gamma$ are the fitted parameters of the models. In particular, $\alpha$ is the intercept, $\beta_r$ indicates the log₂-transformed fold change difference in

expression between individuals of European and African ancestries, and $\gamma$ captures the effects of the set of core covariates on gene expression.

We reasoned that differences in the variance of gene expression between populations might inflate the number of false positives. We therefore used the vcovHC function of sandwich (v.2.5-1)[81] with the Type='HC3' option to compute sandwich estimators of variance that are robust to residual heteroskedasticity. We estimated the $\beta_r$ coefficients and their standard error with the coeftest function of lmtest (v.0.9-40)[82]. The FDR was calculated across all conditions and lineages using the Benjamini–Hochberg procedure (p.adjust function with the 'fdr' method). Genes with an FDR < 1% and $|\beta_r| > 0.2$ were considered to be differentially expressed between populations (that is, 'raw' pop-DEGs). We adjusted for cellular composition within each lineage $L$ by introducing into model (10) a set of variables $(F_j)_{j \in L}$ encoding the frequency in the PBMC fraction of each cell type $j$ comprising the lineage (for example, naive, effector and regulatory subsets of CD4+ T cells).

$$\text{Expr}_i = \alpha' + \beta_a I_i^{\text{EUB}} + Z_i^T \cdot \gamma' + \sum_{j \in L} \delta_j F_{j,i} + \varepsilon_i \qquad (11)$$

The notation is as above, with $\alpha, \beta_a, \gamma'$ the fitted parameters of the model. In this model, $\delta_j$ is the effect on gene expression of a 1% increase in cell type $j$ and $\beta_a$ indicates the cell composition-adjusted $\log_2$-transformed fold change in the difference in expression between AFB and EUB individuals. The significance of $\beta_a$ was calculated as described above, with a sandwich estimator of variance and the coeftest function. FDR was calculated across all conditions and lineages to yield a set of "cell-composition-adjusted" popDEGs. We assessed the impact of cellular composition on differences in gene expression between populations, by defining Student's $t$-test statistic $T_{\Delta\beta}$ as follows:

$$T_{\Delta\beta} = \frac{\hat{\beta}_a - \hat{\beta}_r}{\text{Var}(\hat{\beta}_a - \hat{\beta}_r)} = \frac{\hat{\beta}_a - \hat{\beta}_r}{\hat{s}_a^2 + \hat{s}_r^2 - 2\rho\hat{s}_a\hat{s}_r} \qquad (12)$$

where $\hat{\beta}_r$ and $\hat{\beta}_a$ are the raw and cell-composition-adjusted differences in expression between populations; $\hat{s}_r$ and $\hat{s}_a$ are the estimated standard error of $\hat{\beta}_r$ and $\hat{\beta}_a$, respectively; and $\rho$ is the observed correlation in permuted data between the $\hat{\beta}_r$ and $\hat{\beta}_a$ statistics. Under the null hypothesis that population differences are not affected by cellular composition, $T_{\Delta\beta}$ should follow an approximate Gaussian distribution with mean 0 and variance 1, enabling the definition of a $P$ value $P_{\Delta\beta}$. We then considered the set of raw popDEGs that (1) were not significant after adjustment (FDR > 1% or $|\beta_a| < 0.2$) and (2) displayed significant differences between the raw and adjusted effect sizes ($|T_{\Delta\beta}| > 1.96$) imputable to the effect of cellular composition.

For the assessment of population differences in response to viral stimuli (that is, popDRGs), we used the same approach, but with the replacement of log-normalized counts with the log-fold change difference in expression between the stimulation conditions for each of the two viruses and non-stimulated conditions.

## Pathway enrichment analyses

We performed functional assessments of the effects of cellular composition variability on differences in gene expression between donors in the basal state and in response to each virus, using the fgsea R package (v.1.18.1)[83] and the default options. This made it possible to perform a gene set enrichment analysis with population differences in each lineage ranked by the magnitude of the effect of ancestry on the expression or response of the gene before ($\beta_r$) and after ($\beta_a$) adjustment for differences in cellular composition.

## Fine mapping of eQTL

For eQTL mapping, we used variants with MAF > 5% in at least one of the three populations considered, resulting in a set of 10,711,657 SNPs, of which 4,164,060 were located <100 kb from a gene. We used MatrixEQTL (v.2.3)[84] to map eQTLs in a 100 kb region around each

gene and obtain estimates of eQTL effect sizes and their standard error. eQTL mapping was performed separately for each immune lineage/cell type and condition, based on rank-transformed gene expression values. eQTL analyses were performed adjusting for population, age, chromosomal sex, cell composition (within each lineage), as well as cell mortality and total number of cells in the sample, and a data-driven number of surrogate variables included to capture unknown confounders and remove unwanted variability. Specifically, for each immune lineage/cell type and condition, surrogate variables were obtained using the sva function from the sva R package (v.3.40.0)[85] with option method='two-steps', providing all other covariates as known confounders (mod argument). The number of surrogate variables to use in each lineage/cell type and condition was determined automatically based on the results from num.sv function with method='be'[85].

For each gene, immune lineage/cell type and stimulation condition, $Z$-values (that is, the effect size of each eQTL divided by the standard error of effect size) were then calculated, and the fine mapping of eQTLs was performed using SuSiE (v.0.11.42)[86] (susie_rss function of the susieR R package), with a default value of up to 10 independent eQTLs per gene. Imputed genotype dosages were extracted in a 100 kb window around each gene and regressed against the population of origin (that is, AFB, EUB or ASH). Genes with fewer than 50 SNPs in the selected window were discarded from the analysis. Pairwise correlations between the population-adjusted dosages were then assessed, to define the genotype correlation matrix to be used for the fine mapping of eQTLs. In rare cases (<0.1% of tested gene × condition combinations), the susie_rss function did not converge, even when the number of iterations was increased to >10[6]. These runs were therefore discarded, and the associated eQTLs were assigned a null $Z$-score during FDR computation (see below). For each eQTL, the index SNP was defined as the SNP with the highest posterior inclusion probability (that is, the $\alpha$ parameter in the output of SuSiE) for that eQTL, and the 95% credible interval was obtained as the minimal set of SNPs $S$ such that $\alpha_s > 0.01$ for all $s \in S$ and $\sum_{s \in S} \alpha_s > 0.95$. Only eQTLs with a log-Bayes factor (lbf) > 3 were considered for further analyses.

For each lineage and set of stimulation conditions, each eQTL identified by SuSiE was assigned an eQTL evidence score, defined as the absolute $Z$-value of association between the eQTL index SNP and the associated gene. We then used a pooled permutation strategy to define the genome-wide number of significant eQTLs (that is, eQTL × gene combinations) expected under the null hypothesis, for different thresholds of the eQTL evidence score. We repeated the eQTL mapping procedure on the dataset after randomly permuting genotype labels within each population. We then counted, for each possible evidence score threshold $T$, the number of eQTLs identified in the observed and permuted data. Finally, we retained as a significance threshold the lowest threshold giving several significant eQTLs in the permuted data (false positives) of less than 1% the number of eQTLs identified in the observed data (false positives + true positives).

## Aggregation of eQTLs across cell types and stimulation conditions

The eQTL index SNP may differ between cellular states (immune lineage/cell type and stimulation condition), even in the presence of a single causal variant. It is therefore necessary to aggregate eQTLs to ensure that the same locus is tagged by a single variant across cellular states. To this end, we applied the following procedure, for each gene: (1) let $C_g$ be the set of cellular states where a significant eQTL was detected for gene $g$, and $S_g$ be the associated list of eQTLs (that is, cellular state × index SNP). We aim to define a minimal set of SNPs, $M_g$, that overlaps the 95% credible intervals of all significant eQTLs in $S_g$. (2) For each SNP $s$ in a 100-kb window around each gene, compute the expected number of cellular states in which the SNP has a causal effect on gene expression $E[N_c(s)]$ as:

$$E[N_c(s)] = \sum_{j \in C_g} PP_{sj} \tag{13}$$

where $PP_{sj}$ is the posterior probability that SNP $s$ has a causal effect on the expression of gene $g$ in the cellular state $j$ (cell type × condition). (3) Find the SNP $s$ that maximizes $E[N_c(s)]$, and add it to $M_g$. (4) Remove from $S_g$ all eQTLs for which the 95% credible interval contains SNP $s$. (5) Repeat steps 1–3 until $S_g$ is empty.

At the end of this procedure, $M_g$ provides the list of independent eQTL index SNPs (referred to as eSNPs) for gene $g$, for which we extracted summary statistics across all cellular states.

## Mapping of response eQTLs

For the mapping of response eQTLs (reQTLs), we repeated the same procedure as for the mapping of eQTLs, using the rank-transformed $\log_2$-fold change as input rather than gene expression. This included reQTL mapping using MatrixEQTL[84], fine mapping with SuSiE[86], permutation-based FDR computation, and aggregation of reQTL across immune lineages, cell types and stimulation conditions. Surrogate variables were computed directly from $\log_2$-transformed fold changes. For IAV-infected monocytes (detected only in the IAV condition), fold changes were computed relative to CD14$^+$ monocytes in the non-stimulated condition. This produced a list of independent reQTL index SNPs $M'$, like that obtained for eQTLs, for which we extract summary statistics across all cellular states.

## Sharing of eQTLs across cell types and stimulation conditions

After extracting the set $M$ of independent eSNPs across all genes, we defined cell-type-specific eQTLs as eQTLs significant genome-wide in a single cell type. We accounted for the possibility that some eQTLs may be missed in specific cell types due to a lack of power by introducing a second definition of eQTL sharing based on nominal $P$ values. Specifically, we considered an eQTL to be cell type-specific at a nominal significance if, and only if, it was significant genome-wide in a single cell type and its nominal $P$ value of association was greater than 0.01 in all other cell types. For each pair of cell types, the correlation of eQTL effect sizes was calculated on the set of all eQTLs passing the nominal significance criterion (Student's $t$-test, $P < 0.01$) in at least one of the two cell types. To understand how the effect of genetics on immune response varies between SARS-CoV-2 and IAV, we defined an interaction statistic that enabled us to test for differences in reQTL effect size between the two viruses. Specifically, within each immune lineage/cell type, we defined:

$$T_{int} = \frac{\widehat{\beta}_{IAV} - \widehat{\beta}_{COV}}{Var(\widehat{\beta}_{IAV} - \widehat{\beta}_{COV})} = \frac{\widehat{\beta}_{IAV} - \widehat{\beta}_{COV}}{\hat{s}^2_{IAV} + \hat{s}^2_{COV}} \tag{14}$$

When the reQTL effect size is identical between the two viruses, we expect $T_{int}$ to be normally distributed around 0 with variance 1, allowing to derive an interaction $P$ value. We therefore defined as virus-dependent reQTLs those with a nominal interaction $P < 0.01$ and as virus-specific reQTLs those that passed a nominal $P$ value threshold of 0.01 in only one of the two stimulation conditions.

## Comparison of eQTLs and eGenes across studies

To assess the replicability of the eQTLs detected in our study, we compared the eGenes that we identified with those reported in three single-cell studies of resting and stimulated PBMCs[19,87,88]. For each study, we first reassigned each reported cell type to one of the five major lineages that we identified. We then retrieved, for each lineage/condition, the union of all genes with a reported eQTL in at least one of the cell types associated to that lineage, using the following thresholds: ref. 87, $P < 10^{-5}$; ref. 88, FDR < 0.05; ref. 19, lfsr < 0.1. The resulting gene sets were considered as eGenes for each lineage/condition.

Enrichments of eQTLs in previously identified eGenes were tested for each study and lineage separately, using a Fisher's exact test to assess whether genes reported to contain an eQTL in a given lineage were more likely to present an eQTL in the same lineage in our study. For each comparison, we used as background sets all genes tested for eQTLs in both studies. When the set of tested genes was not reported in the study, as reported previously[87], we used the union of eQTL genes across all cell types, as a proxy for the set of tested genes. For reQTLs, we compared, for each lineage, genes with a reQTL after IAV stimulation in our study with genes with an eQTL at lfsr < 0.1 specifically after IAV stimulation (but not in non-stimulated cells) in ref. 19. Finally, we compared the direction of effect at shared eQTLs between our study (FDR < 0.01) and that of ref. 19 (lfsr < 0.1), focusing on the eQTL index SNP reported by the latter and assessing the percentage of eQTLs with concordant direction of effect in our data.

To assess the extent to which our findings in PBMCs replicate in the lung, we downloaded Genotype-Tissue Expression (GTEx) lung eQTL data[55] from the eQTL catalogue (uniformly processed summary statistics; http://ftp.ebi.ac.uk/pub/databases/spot/eQTL/sumstats/GTEx/ge/GTEx_ge_lung.all.tsv.gz). We next considered the index SNP from each eQTL, focusing on the subset of genes with median TPM > 10 in the lung and eQTLs with MAF > 5% in the GTEx dataset. We considered any eQTL with (1) $P < 0.01$ in the lung and (2) the same effect direction between lung and the lineage/cell type/condition in which it is the most significant in our study as replicated. As a comparison, we evaluated the amount of eQTLs that would be replicated when selecting SNPs at random, matching for MAF in GTEx (bins of 5%) and the distance between the eQTL index SNPs and the nearest gene (that is, bins of 0–1, 1–5, 5–10, 10–20, 20–50 and 50–100 kb), and computed the fold-enrichment in replicated eQTLs as the ratio between the observed and expected number of replicated eQTLs.

## Mediation analyses

For all popDEGs and popDRGs, we evaluated the proportion of the difference in expression or response to viral stimulation between populations attributable to either genetic factors (that is, eQTLs) or cellular composition, using the mediate function of the mediation R package (v.4.5.0)[80]. Mediation analysis made it possible to separate the differences in expression/response between populations that were mediated by genetics (that is, differences in allele frequency of a given eQTL between populations, $\zeta_g$), or cellular composition (that is, difference in cell type proportions between populations, $\zeta_c$) from those occurring independently of the eQTL/cell type considered (independent or direct effect $\delta$). It was then possible to estimate the respective proportion of population differences mediated by genetics $\tau_g$ and cellular composition $\tau_c$ as $\tau_c = \frac{\zeta_c}{\zeta_c + \zeta_g + \delta}$ and $\tau_g = \frac{\zeta_g}{\zeta_g + \zeta_c + \delta}$, with $\zeta_c + \zeta_g + \delta$ corresponding to the total differences in expression/response between populations. For each popDEG and popDRG, we focused on either (1) the most strongly associated SNP in a 100 kb window around the gene, regardless of the presence or absence of a significant (r)eQTL, or (2) the cell type differing most strongly between populations in each lineage (that is, CD16$^+$ monocytes in the myeloid lineage, κ-light-chain-expressing memory cells in the B cell lineage, effector cells in CD4$^+$ T cell lineage, CD8$^+$ EMRA cells in the CD8$^+$ T cell lineage and memory cells in the NK cell lineage). For each popDEG and potential mediator $M$ (that is, eQTL SNP or cell subtype proportion), we then ran mediate with the following models:

$$Expr_i = \alpha + \beta M_i + \delta I_i^{EUB} + Z_i^T \cdot \gamma + \varepsilon_i \tag{15}$$

$$M_i = \alpha' + \delta' I_i^{EUB} + Z_i^T \cdot \gamma' + \varepsilon'_i \tag{16}$$

where $Expr_i$ corresponds to normalized expression values in the cell type/condition under consideration; $\alpha$ and $\alpha'$ are two intercepts; $\beta$ is

the effect of the mediator $M_i$ on gene expression; $\delta$ and $\delta'$ are the (direct) effect of population (captured through the indicator variable $I_i^{EUB}$) on gene expression and on the mediator; $\gamma$ and $\gamma'$ capture the confounding effect of covariates (that is, age and cell mortality) on both gene expression and the mediator; and $\varepsilon_i$ and $\varepsilon_i'$ are normally distributed residuals. For popDRGs, we used the same approach, replacing normalized gene expression values with the log$_2$-transformed fold change in gene expression between the stimulated and unstimulated states.

### Detection of signals of natural selection

We avoided SNP ascertainment bias by performing natural selection analyses with high-coverage sequencing data from the 1KG Project[89]. We downloaded the GRCh38 phased genotype files from the New York Genome Center FTP server and calculated the pairwise $F_{ST}$ (ref. 90) between our three study populations (AFB, EUB or ASH) and all 1KG populations to identify the 1KG populations that were the most genetically similar to our study populations. All selection and introgression analyses (see the 'Archaic introgression analyses' section) were based on the Yoruba from Ibadan, Nigeria (YRI), Utah residents with Northern and Western European ancestries (CEU) and Southern Han Chinese (CHS) populations, as these 1KG populations had the lowest $F_{ST}$ values with our three study groups. We filtered the data to include only autosomal biallelic SNPs and insertions/deletions (indels), and removed sites that were invariant (that is, monomorphic) across the three populations. We identified loci presenting signals of positive selection (local adaptation) with the PBS[39], based on the Reynold's $F_{ST}$ estimator[90] between pairs of populations. PBS values were calculated for the YRI, CEU, and CHS populations separately, with the other two populations used as the control and outgroup. For each population, genome-wide PBS values were then ranked, and variants with PBS values within the top 1% were considered to be putative targets of selection. For annotation of the selected eQTLs, the ancestral and derived states at each site were inferred from six-way EPO multiple alignments for six primate species (available from http://ftp.ensembl.org/pub/release-71/emf/ensembl-compara/epo_6_primate/), and the effect size was reported for the derived allele. For sites without an ancestral/derived state in the EPO alignment, the effect of the allele with the lowest frequency worldwide was reported.

We assessed the extent to which different sets of eQTLs displayed signals of local adaptation in permutation-based enrichment analyses. For each population, we compared the mean PBS values at (r)eQTLs for each set of cell type/stimulation condition with the mean PBS values obtained for 10,000 sets of randomly resampled sites. Resampled sites were matched with eQTLs for MAF (mean MAF across the three populations, bins of 0.01), LD scores (quintiles) and distance to the nearest gene (bins of 0–1, 1–5, 10–20, 20–50 and >100 kb). For each population and set of eQTLs, we defined the fold-enrichment (FE) in positive selection as the ratio of observed/expected values for mean PBS and extracted the mean and 95% confidence interval of this ratio across all resamplings. One-sided resampling $P$ values were calculated as the number of resamplings with a FE > 1 divided by the total number of resamplings. Resampling $P$ values were then adjusted for multiple testing by the Benjamini–Hochberg method.

### Detecting and dating episodes of local adaptation

We inferred the frequency trajectories of all eQTLs and reQTLs during the past 2,000 generations (that is, 56,000 years before the present, with a generation time of 28 years), systematically by using CLUES (commit no. 7371b86, 27 May 2021)[40]. We first used Relate (v.1.1.8)[91] on each population separately to reconstruct tree-like ancestral recombination graphs (ARGs) around each SNP in the genome and to estimate effective population sizes across time on the basis of the rate of coalescence events over the inferred ARGs. Using CLUES[40], we then estimated at each eQTL or reQTL, the most likely allele frequency

trajectories for each sampled ARG and averaged these trajectories across all possible ARGs.

We then analysed changes in inferred allele frequencies over time to identify selection events characterized by a rapid change in allele frequency (Supplementary Fig. 9a). We considered the posterior mean of allele frequency at each generation and smoothed the inferred allele frequency trajectories by loess regression (with span = 0.1) to ensure progressive changes in allele frequency over time and to minimize the artifacts induced by the inference process. Finally, for each variant and in each population, we calculated the change in allele frequency $f$ at each generation as the difference in the smoothed allele frequency between two consecutive generations:

$$\dot{f}(t) = \frac{df}{dt}(t) = f(t+1) - f(t) \tag{17}$$

Under an assumption of neutrality, the count of a particular allele at generation $t+1$ is the result of a Bernoulli trial parameterized $\mathcal{B}(N,f)$, where $N$ is the size of the haploid population. The variance of allele frequencies at generation $t+1$ is therefore greater for alleles present at higher frequencies in generation $t$,

$$V[f] = \frac{f(1-f)}{N}. \tag{18}$$

We adjusted for this by scaling the change in allele frequency $\dot{f}$ by a normalizing factor dependent on the allele frequency at generation $t$, such that the variance of the normalized change in allele frequency $\dot{g}$ was the same across all variants,

$$\dot{g} = \frac{\dot{f}}{\sqrt{f(1-f)}} \tag{19}$$

Finally, at each generation, we divided the normalized change in allele frequency $\dot{g}$ by its s.d. across all eQTLs and reQTLs, to calculate a $Z$-score for detecting alleles for which the normalized change in allele frequency exceeded genome-wide expectations,

$$Z = \frac{\dot{g}}{\text{s.d.}(\dot{g})} \tag{20}$$

For each variant and generation, we then considered an absolute $Z$-score > 3 to constitute evidence of selection and we inferred the onset of selection of a variant as the first generation in which $|Z| > 3$.

### Simulations, power and type I error estimates

We assessed the ability of our approach to detect (and date) events of natural selection correctly from the trajectories of allele frequencies by using simulations with SLiM (v.4.0.1)[92] under various selection scenarios. Simulations were performed under a Wright–Fisher model for a single mutation occurring around 5,000 generations ago, at a frequency varying from $f_{min} = \frac{1}{N}$ to $f_{max} = 1 - \frac{1}{N}$ in steps of 1%, where $N$ is the simulated population size. We allowed population size to vary over time according to published estimates[91] for the YRI, CEU and CHS populations (Supplementary Fig. 9b). We then performed simulations both under an assumption of neutrality (1,000 simulations for each starting frequency) and assuming a 200-generation-long episode of selection (100 simulations for each starting frequency and selection scenario). Selection episodes were simulated with an onset of selection 1,000, 2,000, 3,000 or 4,000 generations ago, and with a selection coefficient ranging from 0.01 to 0.05 (Supplementary Fig. 9c). We saved computation time by performing a tenfold scaling in line with SLiM recommendations. For each selected scenario and variant, simulated allele frequencies were retrieved every ten generations, and smoothed using loess regression with a span of 0.1. We then calculated

normalized differences in smoothed allele frequencies for each simulated variant and scaled these differences at each generation, on the basis of their s.d. among neutral variants, to obtain $Z$-scores. For each selection scenario, we focused on the centre of the selection interval and determined the type I error and power for various thresholds of absolute $Z$-scores varying from 0 to 6. We found that a threshold of 3 yielded both a low type I error (<0.2% false positives) and a satisfactory power for detecting selection events (Supplementary Fig. 9c). Finally, at each generation, we estimated the percentage of simulations, under an assumption of neutrality or a particular selection scenario, for which the absolute $Z$-score exceeded a threshold of 3. We found that significant $Z$-scores were equally rare at each generation under the assumption of neutrality, but that selected variants presented a clear and localized enrichment in significant $Z$-scores for intervals in which we simulated selection (Supplementary Fig. 9d).

### Archaic introgression analyses

For the definition of regions of the modern human genome of archaic ancestry (Neanderthal or Denisovan), we downloaded the VCFs from the high-coverage Neanderthal Vindija[93] and Denisovan Altai[94] genomes (human genome assembly GRCh37; http://cdna.eva.mpg.de/neandertal/Vindija/) and applied the corresponding genome masks (FilterBed files). We then removed sites within segmental duplications and lifted over the genomic coordinates to the GRCh38 assembly with CrossMap (v.0.6.3)[95]. We used two statistics to identify introgressed regions in the CEU and CHS populations: (1) conditional random fields (CRF)[96,97], which uses reference archaic and outgroup genomes to identify introgressed haplotypes; and (2) the $S'$ method[98], which identifies stretches of probably introgressed alleles without requiring the definition of an archaic reference genome.

For CRF-based calling, we phased the data with SHAPEIT4 (v.4.2.1)[60], using the recommended parameters for sequence data, and focused on biallelic SNPs for which the ancestral/derived state was unambiguously defined. We then performed two independent runs of CRF to detect haplotypes inherited from Neanderthal or Denisova. For Neanderthal-introgressed haplotypes, we used the Vindija Neanderthal genome as the archaic reference and YRI individuals merged with the Altai Denisovan genome as the outgroup. For Denisovan-introgressed haplotypes, we used the Altai Denisovan genome as the archaic reference panel and YRI individuals merged with the Vindija Neanderthal genome as the outgroup. The results from the two independent CRF runs were analysed jointly, and we retained alleles with a marginal posterior probability $P_{\text{Neanderthal}} \geq 0.9$ and $P_{\text{Denisova}} < 0.5$ as Neanderthal-introgressed haplotypes and those containing alleles with $P_{\text{Denisova}} \geq 0.9$ and $P_{\text{Neanderthal}} < 0.5$ as Denisovan-introgressed haplotypes. For the $S'$-based calling of introgressed regions, we considered all biallelic SNPs with an allele frequency of <1% in the YRI population to be Eurasian-specific alleles. We then ran the Sprime (v.07Dec18.5e2; https://github.com/browning-lab/sprime) separately for the CEU and CHS populations to identify and score putatively introgressed regions containing a high density of Eurasian-specific alleles. Putatively introgressed regions with a $S'$ score of >150,000 were considered to be introgressed. This cut-off score has been shown to provide a good trade-off between power and accuracy based on simulations of introgression under realistic demographic scenarios[98]. For both calling methods (that is, CRF and $S'$), we used the recombination map from the 1KG Project Phase 3 data release[56].

After the calling of introgressed regions throughout the genome for each population, we defined SNPs of putative archaic origin (archaic SNPs, aSNPs) as those (1) located in an introgressed region defined by either the CRF or $S'$ method; (2) with one of their alleles being rare or absent (MAF < 1%) in the YRI population, but present in the Vindija Neanderthal or Denisovan Altai genomes; and (3) in high LD ($r^2 > 0.8$) with at least two other aSNPs and, to exclude incomplete lineage sorting, comprising an LD block of >10 kb. This yielded a set of 100,345 high-confidence aSNPs (Supplementary Table 8a). We further categorized aSNPs as of Neanderthal origin, Denisovan origin or shared origin according to their presence/absence in the Vindija Neanderthal and Denisovan Altai genomes. Finally, we considered any site that was in high LD with at least one aSNP in the same population in which introgression was detected to be introgressed, and classified introgressed haplotypes as of Neanderthal origin, Denisovan origin or shared origin according to the most frequent origin of aSNPs in the haplotype. For introgressed SNPs, we defined the introgressed allele as (1) the allele rare or absent from individuals of African ancestries if the SNP was an aSNP; and (2) for non-aSNPs, the allele most frequently segregating with the introgressed allele of linked aSNPs. In each population, introgressed alleles with a frequency in the top 1% for introgressed alleles genome-wide were considered to present evidence of adaptive introgression.

The enrichment of introgressed haplotypes in eQTLs or reQTLs was assessed separately for each population (CEU and CHS), first by stimulation condition and then by cell type within each condition. To avoid biases related to an increased power for the detection of eQTLs that segregate at higher frequencies in European genomes, (that is, $n_{\text{EUB}} = 80$ and $n_{\text{ASH}} = 62$), we considered a set of $n = 10,276$ eQTLs mapped on a downsampled dataset composed of the same number of individuals from each population (EUB and ASH) within each cell type and condition. This downsampled set of eQTLs was highly concordant with the original eQTL mapping (that is, >95% sharing at the lineage level). Within each cell type/stimulation condition, we considered the set of all (r)eQTLs for which the index SNP displayed at least a marginal association (Student's $t$-test, $P < 0.01$) with gene expression. For each population and (r)eQTL set, we then grouped (r)eQTLs in high LD ($r^2 > 0.8$), retaining a single representative per group, and counted the total number of (r)eQTLs for which the index SNP was in LD ($r^2 > 0.8$) with an aSNP (that is, introgressed eQTLs). We then used PLINK (v.1.9) --indep-pairwise (with a 500 kb window, 1 kb step, an $r^2$ threshold of 0.8, and a MAF > 5%)[57] to define tag-SNPs for each population, and we determined the expected number of introgressed SNPs by resampling tag-SNPs at random with the same distribution for MAF, LD scores and distance to the nearest gene. We performed 10,000 resamplings for each (r)eQTL set and population. One-sided resampling-based $P$ values were calculated as the frequency at which the number of introgressed SNPs among resampled SNPs exceeded the number of introgressed SNPs among (r)eQTLs. Resampling-based $P$ values were then adjusted for multiple testing using the Benjamini–Hochberg method.

We searched for signals of adaptive introgression by determining whether introgressed haplotypes that altered gene expression were introgressed at a higher frequency than introgressed haplotypes with no effect on gene expression. For each stimulation cell type/condition, we focused on the set of introgressed eQTLs segregating with a MAF > 5% in each population (retaining a single representative per LD group) and compared the frequency of the introgressed allele with that of introgressed tag-SNPs genome-wide. We modelled $r_{\text{(Freq)}}$, the (rank-transformed) frequency of introgressed tag-SNPs according to the presence/absence of a linked eQTL ($I_{\text{eQTL}}$), and the mean MAF of the SNP across the three populations (giving a higher power for eQTL detection).

$$r_{\text{(Freq)}} \approx \alpha + \beta \, I_{\text{eQTL}} + \gamma \, \overline{\text{MAF}} \tag{21}$$

where $I_{\text{eQTL}}$ is an indicator variable equal to 1 if the SNP is in LD with an eQTL ($r^2 > 0.8$) and 0 otherwise; $\overline{MAF}$ is the mean MAF calculated separately for each population; $\alpha$ is the intercept of the model; $\beta$ measures the difference in rank $r_{\text{(Freq)}}$ between eQTLs and non eQTLs; and $\gamma$ is a nuisance parameter capturing the relationship between $\overline{MAF}$ and $r_{\text{(Freq)}}$. Under this model, the difference in frequency between eQTLs and non-eQTLs can be tested directly in a Student's $t$-test with $\mathcal{H}_0 : \beta = 0$.

## Enrichment in COVID-19-associated loci and colocalization analyses

We downloaded summary statistics from the COVID-19 Host Genetics Initiative (release 7: https://www.covid19hg.org/results/r7)[8] for three GWASs: (1) A2—very severe respiratory cases of confirmed COVID-19 versus the general population; (2) B2—hospitalized COVID-19 cases versus the general population; (3) C2—confirmed COVID-19 cases versus the general population. We assessed the enrichment in eQTLs and reQTLs of COVID-19 susceptibility/severity loci by considering, for each eQTL/reQTL, the A2, B2 or C2 $P$ values of the index SNP and calculating the percentage of eQTLs/reQTLs with a significant GWAS $P$ value of $10^{-4}$. This percentage was then compared to that obtained for the resampled set of SNPs, matched for distance to the nearest gene (bins of 0–1, 1–5, 5–10, 10–20, 20–50 and 50–100 kb) and MAF (1% MAF bins). We performed 10,000 resamplings for each set of eQTLs/reQTLs tested. The use of different $P$-value thresholds for COVID-19-associated hits ($10^{-3}$ to $10^{-5}$) yielded similar results. Note that, despite the strong overlap (OR > 200, Fisher's test, $P = 4.2 \times 10^{-40}$) between loci associated with susceptibility (C2) and severity (A2 or B2)[8], 81 out of 105 COVID-19 associated eQTLs (at nominal $P < 10^{-4}$) are associated specifically with either susceptibility ($n = 19$) or severity ($n = 62$), supporting the relevance of considering these traits separately in our analysis.

To identify specific eQTLs/reQTLs colocalized with GWAS hits, we first considered all (r)eQTLs for which the index SNPs were located within 100 kb of a SNP associated with COVID-19 susceptibility/severity ($P < 10^{-5}$). For each immune lineage/cell type, and condition for which the eQTL/reQTL reached genome-wide significance, we next extracted all SNPs in a 500 kb window around the index SNP for which summary statistics were available for both the eQTLs/reQTLs and COVID-19 GWAS phenotypes (A2, B2 and C2) and performed a colocalization test using the coloc.signals function of the coloc (v.5.1.0) R package. We set a prior probability for colocalization $p_{12}$ of $10^{-5}$ (that is, the recommended default value). Any pair of (r)eQTL/COVID-19 phenotypes with a posterior probability for colocalization $PP_{H4} > 0.8$ was considered to display significant colocalization.

## Transcriptome-wide association tests

Using the summary statistics from the COVID-19 Host Genetics Initiative[8], we applied the S-PrediXcan framework (v.0.6.11)[99] to leverage our genotype-expression dataset and identify associations between genotypes and COVID-19 traits that could be mediated by the regulation of gene expression. These analyses were conducted separately in each of the 5 lineages or 22 cell types and the 3 experimental conditions of our setting.

To perform these transcriptome-wide association tests (across the 12,655 human genes of our final dataset), we proceeded in two steps. First, we used the pseudobulk gene expression levels detected in each cell type/lineage and condition, together with the associated genotypes, from each of the 222 donors to build reference transcriptome datasets. We then trained elastic net regression models on these references to estimate the effect on gene expression of each SNP within a 100 kb window around each gene. These models were of the form:

$$\text{Expr}_{ij} = \alpha + X_{ij}^T \cdot \boldsymbol{w}_j + Z_i^T \cdot \boldsymbol{\gamma} + \varepsilon_i \tag{22}$$

where $\text{Expr}_{ij}$ is the rank-transformed expression (log-normalized CPM) of gene $j$ for individual $i$ in the lineage/cell type and condition under consideration; $\alpha$ is an intercept; $X_{ij}$ are the genotypes of common variants in a 100 kb window around gene $j$; $Z_i$ represents the set of core covariates of the sample that includes the individual's age and population of origin, the cellular mortality of the sample and the frequency in the PBMC fraction of each cell type $k$ comprising the lineage. Moreover, $\boldsymbol{w}_j$ and $\boldsymbol{\gamma}$ are parameter vectors capturing the effect of genotypes and covariates, and the $\varepsilon_i$ are normally distributed residuals.

We followed the S-PrediXcan pipeline[99] using the regression coefficients $w_j$ as weights to predict the association between the genetically controlled expression (GReX) of each gene $j$ (given by $\text{GReX}_{ij} = \alpha + X_{ij}^T \cdot \boldsymbol{w}_j$) and the trait of interest. Specifically, we combined these weights with SNP covariances calculated from our data to approximate $Z$-scores of association with COVID-19-trait as

$$Z_j^{\text{TWA}} \approx \sum_l w_{lj} \frac{\widehat{\sigma}_l}{\widehat{\sigma}_j} \frac{\widehat{\beta}_l}{\text{s.e.}(\widehat{\beta}_l)} \tag{23}$$

where the $Z_j^{\text{TWA}}$ statistic measures the association between gene $j$'s GReX and each of the three COVID-19 traits; $w_{lj}$ is the weight of SNP $l$ in the prediction of gene $j$'s expression, $\widehat{\sigma}_l$ and $\widehat{\sigma}_j$ are, respectively, the estimated variances of the SNP and the predicted gene expression, and $\widehat{\beta}_l$ and s.e.($\widehat{\beta}_l$) are the effect size estimated by each GWAS for SNP $l$ and its standard error, respectively.

## Statistical analyses

Unless explicitly specified, all statistical tests are two-sided and based on measurements from independent samples.

## Inclusion and ethics

The current research project builds on samples collected in Ghent (Belgium) and Hong-Kong SAR (China) and has been conducted in collaboration with local researchers. Roles and responsibilities were agreed among collaborators ahead of the research. Research conducted in this study is relevant to local participants and has been reviewed by local ethics committees (committee of Ghent University, Belgium, B670201214647; Institutional Review Board of the University of Hong Kong, UW 20-132), and the relevant French authorities (CPP, CCITRS and CNIL). This study was also monitored by the Ethics Board of Institut Pasteur (EVOIMMUNOPOP-281297). All manipulations of live viruses were performed in a high-security BSL-3 environment.

## Reporting summary

Further information on research design is available in the Nature Portfolio Reporting Summary linked to this article.

## Data availability

The scRNA-seq data generated and analysed in this study have been deposited in the Institut Pasteur data repository, OWEY, which is available online (https://doi.org/10.48802/owey.e4qn-9190). The genome-wide genotyping data generated or used in this study have been deposited in OWEY and can be accessed online (https://doi.org/10.48802/owey.pyk2-5w22). In accordance with the General Data Protection Regulation (GDPR) in force in the European Union, the aforementioned data can be accessed only from the institutional data repository after authorization by the relevant Data Access Committee (DAC). The DAC ensures that data access and use is authorized for academic research relating to the variability of the human immune response, as defined in the informed consent signed by research participants. COVID-19 GWAS summary statistics used in the present study can be downloaded from https://www.covid19hg.org/results/r7. Human (1000G data, low (phase 3) and high (NYGC) coverage), archaic (Vindija and Denisova) and ancestral (EPO6) genomes used can be retrieved from http://ftp.1000genomes.ebi.ac.uk/vol1/ftp/release/20130502 (1000G phase 3), https://www.internationalgenome.org/data-portal/data-collection/30x-grch38 (1000G high coverage), http://cdna.eva.mpg.de/neandertal/Vindija/ (archaic) and http://ftp.ensembl.org/pub/release-71/emf/ensembl-compara/epo_6_primate/ (EPO6). Uniformly processed summary statistics from GTEx lung tissue were downloaded from http://ftp.ebi.ac.uk/pub/databases/spot/eQTL/sumstats/ (GTEx/lung/ge/all: study_id: QTS000015, dataset_id: QTD000271, file: QTD000271.all.tsv.gz). Source data are provided with this paper.

## Code availability

All custom computer code or algorithms used in this study are available from GitHub (https://github.com/h-e-g/popCell_SARS-CoV-2).

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

**Acknowledgements** We thank all of the members of the Human Evolutionary Genetics Laboratory, B.-A. Olin in particular, and M. Van der Wijst, D. de Vries and L. Franke for discussions; L. Speidel for help in running Relate and interpreting the results; and A. Coen and F. Clément for assistance with sample collection. The Human Evolutionary Genetics Laboratory is supported by the Institut Pasteur, the Collège de France, the Centre Nationale de la Recherche Scientifique (CNRS), the Agence Nationale de la Recherche (ANR) grants COVID-19-POPCELL (ANR-21-CO14-0003-01), POPCELL-REG (ANR-22-CE12-0030-01) and COVIFERON (ANR-21-RHUS-0008), the EU HORIZON-HLTH-2021-DISEASE-04-07 grant UNDINE (no. 101057100), the French Government's Investissement d'Avenir program, Laboratoires d'Excellence 'Integrative Biology of Emerging Infectious Diseases' (ANR-10- LABX-62-IBEID) and 'Milieu Intérieur' (ANR-10-LABX-69-01), the Fondation pour la Recherche Médicale (Equipe FRM DEQ20180339214), the Fondation Allianz-Institut de France and the Fondation de France (no. 00106080). K.L., J.T.K.W. and M.P. are supported by the Health and Medical Research Fund Commissioned Research on the Novel Coronavirus Disease, Hong Kong SAR (COVID190126); S.A.V. by the Theme-based Research Scheme of the Research Grants Council of the Hong Kong SAR (T11-705/21-N, SAV: T11-712/19-N); and M.P., R.B. and D.D. by InnoHK, an initiative of the Innovation and Technology Commission, the Government of the Hong Kong SAR. Figure 1a and Supplementary Fig. 3a were created using BioRender.

**Author contributions** M.R. and L.Q.-M. conceived and supervised the study. Y.A., A.B., M.O. and M.R. designed experiments. Y.A., A.B., Z.L., M.O. and S.H.M. conducted the experiments. Y.A., M.O., J.M.-R., G.K. and M.R. designed and performed computational analyses. V.B. conducted the SIMOA experiments. G.L.-R., C.-K.L., K.L., J.T.K.W., M.P., R.B. and S.A.V. managed clinical protocols and recruited donors. N.S., G.B.-S., S.P. and O.S. obtained SARS-CoV-2 strains and helped to design the stimulation experiments. C.d.C., M.M., M.H., V.L., F.L., R.P.-R., L.A., J.-L.C., D.D. and E.P. advised on experiments and data analyses and interpretation. Y.A., A.B., M.R. and L.Q.-M. interpreted the data and wrote the manuscript, with input from all of the authors.

**Competing interests** The authors declare no competing interests.

**Additional information**
**Correspondence and requests for materials** should be addressed to Maxime Rotival or Lluis Quintana-Murci.

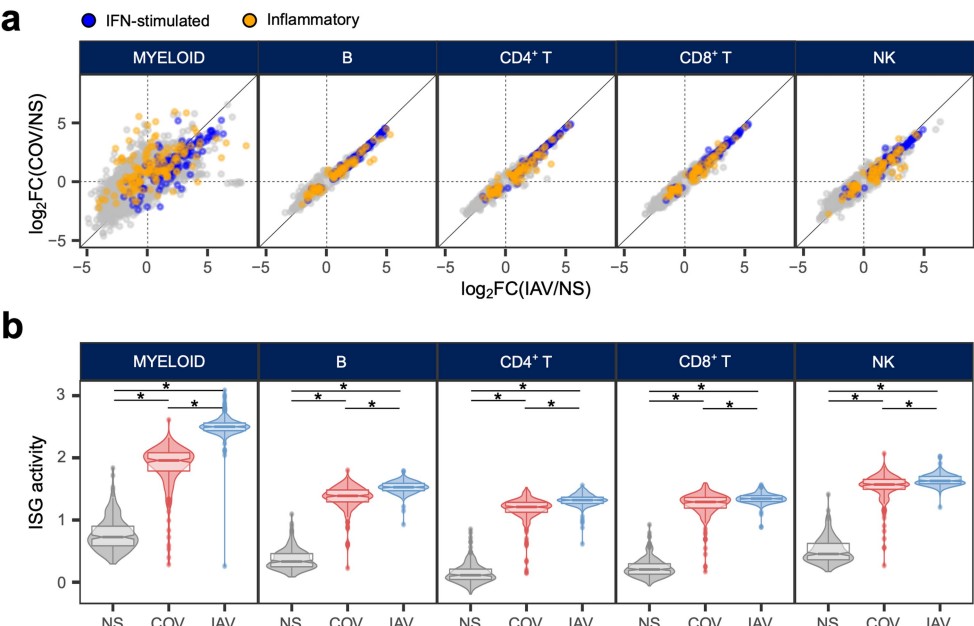

**Extended Data Fig. 1 | Transcriptional responses to SARS-CoV-2 and IAV stimulation. a**, Comparison of transcriptional responses to SARS-CoV-2 and IAV across major immune lineages. Hallmark inflammatory and interferon-stimulated genes are highlighted in orange and blue, respectively. **b**, Distribution of ISG activity in the non-stimulated state and in response to SARS-CoV-2 (COV) and influenza A virus (IAV) across the five major immune lineages. For each lineage and set of stimulation conditions, the violins and boxplots show the distribution of ISG activity scores across all $n = 222$ independent biological samples (middle line: median; box limits: upper and lower quartiles; whiskers: 1.5× interquartile range; points: outliers). *: Wilcoxon's two-sided signed-rank $p$-value $< 2.2 \times 10^{-16}$.

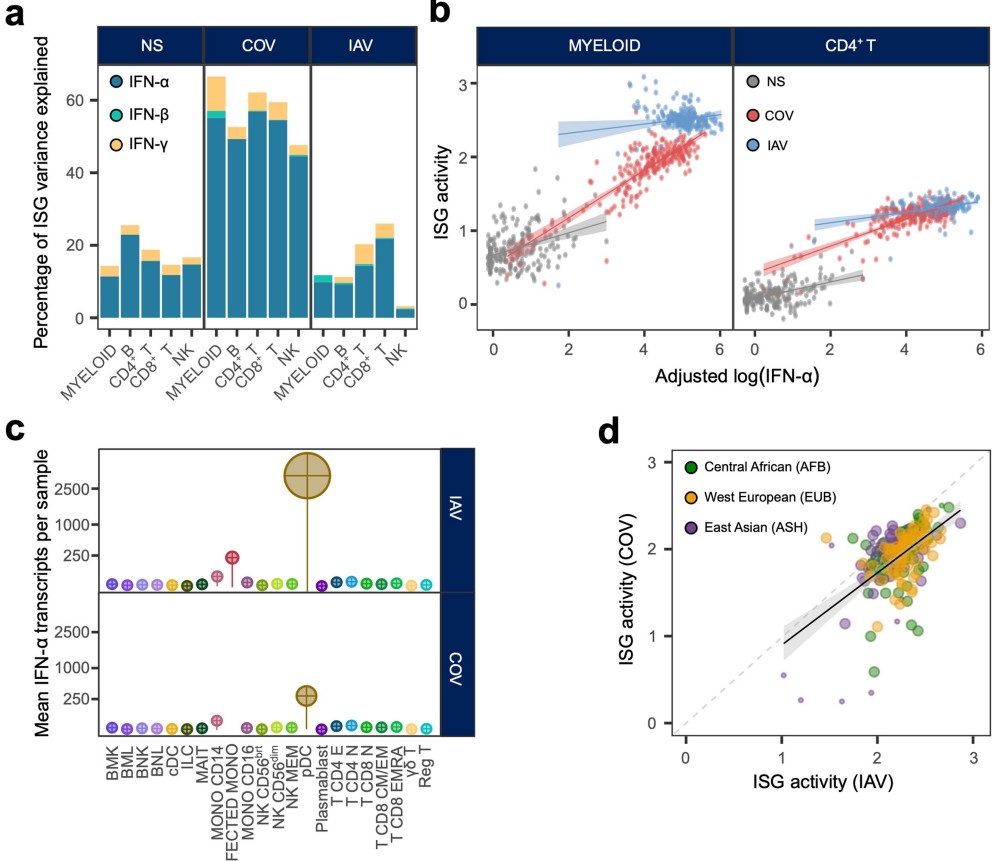

**Extended Data Fig. 2 | Drivers of population variation in expression of interferon-stimulated genes. a**, Proportion of the variance of ISG activity explained by IFN-α, IFN-β and IFN-γ in the non-stimulated condition and in response to SARS-CoV-2 and IAV, across the five major immune lineages. **b**, Correlation between levels of IFN-α in the supernatants (measured by SIMOA) and ISG activity in myeloid and CD4⁺ T cells, adjusted for cellular mortality. Each dot represents a sample (donor × condition) and is coloured according to its stimulation condition (grey: NS, red: COV and blue: IAV). For each lineage and set of stimulation conditions, lines show the expected ISG activity in each sample given the concentration of IFN-α; shaded error bands show the 95% CI (mean ± 2 SEM) around this estimate. **c**, Relative expression of IFN-α-encoding transcripts by each immune cell type in response to SARS-CoV-2 and IAV. Bar lengths indicate the mean number of IFN-α transcripts

contributed by each cell type to the overall pool (cell type frequency × mean number of IFN-α transcripts per cell). Dot area is proportional to the mean level of IFN-α transcripts in each cell type (counts per million). No value is reported in the SARS-CoV-2 condition for infected monocytes as this cell population is specific to the IAV condition. **d**, Correlation of ISG activity scores between SARS-CoV-2 and IAV-stimulated samples. Each dot corresponds to a single individual (*n* = 222) and its colour indicates the self-reported ancestries of the individual concerned (AFB: Central African; EUB: West European; ASH: East Asian). Shaded error band shows the 95% confidence interval (mean ± 2 SEM) of the expected ISG activity in COV-stimulated sample, given ISG activity in IAV-stimulated samples. Samples with a cellular viability below the 10th percentile are indicated by smaller dots.

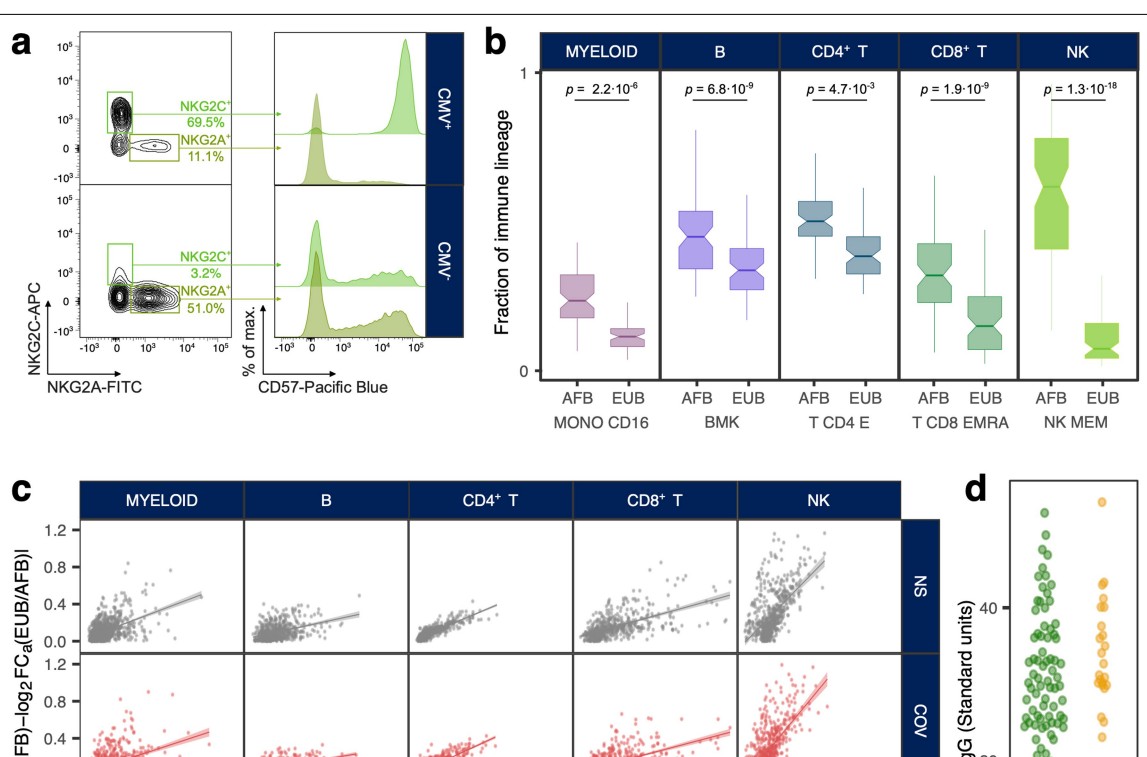

**Extended Data Fig. 3 | Population differences in cellular composition and transcriptional responses to viral stimulation. a**, Validation of the memory-like NK fraction. Flow cytometry data for representative CMV⁺ and CMV⁻ donors, highlighting the higher percentage of memory-like NK cells (NKG2C⁺, NKG2A⁻, CD57⁺) among CMV⁺ donors than among CMV⁻ donors. **b**, Population variation in the percentage of CD16⁺ monocytes, memory lymphocyte subsets and memory-like NK cells. For each major immune lineage, the cell type differing most strongly in frequency between AFB ($n = 80$) and EUB ($n = 80$) donors is shown. Boxplots show the distribution of the percentage of the target cell type in the corresponding lineage in each population (middle line: median; box limits: upper and lower quartiles; whiskers: 1.5× interquartile range). Wilcoxon's two-sided rank-sum $p$-values are shown. **c**, Effect of adjusting for cellular composition on the absolute differences in expression between AFB

and EUB donors, as a function of absolute differences in expression between the two cell types differing most in frequency between these populations (Supplementary Table 4b). For each gene, lineage and condition, the effect of adjustment is measured by the difference between raw ($\log_2FC_r$) and adjusted ($\log_2FC_a$) $\log_2$fold-changes. For each lineage and stimulation condition, lines show the expected change in the magnitude of population gene expression differences after adjusting for cellular composition, conditional on absolute differences in expression between the two cell types that differ most in frequency between these populations; shaded error bands show the 95% CI (mean ± 2 SEM) around this estimate. **d**, Serology assays for CMV across donors according to ancestries. Each dot represents a donor and is coloured according to ancestries (AFB: Central Africans, EUB: West Europeans). The grey line represents the detection threshold used to identify a donor as seropositive.

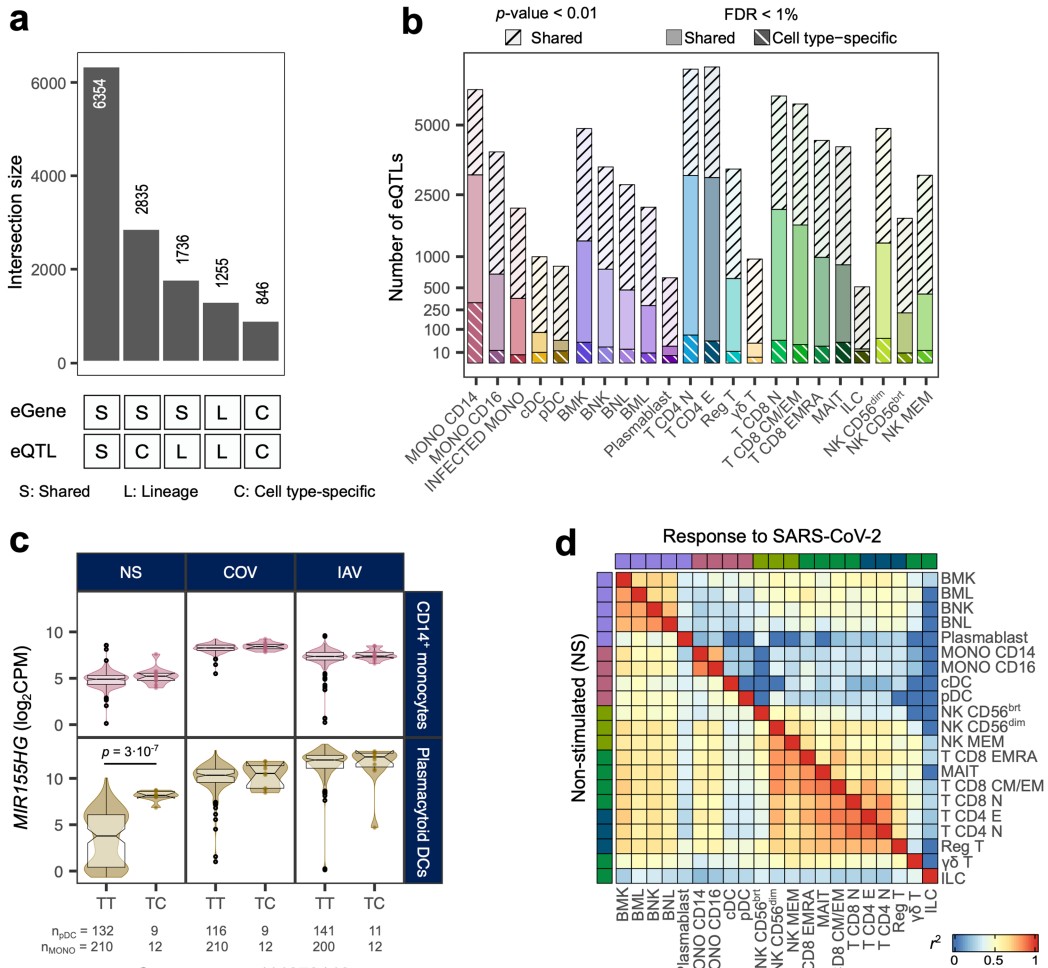

**Extended Data Fig. 4 | Mapping of expression quantitative trait loci at cell-type resolution. a**, Overlap of eQTLs and eGenes (i.e., genes with an eQTL) detected during the mapping of eQTLs at the immune lineage and cell type levels. **b**, Total number of eQTLs detected in each of the 22 different cell types. Coloured bars indicate the number of genome-wide significant eQTLs in each cell type, white stripes (bottom) indicate cell type-specific eQTLs (two-sided Student's *t*-test Benjamini-Hochberg-adjusted *p*-value <0.01; nominal *p*-value >0.01 in all other cell types), and black stripes (top) indicate the total number of eQTLs detected in each cell type including eQTLs from other cell types replicated at a *p*-value <0.01. **c**, Example of a pDC-specific eQTL for *MIR155HG*.

*MIR155HG* expression levels in pDCs and CD14+ monocytes according to rs114273142 genotype in non-stimulated (NS), SARS-CoV-2-stimulated (COV) and influenza A virus-stimulated (IAV) conditions (middle line: median; box limits: upper and lower quartiles; whiskers: 1.5× interquartile range; points: outliers). The number *n* of independent biological samples is indicated where relevant. **d**, Correlation of eQTL (NS; lower triangle) and reQTL (response to SARS-CoV-2; upper triangle) effect sizes across cell types. For each pair of cell types, Spearman's correlation coefficient was calculated for the effect sizes (β) of eQTLs that are significant at a nominal two-sided Student's *t*-test *p*-value <0.01 in each cell type.

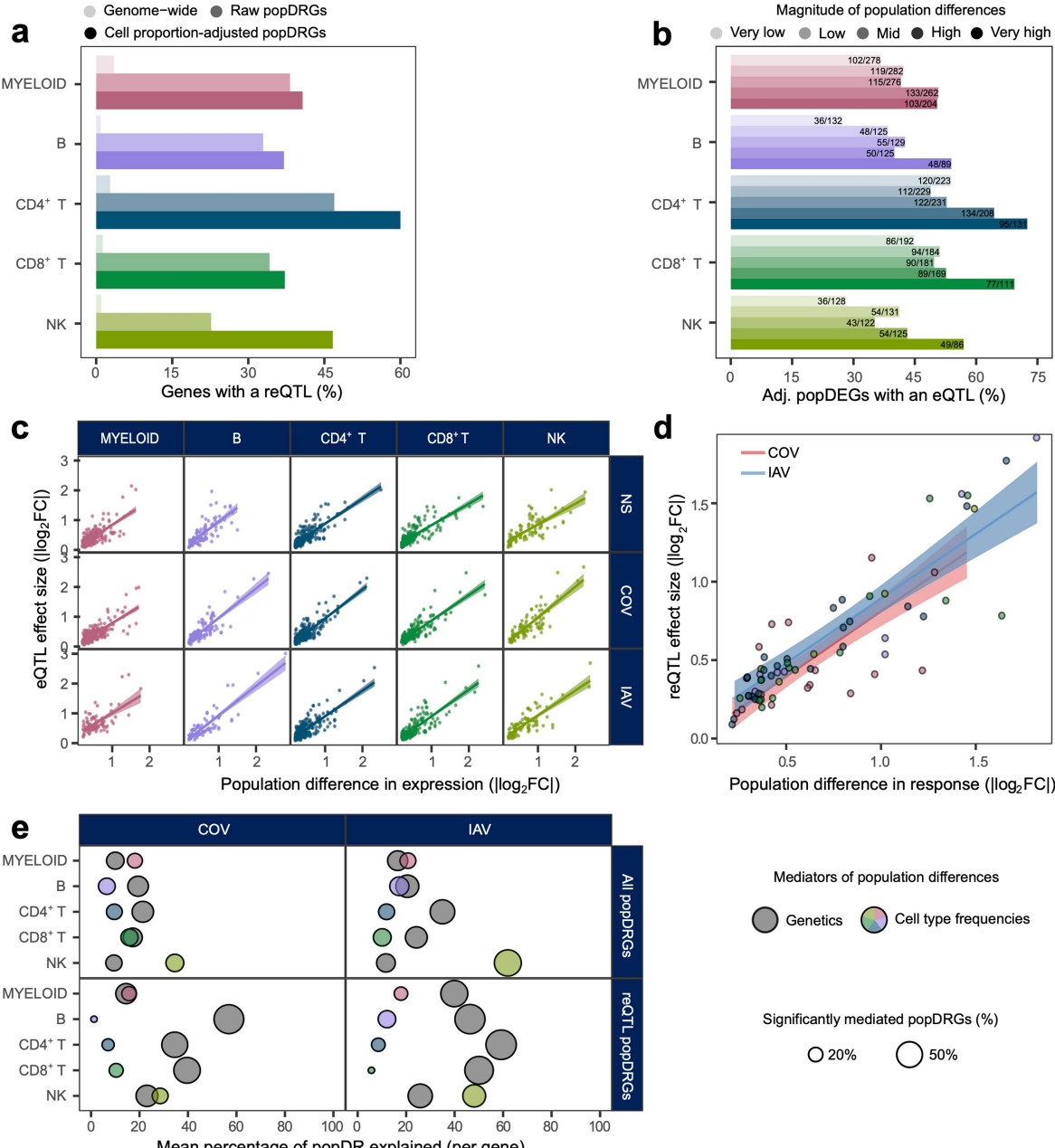

**Extended Data Fig. 5 | Contribution of genetics to population differences in response to RNA viruses. a**, Enrichment in reQTLs among popDRGs. For each lineage, bars indicate the percentage of genes with a significant reQTL, both genome-wide and among the popDRGs identified, before or after adjustment for cell composition (referred to as "adjusted" and "raw" respectively). **b**, Percentage of popDEGs with an eQTL according to the magnitude of differences in expression. In each lineage, popDEGs are assigned to one of five magnitude groups based on quintiles of $\log_2$ fold-change between the AFB and EUB populations. For each lineage and magnitude group, the number of popDEGs with an eQTL and the total number of popDEGs are reported. **c**, Relationship between eQTL effect sizes and population differences in expression (popDEGs only). **d**, Relationship between reQTL effect sizes and population differences in response to immune stimulation (popDRGs only). For each stimulation

condition, the regression line is computed jointly across all lineages. **c** and **d**, Lines show expected (r)eQTL effect sizes conditional on the magnitude of population differences in gene expression or in response to viral stimulation; shaded error bands show the 95% CI (mean ± 2 SEM) around this estimate. **e**, Contribution of genetics and cell composition to population differences in response to stimulation by COV and IAV. For each lineage and stimulation condition, the x-axis indicates the mean percentage of population differences in response to stimulation mediated by either genetics or cell composition, across all popDRGs (upper panels) or the set of popDRGs with a significant reQTL (lower panels). The size of the dots reflects the percentage of genes with a significant mediated effect (FDR<1%). The statistical significance of mediated effects for each gene is reported in Supplementary Table 6.

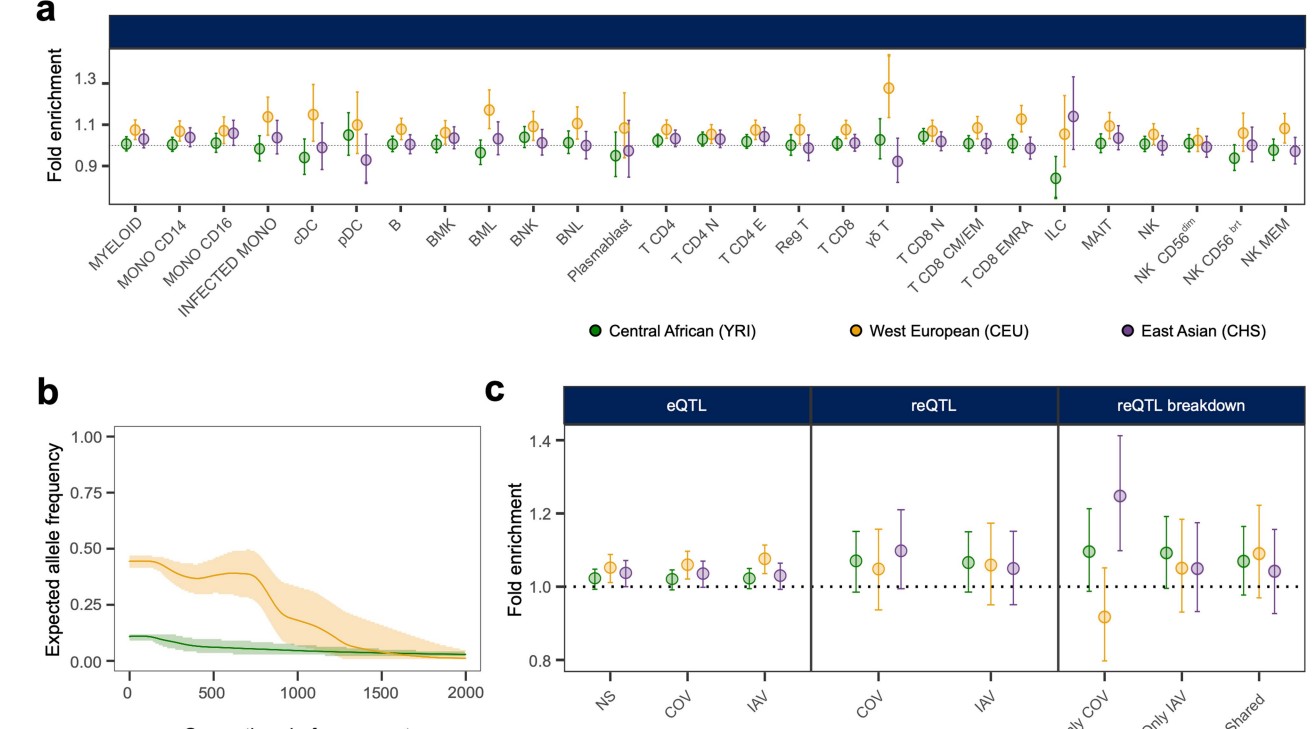

**Extended Data Fig. 6 | Positive selection signals across cell types and populations. a**, Fold-enrichments (FE) of eQTLs in local adaptation signals across the 22 cell types. Adaptive loci are defined separately in Central Africans (YRI), West Europeans (CEU) and East Asians (CHS), based on the population branch statistic (top 1% PBS). Occurrence of adaptive signals at reQTLs is compared to randomly selected variants, matched for MAF, distance to nearest gene, and LD score. **b**, Allele frequency trajectories over the past 2,000 generations in YRI (green) and CEU (yellow) of the *GBP7* reQTL (rs1142888-G). Lines indicate the maximum a posteriori estimate of allele frequency at each epoch in each population; shaded areas indicate the 95% CIs around these estimates (2.5th –97.5th percentiles of posterior distribution). **c**, Fold-enrichments (FE) of eQTLs and reQTLs in local adaptation signals (top 1% PBS), for eQTLs and reQTLs relative to random variants, matched for MAF, distance to nearest gene, and LD score. reQTLs are analysed either for each stimulus separately (reQTL) or splitting into stimulus-specific and shared reQTLs (reQTL breakdown). **a** and **c**. Data are presented as the mean and 2.5th –97.5th percentiles (95% CIs) of FE observed over $N$ =10,000 resamplings.

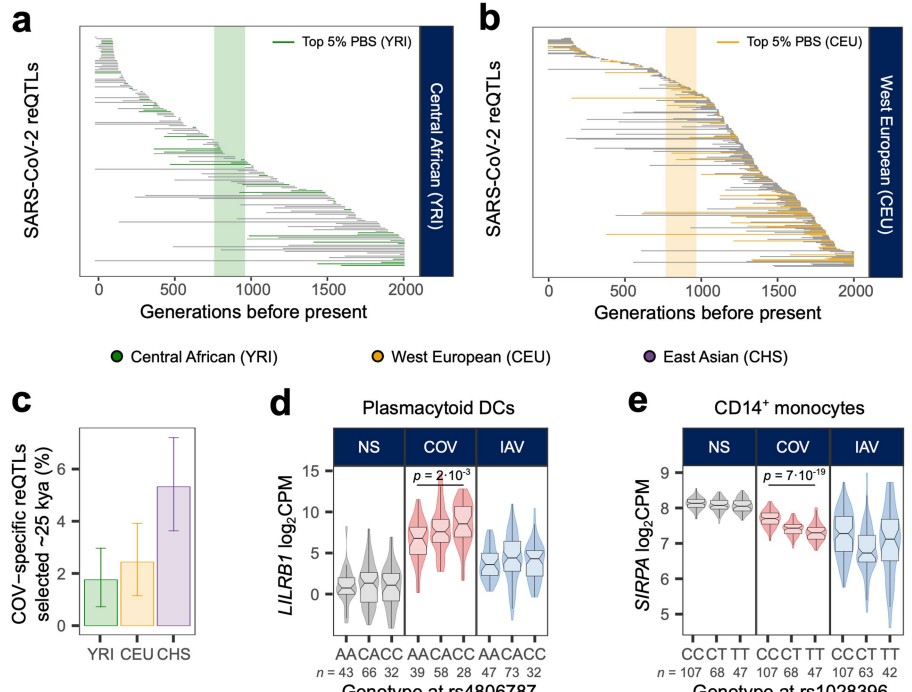

**Extended Data Fig. 7 | Onset of positive selection events at SARS-CoV-2 reQTLs. a** and **b**, Estimated period of selection over the past 2,000 generations, for 148 and 279 SARS-CoV-2 reQTLs with significant evidence of natural selection in Central Africans and West Europeans, respectively (max. |$Z$-score| > 3). In both panels, variants presenting strong signals of positive selection (i.e., top 5% for PBS) are shown in colour. The transparent rectangle highlights the period between 770 and 970 generations ago (i.e., 21.5-27.2 thousand years ago) associated with genetic adaptation targeting host coronavirus-interacting proteins in East Asians. Variants are ordered along the $x$-axis in descending order of time to onset of natural selection. **c**, Percentage of SARS-CoV-2-specific reQTLs presenting selection signals in different populations, between 770 and 970 generations ago. Data are presented as the median and 2.5th – 97.5th percentiles (95% CIs) of percentages observed over $N$ = 1,000 resamplings. **d** and **e**, Examples of SARS-CoV-2-induced reQTLs at *LILRB1* (rs4806787) in plasmacytoid dendritic cells (pDCs) and *SIRPA* (rs1028396) in CD14+ monocytes. Student's two-sided $t$-test $p$-values < 0.01 are shown; middle line: median; notches: 95% CIs of median, box limits: upper and lower quartiles; whiskers: 1.5× interquartile range; points: outliers. The number $n$ of independent biological samples is indicated where relevant.

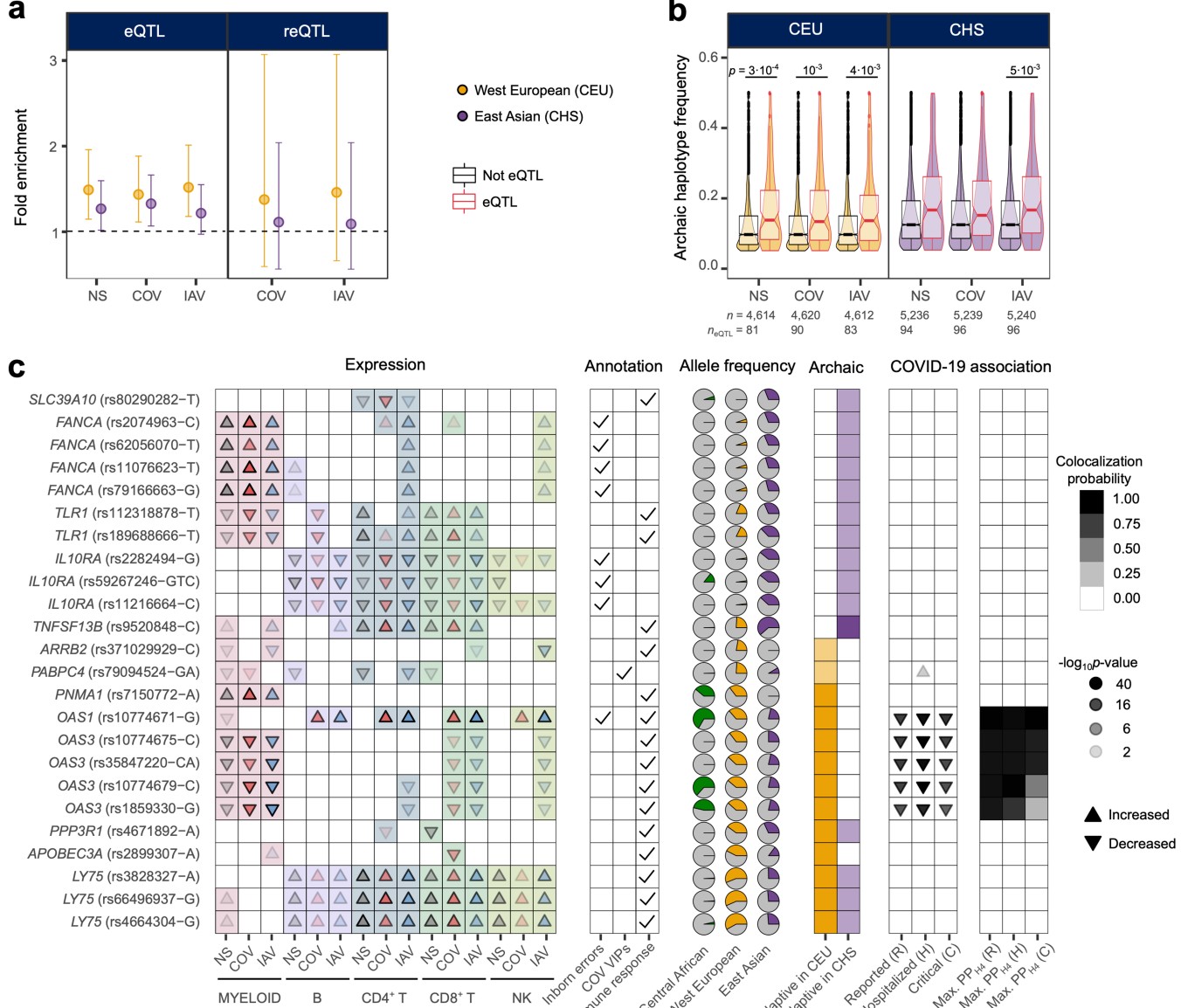

**Extended Data Fig. 8 | Adaptive introgression at regulatory loci.**
**a**, Enrichment of eQTLs and reQTLs in introgressed haplotypes (mean and 2.5th –97.5th percentiles of observed/expected ratios across $N$ = 10,000 resamplings). **b**, For each population and condition, the frequencies of introgressed haplotypes are compared according to their effects on gene expression (eQTL *vs.* non-eQTL; Benjamini-Hochberg-adjusted two-sided Wilcoxon's rank-sum *p*-values < 0.01 are shown). The numbers of independent SNPs and eQTLs considered, *n* and $n_{eQTL}$ respectively, are indicated. Middle line: median; notches: 95% CIs of median, box limits: upper and lower quartiles; whiskers: 1.5× interquartile range; points: outliers. Benjamini-Hochberg -adjusted two-sided Student's *t*-test *p*-values < 0.01 are shown. For **a** and **b**, each population was downsampled to the same number of donors prior to eQTL mapping to avoid biases due to differences in statistical power. **c**, Adaptively

introgressed eQTLs of host defence genes. From left to right: (i) effects of the introgressed allele on gene expression across immune lineages and stimulation conditions, (ii) clinical and functional annotations of associated genes, (iii) present-day population frequencies of the introgressed alleles, (iv) percentile of archaic allele frequency at the locus (CEU and CHS; dark shades: top 1%, light shades: top 5%), and (v) effects of the target allele on COVID-19 risk (infection, hospitalization, and critical state). Arrows indicate the increase/decrease in gene expression or disease risk with each copy of the introgressed allele. Opacity increases with significance (two-sided Student's *t*-test -log10 *p*-value). In the leftmost panel, arrow colours indicate the stimulation condition (grey: NS, red: COV, blue: IAV). For each eQTL, the introgressed allele is defined as the allele segregating with the archaic haplotype in Eurasians.

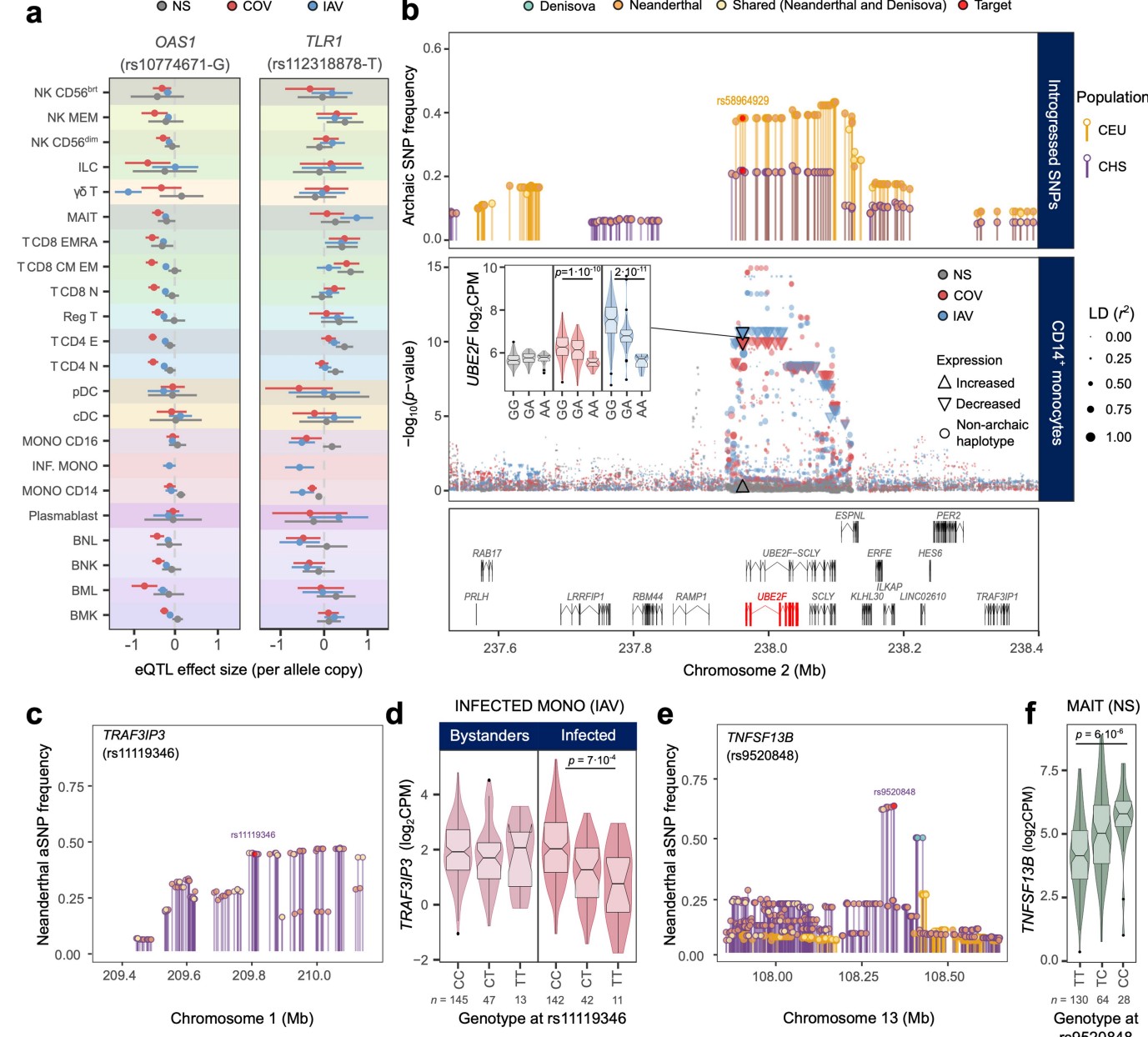

**Extended Data Fig. 9 | Cell type-dependent effects on gene expression of Neanderthal introgression. a**, Effects on gene expression of two loci (*OAS1* and *TLR1*) presenting strong evidence of adaptive introgression. For each locus, estimated eQTL effect size and 95% CIs estimated in 222 unrelated donors are shown across 22 cell types and stimulation conditions. **b-f**, Examples of adaptive introgression at three introgressed loci (*UBE2F*, *TRAF3IP3* and *TNFSF13B)*. **b**, Upper panel: frequency and origin of archaic alleles at the *UBE2F* locus. Each dot represents an archaic allele, and its colour indicates whether it was unique to the Vindija Neanderthal genome (orange), shared between the Vindija Neanderthal and Denisova genomes (light yellow), or specific to Denisova (green). The reQTL index SNP is shown in red (rs58964929). The *y*-axis indicates allele frequency in West Europeans (CEU, yellow) and East Asians (CHS, purple). Middle panel: monocyte eQTL *p*-values (two-sided Student's *t*-test), colour-coded according to stimulation conditions (grey: non-stimulated

(NS), red: SARS-CoV-2-stimulated (COV), blue: IAV-stimulated (IAV)). Each dot represents a SNP. Dot area is proportional to the LD (*r²*) values between the SNP and nearby archaic alleles. For archaic alleles, arrows indicate direction of allele effect on gene expression. Lower panel: Genes at locus, with *UBE2F* highlighted in red. **c**, and **e**, same as **b** (upper panel) for *TRAF3IP3* and *TNFSF13B*. **d**, The Neanderthal-introgressed eQTL at *TRAF3IP3* is apparent only in IAV-infected monocytes and not detected in bystander cells (stimulated but not infected). **f**, Effects of the introgressed eQTL at *TNFSF13B* in MAIT cells (i.e., the cell type with the largest effect size). For **b**, **d** and **f**, middle line: median; box limits: upper and lower quartiles; whiskers: 1.5× interquartile range; points: outliers. Benjamini-Hochberg-adjusted two-sided Student's *t*-test *p*-values < 0.01 are shown, and the number *n* of independent biological samples is indicated where relevant.

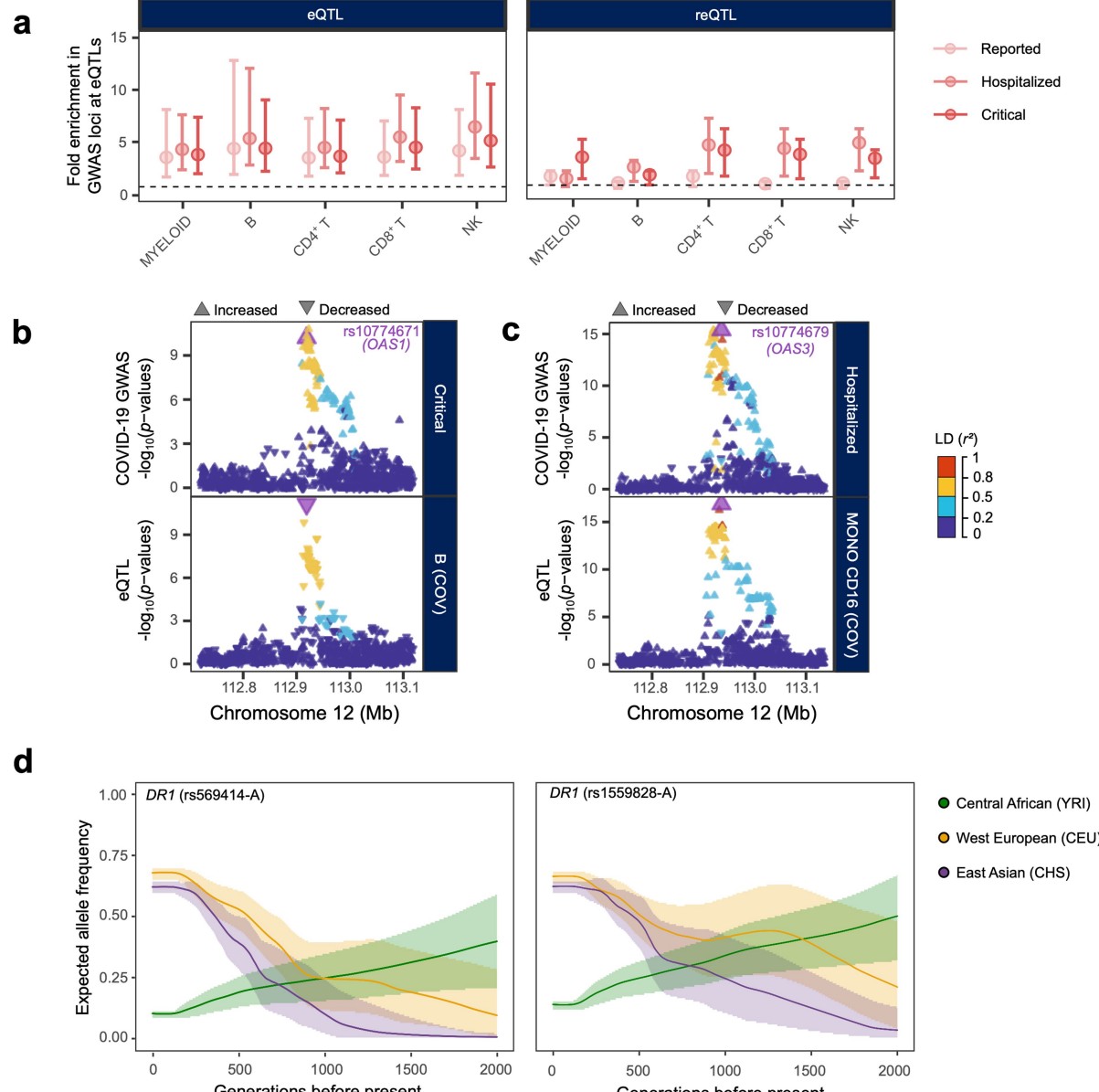

**Extended Data Fig. 10 | Colocalization of eQTLs and reQTLs with COVID-19-associated loci. a**, Enrichment in COVID-19-associated loci at eQTLs and reQTLs in each major lineage. Data are presented as the mean and 2.5th −97.5th percentiles (95% CIs) of fold-enrichments observed over $N = 10,000$ resamplings. **b** and **c**, Colocalization of eQTLs with COVID-19 GWAS hits at the *OAS1-3* locus. For each eQTL, the upper panel shows the two-sided Student's *t*-test −log₁₀ (*p*-value) profile for association with COVID-19 phenotypes and the lower panel represents the profile of −log₁₀ (*p*-values) for association with expression in a representative cell type. Arrows indicate the direction of the effect at each SNP.

The colour code reflects LD ($r^2$) with the consensus SNP, shown in purple, identified by colocalization analysis. **d**, Allele frequency trajectories over the last 2,000 generations in the three populations of two *DR1* eQTLs (rs569414 and rs1559828) that colocalize with COVID-19 severity loci (alleles rs569414-A and rs1559828-A are associated with decreased COVID-19 severity). Full lines indicate the maximum a posteriori estimate of allele frequency at each epoch in each population; shaded areas indicate the 95% CIs around these estimates (2.5th–97.5th percentile of the posterior distribution).

# Reporting Summary

## Statistics

For all statistical analyses, confirm that the following items are present in the figure legend, table legend, main text, or Methods section.

| n/a | Confirmed | |
|---|---|---|
| ☐ | ☒ | The exact sample size (*n*) for each experimental group/condition, given as a discrete number and unit of measurement |
| ☐ | ☒ | A statement on whether measurements were taken from distinct samples or whether the same sample was measured repeatedly |
| ☐ | ☒ | The statistical test(s) used AND whether they are one- or two-sided *Only common tests should be described solely by name; describe more complex techniques in the Methods section.* |
| ☐ | ☒ | A description of all covariates tested |
| ☐ | ☒ | A description of any assumptions or corrections, such as tests of normality and adjustment for multiple comparisons |
| ☐ | ☒ | A full description of the statistical parameters including central tendency (e.g. means) or other basic estimates (e.g. regression coefficient) AND variation (e.g. standard deviation) or associated estimates of uncertainty (e.g. confidence intervals) |
| ☐ | ☒ | For null hypothesis testing, the test statistic (e.g. *F*, *t*, *r*) with confidence intervals, effect sizes, degrees of freedom and *P* value noted *Give P values as exact values whenever suitable.* |
| ☒ | ☐ | For Bayesian analysis, information on the choice of priors and Markov chain Monte Carlo settings |
| ☒ | ☐ | For hierarchical and complex designs, identification of the appropriate level for tests and full reporting of outcomes |
| ☐ | ☒ | Estimates of effect sizes (e.g. Cohen's *d*, Pearson's *r*), indicating how they were calculated |

*Our web collection on statistics for biologists contains articles on many of the points above.*

## Software and code

Policy information about availability of computer code

| | |
|---|---|
| Data collection | GenomeStudio (v.2011.1), FlowJo software (v10.7.1), STARsolo aligner (v2.7.8a) |
| Data analysis | PLINK (v1.9), GATK (v4.1.2.0), bcftools (v1.16), tabix (v0.2.6), Beagle5.1 (18may20.d20), SHAPEIT4 (v4.2.1), samtools (v1.10), Demuxlet (v0.1), Freemuxlet (v0.1), R (v4.1.0) with the following packages : scran (v1.20.1), SingleCellExperiment (v1.14.1), batchelor (v1.8.1), Seurat (v4.1.1), Harmony (v0.1.0), kBET (v0.99.6), scuttle (v1.2.1), lme4 (v1.1-27.1), sandwich (v2.5-1), lmtest (v0.9-40), care (v1.1.11), fgsea (v1.18.1), MatrixEQTL (v2.3), sva (v3.40.0), SuSiER (v0.11.42), mediation (v4.5.0), and CrossMap (v0.6.3). CLUES (commit n°7371b86, 27 may 2021), Relate (v1.1.8), SliM (v.4.0.1), S-prime (v.07Dec18.5e2), CRF (Sankararaman et al., Nature 2014), coloc (v 5.1.0), S-PrediXcan (v0.6.11), other custom-generated scripts are deposited on GitHub (www.github.com/h-e-g/popCell_SARS-CoV-2). |

For manuscripts utilizing custom algorithms or software that are central to the research but not yet described in published literature, software must be made available to editors and reviewers. We strongly encourage code deposition in a community repository (e.g. GitHub). See the Nature Portfolio guidelines for submitting code & software for further information.

## Data

Policy information about availability of data

All manuscripts must include a data availability statement. This statement should provide the following information, where applicable:
- Accession codes, unique identifiers, or web links for publicly available datasets
- A description of any restrictions on data availability
- For clinical datasets or third party data, please ensure that the statement adheres to our policy

The single-cell RNA sequencing data generated and analyzed in this study have been deposited in the Institut Pasteur data repository, OWEY, which can be accessed at: https://doi.org/10.48802/owey.e4qn-9190. The genome-wide genotyping data generated or used in this study have been deposited in OWEY and can be accessed at https://doi.org/10.48802/owey.pyk2-5w22. Data access and use is restricted to academic research related to the variability of the human immune response. COVID-19 GWAS summary statistics used in the present study can be downloaded from https://www.covid19hg.org/results/r7. Human (1000G data, low [phase 3] and high [NYGC] coverage), archaic (Vindija and Denisova) and ancestral (EPO6) genomes used can be retrieved from ftp://ftp.1000genomes.ebi.ac.uk/vol1/ftp/release20130502 (1000G phase 3), https://www.internationalgenome.org/data-portal/data-collection/30x-grch38 (1000G high coverage), http://cdna.eva.mpg.de/neandertal/Vindija/ (archaic) and ftp://ftp.ensembl.org/pub/release-71/emf/ensembl-compara/epo_6_primate/ (EPO6), respectively. Uniformly processed summary statistics from GTEx lung tissue were downloaded from http://ftp.ebi.ac.uk/pub/databases/spot/eQTL/sumstats/ (GTEx/lung/ge/all: study_id: QTS000015, dataset_id: QTD000271, file: QTD000271.all.tsv.gz).

## Human research participants

Policy information about studies involving human research participants and Sex and Gender in Research.

| Reporting on sex and gender | The EvoImmunoPop cohort analyzed in this study (EUB and AFB donors) is constituted of self-reported male individuals. This choice, which was originally made in an effort to minimize non-genetic variation and increase power for the mapping of eQTLs, limits the generality of our findings, which are likely to be male-biased. However, this bias is partially balanced by the presence of female individuals among ASH donors (41 self-reported females). In addition, recent work has shown that the genetic basis of gene expression variation is largely shared between males and females (see 10.1126/science.aba3066), suggesting adequate transferability of our results. Self-reported gender was compared with chromosomal sex (inferred based on genotyping data) and was found to be concordant in all but one individual. All analyses including ASH individuals, who include both males and females, were adjusted for chromosomal sex. |
|---|---|
| Population characteristics | The study populations were composed of 80 male donors of self-reported European descent living in Belgium (EUB), 80 male donors of self-reported African descent living in Belgium (AFB), and 71 donors of East Asian descent living in Hong Kong (ASH; 30 males and 41 females). Inclusion of EUB and AFB was restricted to nominally healthy donors between 19 and 50 years of age at the time of sample collection (2012-2013). Inclusion of ASH donors was restricted to nominally healthy donors between 19 and 63 years of age and who were SARS-CoV-2 naive at the time of the sample collection (2020). |
| Recruitment | Recruitment of donors of West European and Central African ancestries was performed at the Center for Vaccinology (CEVAC) of Ghent University Hospital (Ghent, Belgium), based on self-reported ancestry. Recruitment of donors of East Asian ancestry was performed at the School of Public Health of University of Hong Kong (Hong Kong SAR, China). In both cohorts, sampling of related individuals was avoided because relatedness can confound population genetic analyses. We do not anticipate any bias in our results that could be due to this recruitment strategy. |
| Ethics oversight | All donors were sampled after written informed consent had been obtained, and the study was approved by the ethics committee of Ghent University (Belgium, n° B670201214647), the Institutional Review Board of the University of Hong Kong (n° UW 20-132), and the relevant French authorities (CPP, CCITRS and CNIL). This study was also monitored by the Ethics Board of Institut Pasteur (EVOIMMUNOPOP-281297). |

Note that full information on the approval of the study protocol must also be provided in the manuscript.

# Field-specific reporting

Please select the one below that is the best fit for your research. If you are not sure, read the appropriate sections before making your selection.

☒ Life sciences    ☐ Behavioural & social sciences    ☐ Ecological, evolutionary & environmental sciences

For a reference copy of the document with all sections, see nature.com/documents/nr-reporting-summary-flat.pdf

# Life sciences study design

All studies must disclose on these points even when the disclosure is negative.

| Sample size | Target sample sizes of 80 individuals per population and stimulation condition, and ~1,500 cells per sample, were determined based on (i) sample size of previous single-cell eQTL studies (Randolph et al. Science, 2021) and (ii) to ensure >80% power for the detection of eQTLs (MAF>5%) with effect sizes higher than 0.2, at a family-wise error rate of 5%, assuming 10 million SNP-gene pairs tested and residual variance in gene expression of 0.2. |
|---|---|

| Data exclusions | One East Asian donor was excluded due to the presence of low-quality cells upon thawing. Stimulation experiments were performed on the remaining 230 individuals, for a target number of 770 samples (8 individuals × 7 conditions (2 virus × 2 time points + non stimulated × 3 time points) + 222 individuals × 3 conditions + 48 replicates (16 individuals ×3 conditions)), each processed on two separate libraries. One library (L117) failed during the library preparation stage and was thus discarded. Finally, eight East Asian donors were discarded because the number of cells recovered after quality control was too low (< 500 singlets in at least one sample). |
|---|---|
| Replication | The reproducibility of scRNA-seq profiles was evaluated in two ways. First, we processed cells from each sample on two separate 10x Genomics libraries in each run, enabling us to assess technical variability associated with library preparation for each sample. Second, for 16 samples, we performed an additional run, allowing us to evaluate replicability across experiments using the same protocol on samples from the same individual.  All attempts at replication were successful (see Supplementary Figure 5). |
| Randomization | We used a balanced design where each experimental run was mostly composed of an approximately equal number of donors of African, European, and East Asian ancestries. Donors were randomly selected within each population. In each experimental run, resting and stimulated cells from 12 different donors were pooled together according to a pre-established scheme (four resting, four SARS-CoV-2 and four IAV samples per library, eight libraries per experimental run) prior to library preparation. cDNA libraries were then pooled and sequenced by groups of eight within each sequencing flow-cell (either pooling libraries from each run, or from 2 different runs). Thus, each flow-cell contained a randomized, balanced set of > 48 samples. Note that within each population (AFB, EUB and ASH) genotypes are randomized by meiotic recombination, which ensures adequate mixing of alleles across batches. |
| Blinding | When performing stimulations, researchers were blinded to the population of origin of the individual. Genotypes and environmental exposures were unknown a priori during data collection, and researchers were blinded to the population of origin of the individual when assessing serologies. Sequencing and quantification of gene expression were performed using automated pipelines and did not take into account the identity of the sample or the population of origin. |

# Reporting for specific materials, systems and methods

We require information from authors about some types of materials, experimental systems and methods used in many studies. Here, indicate whether each material, system or method listed is relevant to your study. If you are not sure if a list item applies to your research, read the appropriate section before selecting a response.

## Materials & experimental systems

| n/a | Involved in the study |
|---|---|
| ☐ | ☒ Antibodies |
| ☒ | ☐ Eukaryotic cell lines |
| ☒ | ☐ Palaeontology and archaeology |
| ☒ | ☐ Animals and other organisms |
| ☒ | ☐ Clinical data |
| ☒ | ☐ Dual use research of concern |

## Methods

| n/a | Involved in the study |
|---|---|
| ☒ | ☐ ChIP-seq |
| ☐ | ☒ Flow cytometry |
| ☒ | ☐ MRI-based neuroimaging |

## Antibodies

| Antibodies used | We used the following antibodies for our study :<br>1) CITE-seq: full description of the antibodies is provided in the Supplementary Table S3B (clone, reference and supplier).<br> - TotalSeqTM-B 0046 anti-human CD8  (clone SK1, supplier Biolegend, reference: 344757, dilution: 1/50)<br> - TotalSeqTM-B 0047 anti-human CD56 (NCAM)  (clone 5.1H11 , supplier Biolegend, reference: 362561, dilution: 1/50)<br> - TotalSeqTM-B 0049 anti-human CD3  (clone SK7 , supplier Biolegend, reference: 344853, dilution: 1/50)<br> - TotalSeqTM-B 0050 anti-human CD19  (clone HIB19 , supplier Biolegend, reference: 302263, dilution: 1/50)<br> - TotalSeqTM-B 0053 anti-human CD11c  (clone S-HCL-3 , supplier Biolegend, reference: 371523, dilution: 1/50)<br> - TotalSeqTM-B 0063 anti-human CD45RA  (clone HI100 , supplier Biolegend, reference: 304161, dilution: 1/50)<br> - TotalSeqTM-B 0064 anti-human CD123  (clone 6H6 , supplier Biolegend, reference: 306047, dilution: 1/50)<br> - TotalSeqTM-B 0072 anti-human CD4  (clone RPA-T4 , supplier Biolegend, reference: 300565, dilution: 1/50)<br> - TotalSeqTM-B 0081 anti-human CD14  (clone M5E2 , supplier Biolegend, reference: 301857, dilution: 1/50)<br> - TotalSeqTM-B 0083 anti-human CD16  (clone 3G8 , supplier Biolegend, reference: 302063, dilution: 1/50)<br> - TotalSeqTM-B 0085 anti-human CD25  (clone BC96 , supplier Biolegend, reference: 302647, dilution: 1/50)<br> - TotalSeqTM-B 0154 anti-human CD27  (clone O323 , supplier Biolegend, reference: 302851, dilution: 1/50)<br> - TotalSeqTM-B 0159 anti-human HLA-DR  (clone L243 , supplier Biolegend, reference: 307661, dilution: 1/50)<br> - TotalSeqTM-B 0390 anti-human CD127 (IL-7Rα)  (clone A019D5 , supplier Biolegend, reference: 351354, dilution: 1/50)<br> - TotalSeqTM-B 0410 anti-human CD38  (clone HB-7 , supplier Biolegend, reference: 356639, dilution: 1/50)<br><br>2) Flow cytometry: full description of the antibodies is provided in the section Flow Cytometry of the Methods (clone and supplier).<br> - CD3 VioGreen (clone BW264/56, Miltenyi Biotec, dilution: 1/50),<br> - CD14 V500 (clone M5E2, BD Biosciences, dilution: 1/50),<br> - CD57 Pacific Blue (clone HNK-1, Biolegend, dilution: 1/20),<br> - NKp46 PE (clone 9E2/NKp46, BD Biosciences, dilution: 1/10),<br> - CD16 PerCP-Cy5.5 (clone 3G8, BD Biosciences, dilution: 1/20),<br> - CD56 APC-Vio770 (clone REA196, Miltenyi Biotec, dilution: 1/50),<br> - NKG2A FITC (clone REA110, Miltenyi Biotec, dilution: 1/50),<br> - NKG2C APC (clone REA205, Miltenyi Biotec, dilution: 1/50) |
|---|---|

3) SIMOA: full description of the antibodies is provided in the section Supernatants cytokine assays of the Methods (clone and supplier or origin).
 - IFN-α capture antibody (clone 8H1, supplier: Evitria, Switzerland, origin: APS1/APECED patient, concentration: 0.3 mg/mL)
 - IFN-α detector antibody (clone 12H5 ,supplier: Evitria, Switzerland, origin: APS1/APECED patient, concentration: 0.3 μg/mL)
 - IFN-γ capture antibody (clone MD-1, supplier: BioLegend, concentration: 0.3 mg/mL)
 - IFN-γ detector antibody (clone MAB285, supplier: R&D Systems, concentration: 0.3 μg/mL)
 - IFN-β capture antibody (clone 710322-9 IgG1 kappa, supplier: PBL Assay Science, origin mouse monoclonal antibody, concentration: 0.3 mg/mL)
 - IFN-γ detector antibody (710323-9 IgG1 kappa, supplier: PBL Assay Science, origin mouse monoclonal antibody, concentration: 0.3 μg/mL)

**Validation**

Validation of commercial antibodies was done on a regular internal quality control for each lot by the manufacturer.
Flow cytometry:
Miltenyi Biotech
https://www.miltenyibiotec.com/upload/assets/dataSheet_p42150_eng_GBR.pdf
https://www.miltenyibiotec.com/upload/assets/IM0022906.PDF
https://www.miltenyibiotec.com/upload/assets/dataSheet_p42217_eng_GBR.pdf
https://www.miltenyibiotec.com/upload/assets/dataSheet_p68857_eng_GBR.pdf
BD Biosciences
https://www.bdbiosciences.com/content/bdb/paths/generate-tds-document.us.561391.pdf
https://www.bdbiosciences.com/content/bdb/paths/generate-tds-document.us.562101.pdf
https://www.bdbiosciences.com/content/bdb/paths/generate-tds-document.us.560717.pdf
Biolegend
https://d1spbj2x7qk4bg.cloudfront.net/en-us/products/pacific-blue-anti-human-cd57-antibody-8827?pdf=true&displayInline=true&leftRightMargin=15&topBottomMargin=15&filename=Pacific%20Blue%E2%84%A2%20anti-human%20CD57%20Antibody.pdf&v=20220914123035

CITE-seq:
Biolegend
https://d1spbj2x7qk4bg.cloudfront.net/en-us/products/totalseq-b0046-anti-human-cd8-antibody-18042?pdf=true&displayInline=true&leftRightMargin=15&topBottomMargin=15&filename=TotalSeq%E2%84%A2-B0046%20anti-human%20CD8%20Antibody.pdf&v=20220902063018
https://d1spbj2x7qk4bg.cloudfront.net/en-us/products/totalseq-b0047-anti-human-cd56-ncam-antibody-18156?pdf=true&displayInline=true&leftRightMargin=15&topBottomMargin=15&filename=TotalSeq%E2%84%A2-B0047%20anti-human%20CD56%20(NCAM)%20Antibody.pdf&v=20210121043158
https://d1spbj2x7qk4bg.cloudfront.net/en-us/products/totalseq-b0049-anti-human-cd3-antibody-19288?pdf=true&displayInline=true&leftRightMargin=15&topBottomMargin=15&filename=TotalSeq%E2%84%A2-B0049%20anti-human%20CD3%20Antibody.pdf&v=20220907063026
https://d1spbj2x7qk4bg.cloudfront.net/en-us/products/totalseq-b0050-anti-human-cd19-antibody-16831?pdf=true&displayInline=true&leftRightMargin=15&topBottomMargin=15&filename=TotalSeq%E2%84%A2-B0050%20anti-human%20CD19%20Antibody.pdf&v=20221026111349
https://d1spbj2x7qk4bg.cloudfront.net/en-us/products/totalseq-b0053-anti-human-cd11c-antibody-18043?pdf=true&displayInline=true&leftRightMargin=15&topBottomMargin=15&filename=TotalSeq%E2%84%A2-B0053%20anti-human%20CD11c%20Antibody.pdf&v=20220820063106
https://d1spbj2x7qk4bg.cloudfront.net/en-us/products/totalseq-b0063-anti-human-cd45ra-antibody-16850?pdf=true&displayInline=true&leftRightMargin=15&topBottomMargin=15&filename=TotalSeq%E2%84%A2-B0063%20anti-human%20CD45RA%20Antibody.pdf&v=20220820063106
https://d1spbj2x7qk4bg.cloudfront.net/en-us/products/totalseq-b0064-anti-human-cd123-antibody-18968?pdf=true&displayInline=true&leftRightMargin=15&topBottomMargin=15&filename=TotalSeq%E2%84%A2-B0064%20anti-human%20CD123%20Antibody.pdf&v=20220830085839
https://d1spbj2x7qk4bg.cloudfront.net/en-us/products/totalseq-b0072-anti-human-cd4-antibody-16820?pdf=true&displayInline=true&leftRightMargin=15&topBottomMargin=15&filename=TotalSeq%E2%84%A2-B0072%20anti-human%20CD4%20Antibody.pdf&v=20220824063016
https://d1spbj2x7qk4bg.cloudfront.net/en-us/products/totalseq-b0081-anti-human-cd14-antibody-16827?pdf=true&displayInline=true&leftRightMargin=15&topBottomMargin=15&filename=TotalSeq%E2%84%A2-B0081%20anti-human%20CD14%20Antibody.pdf&v=20220817071325
https://www.biolegend.com/en-us/products/totalseq-b0083-anti-human-cd16-antibody-16829
https://d1spbj2x7qk4bg.cloudfront.net/en-us/products/totalseq-b0085-anti-human-cd25-antibody-16836?pdf=true&displayInline=true&leftRightMargin=15&topBottomMargin=15&filename=TotalSeq%E2%84%A2-B0085%20anti-human%20CD25%20Antibody.pdf&v=20220820063106
https://d1spbj2x7qk4bg.cloudfront.net/en-us/products/totalseq-b0154-anti-human-cd27-antibody-16839?pdf=true&displayInline=true&leftRightMargin=15&topBottomMargin=15&filename=TotalSeq%E2%84%A2-B0154%20anti-human%20CD27%20Antibody.pdf&v=20220820063106
https://d1spbj2x7qk4bg.cloudfront.net/en-us/products/totalseq-b0159-anti-human-hla-dr-antibody-16879?pdf=true&displayInline=true&leftRightMargin=15&topBottomMargin=15&filename=TotalSeq%E2%84%A2-B0159%20anti-human%20HLA-DR%20Antibody.pdf&v=20220830045305
https://d1spbj2x7qk4bg.cloudfront.net/en-us/products/totalseq-b0390-anti-human-cd127-il-7ralpha-antibody-16859?pdf=true&displayInline=true&leftRightMargin=15&topBottomMargin=15&filename=TotalSeq%E2%84%A2-B0390%20anti-human%20CD127%20(IL-7R%CE%B1)%20Antibody.pdf&v=20220820063106
https://d1spbj2x7qk4bg.cloudfront.net/en-us/products/totalseq-b0410-anti-human-cd38-antibody-18086?pdf=true&displayInline=true&leftRightMargin=15&topBottomMargin=15&filename=TotalSeq%E2%84%A2-B0410%20anti-human%20CD38%20Antibody.pdf&v=20220820063106

Simoa:
Validation of the antibodies used for the Simoa assays has been previously described and published in the following articles: Rodero et al., Detection of interferon alpha protein reveals differential levels and cellular sources in disease. J. Exp. Med. 214 (2017); Hadjadj et al., Impaired type I interferon activity and inflammatory responses in severe COVID-19 patients. Science 369 (2020).

# Flow Cytometry

## Plots

Confirm that:

☒ The axis labels state the marker and fluorochrome used (e.g. CD4-FITC).

☒ The axis scales are clearly visible. Include numbers along axes only for bottom left plot of group (a 'group' is an analysis of identical markers).

☒ All plots are contour plots with outliers or pseudocolor plots.

☒ A numerical value for number of cells or percentage (with statistics) is provided.

## Methodology

| | |
|---|---|
| Sample preparation | Frozen PBMCs from three AFB (CMV+) and six EUB (3 CMV+, 3 CMV-) donors were thawed, centrifuged, counted and allowed to rest overnight. For each donor, 1E6 cells were resuspended in PBS supplemented with 2% FBS and incubated with human Fc blocking solution (BD Biosciences) for 10 minutes at 4°C. Cells were then stained with the following antibodies for 30 minutes at 4°C: CD3 VioGreen (Miltenyi Biotec), CD14 V500 (BD Biosciences), CD57 Pacific Blue (Biolegend), NKp46 PE (BD Biosciences), CD16 PerCP-Cy5.5 (BD Biosciences), CD56 APC-Vio770 (Miltenyi Biotec), NKG2A FITC (BioLegend), NKG2C APC (BioLegend). |
| Instrument | Samples were acquired on a MACSQuant 10 cytometer (Miltenyi, S/N 2428). |
| Software | Data were analyzed using FlowJo v10.7.1. |
| Cell population abundance | Between 0.3x1E6 and 0.5x1E6 PBMCs per donor were acquired on the cytometer. The percentage of NKG2C+ and NKG2A+ cells were determined following the gating strategy described below. |
| Gating strategy | Singlets were first selected using FSC-H/FSC-A markers, then with the markers SSC-H/SSC-A. NK cells were then determined as NKp46+/CD3+CD14-Viogreen- cells. From this gate, our subsets of interest were defined as: NKG2C+ cells with NKG2C+/NKG2A- gate; NKG2A+ cells with NKG2A+/NKG2C- gate. Finally, a histogram overlay for CD57 marker was made using the cells from the two previous gates (NKG2C+ and NKG2A+ cells), to confirm the phenotypic characteristic of NK memory cells. |

☒ Tick this box to confirm that a figure exemplifying the gating strategy is provided in the Supplementary Information.

