## [Peer Review File · Nature]

Manuscript Title: Dissecting human population variation in single-cell responses to SARS CoV-2

Reviewer Comments & Author Rebuttals

Reviewer Reports on the Initial Version:

Referees' comments:

Referee #1 (Remarks to the Author):

This study measured single-cell RNA-seq profiles PBMCs collected from multiple continental populations (African, European, Asian) and simulated with SARS-CoV-2 or IAV. The authors found relationships between the immune responses to the two viral stimulations, immune cell type composition, and population differences. They highlight evidence of selection at specific immune eQTLs and COVID-19 loci.

I appreciated that the authors tackled two important questions for the field: how cell state abundance may mediate gene expression and eQTL differences, and how environmental factors may mediate population differences. The authors had sophisticated statistical strategies and stringent quality control, which will make this multi-ancestry dataset very useful for other researchers. However, while the authors presented a few interesting examples, I did not find the study to make substantially novel observations overall, and was not fully convinced by some of the association testing. Particularly in the immune profiling sections, when mapping out complex relationships the authors seemed to omit certain key confounders and analyses, outlined in the specific points below.

Major points:

1. I have some concerns about the cohort design. First, the African and European samples are all of male sex, which is a limitation of the cohort that is not stated in the main text. Given the substantial differences between immune composition and immune response in males and females (e.g., Huang et al. 2021 PNAS, reviewed in Wilkinson et al. 2022 Ann Rev Imm and Jacobsen and Klein 2021 Front Immun), this restricts the overall conclusions regarding population differences in immune composition and viral response. (The Asian samples do include female participants and come from a different recruitment, which the authors account for by excluding it from many analyses.)
2. I could not find a description of the differential abundance (between populations) analysis in the methods, but it's not clear if it corrected for other potential confounders. For example, age is known to be associated with immune cell type proportions, and while the African and European cohorts had similar ranges of ages, this covariate along with others such as cell quality/mortality should be accounted for in differential abundance analysis through methods such as MASC (Fonseka and Rao, et al. 2018, Sci Trans Med) or other regression models. (It would help to have Extended Data or supplementary figures with the age distributions, cell quality/mortality distributions, and any other relevant covariates available for each cohort.)
3. I also have concerns about the CMV analysis. First, given that almost all African participants are CMV+, they may confound an analysis between CMV and cell type abundances (line 192). E.g., as seen in Fig. 2f, if African participants have higher Memory-like NK cell counts than EUR but due to a

different environmental factor other than CMV that wasn't tested, they confound the claim about lymphoid expansion in Africans being because of CMV (line 70-71).

Second, I couldn't find methods describing how the mediation analysis between CMV and cell type proportions was conducted.

Third, the claim in lines 195-198 seems overstated, as Supplementary Note 2 solely points to the Memory-like NK cells as evidence.

Fourth, the discussion presents the CMV analysis as evidence that "the variation of [cellular proportions] is largely due to environmental exposures" and that "population differences in cellular activation states may be driven primarily by lifelong pathogen exposure". While I appreciate the importance of considering environmental factors in immune profiling, testing one exposure is insufficient to make this claim.

4. How was the relationship between genetics and cell state modeled in the mediation analysis (line 1251)? Genetic variants may themselves influence cellular composition (e.g., Orru et al 2013), which would lead to the appearance of a (pseudo)bulk eQTL across the lineage. Also, a p value would help interpret significance of observations.

Minor points:

1. I would have expected to see a differential cell type abundance analysis between stimulation conditions to complement population and expression associations. (Perhaps this analysis was done? I couldn't easily find it.)

2. Did the authors compare the response QTLs to previous viral stimulation studies? (E.g., Randolph et al. 2021 Science)

3. In addition to the missing methods for the CMV mediation analysis and the differential cell type abundance analysis, I couldn't find methods for the variance-explained analyses (e.g., line 126, 145, others).

4. Certain claims seemed overstated (if based on this study) or require a citation (if based on previous studies. E.g., "highlighting the greater inflammatory potential of this virus" (line 137), "[IFN- α 's] determinant role in response to SARS-CoV-2" (line 145). Lines 233-235 do not follow from the preceding analyses of SARS-CoV-2/IAV's reQTL-sharing.

Referee #2 (Remarks to the Author):

The manuscript by Aquino et al. characterizes the transcriptional response of PBMCs exposed to SARS-CoV-2 and Influenza A (IAV) at single-cell resolution. The authors focus on differences in the West European and Central African population samples and identify both population-specific differentially-expressed genes (popDEGs) and differential response genes (popDRGs). They further identified cell proportion differences by ancestry that seem driven by greater latent CMV infection in the Central African samples. They next identified cis-eQTLs and response eQTLs (reQTLs), with a focus on between-lineage and between-virus comparisons. They further show that variation in immune responses between populations is driven largely by cellular heterogeneity, although specific loci are affected by genetic variation. They show that popDEGs present in the European population are associated with signals of rapid adaptation, and further demonstrate that in the European population eQTLs are enriched for introgressed archaic alleles. Finally, the authors demonstrate that (r)eQTLs are enriched for COVID-19 GWAS loci. Ultimately, the authors conclude that cellular

heterogeneity driven by environmental exposures explain differences in European and African population responses to SARS-CoV-2 infection.

This manuscript is exceptionally well-written and thorough in terms of experimentation, explanation, and supporting data. The authors go above-and-beyond in terms of providing data and justification for experimental and analytical choices. The authors do a superb job of incorporating genetic data with scRNA-seq and leveraging the data for multiple insightful analyses. We particularly appreciate the author's discussion of how differential environmental exposures between populations can be conflated with ancestry effects in the Discussion. Lastly, the code and data generated by this project will serve as a key resource for future work on the 'omics of response to SARS-CoV-2 infection. This manuscript is highly promising, and would serve as a useful resource for other investigators. Still some questions remain to bolster the scientific topics and clarify things throughout.

Major comments:

- Why is it that introgressed alleles only seem to have significant effects on gene expression in Europeans? The higher rates of Neanderthal introgression in eastern Asia make it a more powered analysis in that group, and the sample sizes aren't too different. Overall, while the evidence is strongest for archaic admixture in this group the results, or lack thereof, is a bit puzzling and at a minimum deserves more clarity in the discussion.
- Note, as is clear in the Covid-19 HG GWAS manuscript, the results for susceptibility and severity are highly correlated, likely due to who would get tested particularly in the USA.
- Given the richness of the models given here, and the availability of summary statistic data, there is a major opportunity to use metaxcan/fusion or similar methods to actually perform a TWAS given existing data, and compare/contrast those results to those in the Covid-19 HG manuscript. Even if in a constrained set of models (say, genes where archaic introgression is particularly relevant) and not genome-wide could be useful. This still could have potential above and beyond the performed colocalization techniques.

Minor comments:

- The majority of analyses in the manuscript are performed on only male samples. This caveat should be more clearly stated and included as a potential limitation in the Discussion.
- In the text, it would be helpful to make it clear that all samples originate from the same geographic location (Ghent) as this provides important context for understanding potential shared and divergent environmental exposures in the different ancestry populations.
- In the main text, please clarify that all samples were collected prior to the COVID-19 pandemic and thus had no potential to be previously exposed to SARS-CoV-2.
- Are the extremely-high levels of CMV infection in the western African group surprising? It doesn't seem to be a surprise that CMV infection would
- Figure 1E: The "COV" panel is missing a "lollipop" representing the "INFECTED MONO" cells that is present in the "IAV" panel.
- Extended Data Figure 1: please use divergent colors to color the study populations relative to 1000 Genomes populations. Currently, the colors do not make it easy for readers to discern between populations.
- Extended Data Figure 6: Please provide significance bars for part A.
- Extended Data Figure 7: Please provide significance bars for part B.

Referee #3 (Remarks to the Author):

The manuscript by Aquino et al represents a substantial piece of work, sequencing immune cells from over 200 individuals across three ancestries following exposure to SARS-CoV2 and influenza. The manuscript identifies differences in gene expression at baseline and in response to stimulus between populations and cell types, and then attempts to pin down the drivers of that variation, be they genetic or reflective of life long environmental exposures. There is also a substantial human evolutionary component to the work, where the authors attempt to link observations at the experimental level with inferred selective events in the past. The methods are exhaustive and carefully detailed, refreshingly so, and for all that the authors make some curious choices - choosing to use lmer or the inverse normal rank transform instead of established approaches for DE in pseudobulk, I note it only as a remark, definitely not as something to change in revisions - I think that the presented work is very much robust. In addition, the figures are beautiful, consistent across plots and clearly communicate complex analyses.

My main comments are below:

1. The ASH samples seem to come and go, and are absent from the bulk of the paper's analyses, as far as I can tell. For instance, the CMV status of the ASH samples does not appear to have been quantified (no % given in line 191), and they are missing from all of figure 2 on cell type abundances and related analyses, and Asian samples only reappear in time for the population genetics work in the latter part of the manuscript. Why these omissions? Were the analyses performed and not reported, or not performed? Doubtlessly the analyses of the paper would be stronger if they took full advantage of all three population datasets.

2. eQTL results, 1: Recent large scale PBMC sequencing efforts (OneK1K, Yazar et al 2022; but also Perez et al 2022) have also identified eQTLs in single cells and examined replication across individuals and cell types. Yazar et al uses a much more conservative approach to identifying eQTL sharing, and claims most eQTLs are cell type specific; Perez et al compared European and Asian samples with and without lupus. How do these findings compare with the ones in the current study? Please note that I am **not** asking for anything remotely approaching a full comparison (that is far beyond the scope of the work!), but rather for additional contextualisation of the findings of the present study in light of other population scale atlases of gene expression in PBMCs. This can well be in the discussion, but I find it is important.

3. eQTL results, 2: Were any of the identified eQTLs ancestry-specific? From line 1138 I gather all samples were combined to increase power during mapping, and pop was treated as a confounder, but it should still be possible to examine genotype at eQTLs to potentially identify the genetic drivers

of population-specific (r)eQTLs in a systematic way that is orthogonal to the popDRG/DEG approach from lines 236 onwards (eg Gay et al, 2020)? The analyses in that section are very interesting, and so this might not be worthwhile, but at the very least line 250 suggests the existence of some of the ancestry-specific eQTLs, as do the PBS results in the next section. (Again this need not take the form of substantial new analyses!)

4. I also found the selection results in general interesting, and striking, given recent work (Mostazavi et al, bioRxiv, 2022, also an earlier perspective by Umans et al) that argues that eQTLs should be depleted from strong signals of selection, given how they are mapped and identified. Do the authors expect that the strong selective effects of immune pressures explain some of the apparent discordance between their results and the theoretical model proposed?

5. Line 300: Is the lack of a fold enrichment for introgressed SNPs in EAS attributable to the CLUES-inferred coronavirus sweep from the Souilmi et al paper, which likely postdates introgression in the region and consequently blunted any enrichment of introgressed SNPs with immune function, or do the authors simply think the process is different in both places/there's no way of telling with the currently available data?

Minor:

DE testing threshold changed from logFC 0.5 in line 128, for DRGs between cell types, to 0.2 in line 169 for popDEGs and popDRGs. Why?

Line 192: Is it a correlation if one of the variables is simply positive/negative status for CMV infection? Shouldn't it be an association?

Table S5 doesn't report the population pairs driving the observed differences, please add that in? I did assume from the rest of the text that it is all Africa vs Europe.

I couldn't find a summary of the overall composition of the data clusters (how many cells assigned to each category)

Reference to scran (ref 60) should be "Lun ATL, McCarthy DJ, Marioni JC (2016). "A step-by-step workflow for low-level analysis of single-cell RNA-seq data with Bioconductor." *F1000Res.*, 5, 2122" not "Amezquita, R.A. et al. Orchestrating single-cell analysis with Bioconductor. *Nat Methods* 17, 137-145 (2020)."

Line 416: A bit speculative - can this claim be strengthened by looking up evidence of MUC20 expression in the relevant cell types? (Alternatively, should it be tempered?)

Referee #4 (Remarks to the Author):

This manuscript presents findings from single cell RNA sequencing studies of PBMCs from healthy donors of various ancestry background treated with SARS-CoV2 compared to influenza. They show that SARS-CoV2 induces a more heterogeneous ISG activity than influenza, increased pro-inflammatory signature in myeloid cells, and higher rates of CMV infection affecting the lymphoid cells in African descent individuals.

The study provides interesting insights into how ancestral background over time influences current populations' immune response to respiratory viruses. This potentially has implication in better understanding the differential clinical outcomes of individuals from COVID-19 and influenza infection, amongst other infections. More-so, this work highlights the potential differential preferences of specific interferon pathways that have evolved over time through specific ancestral lineages based on prior evolutionary exposure pressures. Additional strengths include the use of SIMOA single molecule array to quantify proteins

Comments:

The data provided is extensive with robust statistical analytics. The work heavily relies on PBMCs from donors of various ancestral background. However, it is unclear how much admixture there is within each of the ancestral cohort being studied.

The results presented is limited based on the specific strains of virus being used. It is unclear why PR8 was used to stimulate human PBMCs since this is often and mostly used as a mouse-adapted strain of influenza. Other more clinically relevant strain could have been used. It would also be helpful if multiple human strains of COV and IAV be used to stimulate the PBMCs to determine common and differential responses.

For Figure 1F. the East Asian group had more ISG activity with IAV than COV- are these mostly outliers?

For the studies presented in Figure 1, why was the focus only on Central African and West Europeans? Why weren't the East Asian group included. Are these convenient samples?

Figure 2F data presented did not include CMV negative controls for the AFB cohort. Again, where were East Asian samples not included in Figure 2.

For studies in presented related to popDRGs, how would one interpret the data and know which cell / gene expression findings ultimately have a dominant effect in their response to the respective viruses?

DCs seem to be the only source of IFN α after stimulation which is shared between the populations. the significance of this finding needs to be better clarified.

Are there other miRNA that are modulates type I interferon other than miR155. These need to be presented and discussed to show how miR155 compares to other know related miRNA such as MIR744, miR211.

In page 11, Figure 6a- why was one sided analyses performed and not two sided analysis.

The authors report there is a strong allele frequency differentiation in specific population involved in interferon immunity. Are these preferentially selected for each ancestry and what is the implication of these findings which needs to be better discussed.

Overall, there is interesting implication on how ancestral genes influence viral immunity. However, the study lacked validations of their findings outside the PBMC cohort. This will enhance the significance of the work.

Author Rebuttals to Initial Comments:

Point-by-point responses to Reviewers' comments:

Referee #1 (immune single-cell genomics):

This study measured single-cell RNA-seq profiles PBMCs collected from multiple continental populations (African, European, Asian) and stimulated with SARS-CoV-2 or IAV. The authors found relationships between the immune responses to the two viral stimulations, immune cell type composition, and population differences. They highlight evidence of selection at specific immune eQTLs and COVID-19 loci.

I appreciated that the authors tackled two important questions for the field: how cell state abundance may mediate gene expression and eQTL differences, and how environmental factors may mediate population differences. The authors had sophisticated statistical strategies and stringent quality control, which will make this multi-ancestry dataset very useful for other researchers. However, while the authors presented a few interesting examples, I did not find the study to make substantially novel observations overall and was not fully convinced by some of the association testing. Particularly in the immune profiling sections, when mapping out complex relationships the authors seemed to omit certain key confounders and analyses, outlined in the specific points below.

RESPONSE: We thank the Reviewer for their encouraging comments, appreciation of the statistical analyses used, and interest of the dataset generated.

With respect to novelty, we would like to highlight several aspects of our study that, we believe, are novel and original. Our study is the first to characterize (i) how populations of different ancestries differ in their immune responses to SARS-CoV-2, (ii) the genetic and environmental drivers of these differences, and (iii) the cellular and molecular consequences of population-specific natural selection and Neanderthal introgression on immune responses to viral exposures. We also report previously unappreciated differences in cell type proportions across populations that we attribute, at least partially, to population differences in the prevalence of CMV infection. Furthermore, we identify novel ancestry-related eQTLs that (i) colocalize with SARS-CoV-2 risk loci and (ii) regulate genes whose expression could mediate the effects of genetics on COVID-19 risk, which we characterize from an evolutionary standpoint. While some of the conclusions of this study are indeed consistent with prior expectations and clinical studies of COVID-19, we believe that this strengthens the validity of our new findings.

The specific responses to the comments related to the association testing and corrections for confounders are presented here below.

Major points:

1. I have some concerns about the cohort design. First, the African and European samples are all of male sex, which is a limitation of the cohort that is not stated in the main text. Given the substantial differences between immune composition and immune response in males and females (e.g., Huang et al. 2021 PNAS, reviewed in Wilkinson et al. 2022 Ann Rev Imm and Jacobsen and Klein 2021 Front Immun), this restricts the overall conclusions regarding population differences in immune composition and viral response. (The Asian samples do include female participants and come from a different recruitment, which the authors account for by excluding it from many analyses.)

RESPONSE:

Regarding male-bias: We entirely agree with the Reviewer that the lack of female individuals in the *EvoImmunoPop* cohort (AFB and EUB donors) somehow limits the generalisation of our findings. The choice to enrol male individuals only was made 10 years ago, at the time of the recruitment, to minimize the impact of non-genetic variation (notably related to sex hormones) and increase power for eQTL mapping. While there are some sex differences in cell composition and cytokine responses, it has been shown that the genetic basis of gene expression variation is largely shared between sexes (Oliva et al. 2020), suggesting good transferability of our main findings. Furthermore, previous studies from our lab have estimated that age and sex have a rather limited effect on immune response variability relative to cellular composition and *cis*-genetic effects (Piasecka et al. 2018), which are the main focus of the present study. While we agree that extending our study to include a balanced representation of males and females would be a great asset, this would require a completely new sampling effort to avoid introducing batch effects related to the recruitment date/sampling protocol. We humbly believe that this is a project and a question in itself, which goes beyond the scope of the present study.

Following the Reviewer's remark, this potential limitation is now explicitly mentioned in the Discussion section (L456-463).

Regarding the use of Asian samples: As noted by the Reviewer, East Asian samples were excluded from several analyses. This choice was motivated by our will to be as conservative as possible in our analyses and conclusions. Indeed, while African and European donors were recruited in the same centre, the same year, and samples were processed jointly (Ghent, Belgium), the Asian donors were recruited 10 years later and their samples were processed independently (Hong Kong). The resulting differences in sampling time and protocols of PBMC purification may introduce a bias when it comes to cross-population comparisons. Thus, to be as cautious as possible, we decided to exclude East Asians when performing population comparisons, as we could not reliably distinguish true population differences (environmental or genetic) from experimental factors. Yet, East Asian samples were included for all analyses related to eQTL mapping (adjusting for the population/recruitment centre) and evolutionary questions (natural selection and Neanderthal introgression), as these analyses are based on intra-population comparisons that are not affected by experimental factors.

We have now explicitly clarified the differences that exist between populations (new Supplementary Table 1) as well as the rationale for excluding the East Asian samples from cross-population comparisons in the main text (L163-168).

2. I could not find a description of the differential abundance (between populations) analysis in the methods, but it's not clear if it corrected for other potential confounders. For example, age is known to be associated with immune cell type proportions, and while the African and European cohorts had similar ranges of ages, this covariate along with others such as cell quality/mortality should be accounted for in differential abundance analysis through methods such as MASC (Fonseka and Rao, et al. 2018, Sci Trans Med) or other regression models. (It would help to have Extended Data or supplementary figures with the age distributions, cell quality/mortality distributions, and any other relevant covariates available for each cohort.)

RESPONSE: We agree with the Reviewer that a detailed description of the analyses of differential abundance was lacking, and we apologize for this. This has now been fixed by adding a detailed description of these analyses in the Methods section (L1181-1199).

Numbers presented in the original manuscript were obtained using a Wilcoxon rank test and were thus unadjusted for age or cell mortality. Following the Reviewer's request, we now report population differences in cellular proportions that are adjusted for age, cell mortality, and the total number of cells per sample (L173 & 177 and updated Supplementary Table 4a). Note that the overall conclusions remain unchanged. We thank the Reviewer for pointing this out.

As requested, we also provide, in the new Supplementary Table 1, detailed information on the demographic characteristics of each population group, including age distribution, sex, cell mortality, CMV status, and mean number of cells per sample (total and breakdown by lineage).

3. I also have concerns about the CMV analysis. First, given that almost all African participants are CMV+, they may confound an analysis between CMV and cell type abundances (line 192). E.g., as seen in Fig. 2f, if African participants have higher Memory-like NK cell counts than EUR but due to a different environmental factor other than CMV that wasn't tested, they confound the claim about lymphoid expansion in Africans being because of CMV (line 70-71). Second, I couldn't find methods describing how the mediation analysis between CMV and cell type proportions was conducted.

Third, the claim in L195-198 seems overstated, as Supplementary Note 2 solely points to the Memory-like NK cells as evidence.

Fourth, the discussion presents the CMV analysis as evidence that "the variation of [cellular proportions] is largely due to environmental exposures" and that "population differences in cellular activation states may be driven primarily by lifelong pathogen exposure". While I appreciate the importance of considering environmental factors in immune profiling, testing one exposure is insufficient to make this claim.

RESPONSE:

Regarding the impact of CMV on cellular proportions and the risk of confounding: The Reviewer is correct in that almost all African donors are CMV+, preventing a direct assessment of the effect of CMV on cell composition in this population. Yet, the statement that "*CMV seropositivity was strongly associated with the proportions of memory-like NK and CD8+ EMRA T cells*" was derived from the observation made in Europeans (Fig. 2f), which we now replicate in East Asians (see below).

The Reviewer is also right in that CMV serostatus alone cannot fully explain the increased lymphoid expansion in Africans (as can be seen from the comparison of CMV+ donors of African and European ancestry in Figure 2f). However, we believe that the initial claim from

the abstract that ‘*higher rates of cytomegalovirus infection [affect] lymphoid cells in African-descent individuals*’ holds true. Indeed, we did not claim that CMV status *alone* explains differences in cell composition in the lymphoid fraction, but that it significantly *contributes* to these differences. The mediation analysis we perform aims at estimating the fraction of population differences that can be attributed to CMV status. To do so, it relies on the implicit assumption that the effect of CMV status on the percentage of NK memory and EMRA CD8⁺ T cells is the same across populations. Specifically, it quantifies the differences in cellular proportions (β) between CMV⁺ and CMV⁻ individuals of European-descent and estimates the difference in cellular proportions that are attributable to CMV as $\beta \cdot \Delta CMV$, where ΔCMV is the difference in prevalence of CMV between European and African-individuals. The percentage of differences mediated then corresponds to the ratio between the effect attributable to CMV status and the total difference in cellular proportions between populations. As such, our analysis would not be confounded by CMV-independent changes in the lymphoid fraction, if the effect of CMV on cell type abundance is comparable between Africans and Europeans. Following the Reviewer’s remark, we now further substantiate this by assessing the effect of CMV status on NK memory cells in East Asians and confirm that the effect is indeed consistent across populations.

To avoid any ambiguity with respect to these analyses and the corresponding conclusion, we have now extended and clarified them in the new Supplementary Note 4 and Supplementary Figure 2. They are also briefly mentioned in the main text (L204-210) and the corresponding sentence in the abstract has been slightly rephrased.

Regarding the description of the mediation analysis: Such a description was indeed missing, and we apologize for this. We have now revised the Methods section to include a detailed description of the mediation analysis between CMV status and cell type proportions (L1227-1252).

Regarding the claim made on line 195-198 suggesting a link between the differences in response to SARS-CoV-2 between CMV⁺ and CMV⁻ donors and the higher severity of COVID-19 among CMV⁺ individuals: we agree that it was an overstatement. We have now removed it.

Regarding the claim made in the discussion that population differences in cellular activation states may be driven primarily by lifelong pathogen exposures: our aim was to emphasize that the observed population differences in cellular proportions are likely attributable to environmental factors (including CMV exposure but not limited to it) rather than to genetic differences. Following the Reviewer’s comment, with which we agree, we have now tempered our statement and present CMV just as an example of the effects of the environment on cell type proportions (L413-418).

4. How was the relationship between genetics and cell state modeled in the mediation analysis (line 1251)? Genetic variants may themselves influence cellular composition (e.g., Orru et al 2013), which would lead to the appearance of a (pseudo)bulk eQTL across the lineage. Also, a p value would help interpret significance of observations.

RESPONSE: We thank the Reviewer for giving us the opportunity to clarify this important point. The mediation analyses do not explicitly model the relationship between genetics and cell composition, because they assume that any single locus (i.e., 100-kb window around any gene of interest) has a negligible contribution to variation in cell composition. While the

Reviewer is right in that genetic variants can influence cellular composition, such genetic determinants tend to be highly polygenic (Vuckovic et al. 2020) and have a moderate effect on cell composition. For example, 20 out of the 23 variants reported in Orru et al. 2013 account for < 10% of the variance in frequency of their associated cell type. Following the Reviewer's remark, and to verify the assumption that *cis*-eQTLs do not significantly alter cellular composition, we have now performed an additional analysis to map the genetic determinants of cellular composition in our setting. Specifically, we have conducted 79 genome-wide association studies of cell proportions: one for each of the 22 cell types and 5 immune lineages in each of the 3 experimental conditions (except for the IAV-infected CD14+ monocytes, which are only present in the IAV condition). By doing so, we identify 18 loci with a genome-wide significant *p*-value at 5×10^{-8} , of which only one remained significant after adjusting for the number of cell types tested (min. adjusted *p*-value = 3.3×10^{-8}). Most importantly, none of these 18 loci overlap with an eQTL, suggesting that the *cis*-genetic variants considered in our mediation analysis do not affect gene expression *through an indirect effect* on cellular composition. Furthermore, we have tested whether the eQTLs identified at lineage level could be driven by an effect of the same index SNP on cellular proportions, and have not found any significant association (even when relaxing the threshold of significance to 10^{-5}). Overall, these analyses lend further support to the conclusions of our mediation analyses.

These new analyses are presented in the new Supplementary Notes 3 and 9, Supplementary Table 4e, and in the Methods section (L1201-1225).

Regarding the significance of observations, note that for each gene, lineage and condition, we report, in Supplementary Table 6, the *p*-values indicating whether genetic and/or cellular composition significantly contribute to population differences in gene expression. We have now clarified this in the legends of Figure 3h and Extended Data Figure 7g.

Minor points:

1. I would have expected to see a differential cell type abundance analysis between stimulation conditions to complement population and expression associations. (Perhaps this analysis was done? I couldn't easily find it.)

RESPONSE: We thank the Reviewer for this suggestion. These analyses were not included in the original manuscript given the nature of our cell type calling strategy: cell types were called separately for each condition, so the differences in cellular abundance detected could reflect either stimulation-induced differentiation/cell death (i.e., biological differences) or preferential assignment to specific cell types upon stimulation (i.e., technical differences). Following the Reviewer's request, we have now conducted this analysis and compared the number of cells assigned to each cell type at basal state and upon stimulation. We show that for most cell types, cellular proportions remain relatively stable upon stimulation by IAV or SARS-CoV-2, with >60% of cell types displaying limited changes in frequency in response to each stimulus (<5 cells or <10% of cells relative to the NS condition). The strongest change concerns the myeloid lineage, where we observe a drastic reduction in the number of CD14⁺ monocytes upon IAV stimulation, which is likely due to increased cell death and their transition to an infected state.

The results of this analysis are now briefly presented in the main text (L128-130) and developed in detail in the new Supplementary Note 2 and Supplementary Table 3e. The corresponding Methods have been updated (L1010-1016).

2. Did the authors compare the response QTLs to previous viral stimulation studies? (E.g., Randolph et al. 2021 Science)

RESPONSE: We thank the Reviewer for this suggestion. We have now added a systematic comparison of the list of eGenes (i.e., genes with an eQTL) detected by our study with other eQTL studies performed on both stimulated and resting PBMCs (Perez et al. 2022; Yazar et al. 2022; Randolph et al. 2021) (Supplementary Table 5e). There are numerous differences across eQTL studies that make this comparison difficult, including differences in experimental protocol, statistical methodology, populations under study, sample size, viral strains and MOI used for stimulation, or even sets of genes considered for analysis. Nevertheless, encouragingly, we find significant overlaps across studies. Specifically, when comparing our dataset with eQTL studies of resting PBMCs (Perez et al. 2022; Yazar et al. 2022), we observe strong enrichments of our basal eQTLs in eGenes from these studies across all lineages ($OR > 4.5$, Fisher's exact p -value $< 1.2 \times 10^{-85}$), with 51-76% of the eGenes we detect at basal state being reported in at least one other study. Note, however, that we identify a total 3,544 genes with previously unreported eQTLs, including 2,658 with high confidence ($FDR < 10^{-4}$). Similarly, when focusing on stimulated PBMCs, we find a >3.2 -fold enrichment of our reQTLs among eGenes specific to the IAV-stimulated conditions in the study of Randolph et al. 2021.

These new analyses are presented in Supplementary Note 8, Supplementary Figure 3 and Supplementary Table 5e, and are briefly mentioned in the main text (L238-241).

3. In addition to the missing methods for the CMV mediation analysis and the differential cell type abundance analysis, I couldn't find methods for the variance-explained analyses (e.g., line 126, 145, others).

RESPONSE: We thank the Reviewer for the careful attention paid to the methods. We have now detailed this analysis in the corresponding Methods section (L1130-1154).

4. Certain claims seemed overstated (if based on this study) or require a citation (if based on previous studies. E.g., “highlighting the greater inflammatory potential of this virus” (line 137), “[IFN- α 's] determinant role in response to SARS-CoV-2” (line 145). L233-235 do not follow from the preceding analyses of SARS-CoV-2/IAV's reQTL-sharing.

RESPONSE:

Regarding the greater inflammatory potential of SARS-CoV-2, we referred to both the enrichment of pro-inflammatory genes among genes up-regulated specifically by SARS-CoV-2, and the confirmation of this trend at the protein level (Supplementary Figure 1g). We realize this could be interpreted as a much more general statement. Thus, we have now rephrased the sentence to focus on the consistency of our findings with previous observations of a SARS-CoV-2-specific inflammatory signature *in vitro* and *in vivo* (Leon et al. 2022; Lee et al. 2020) (L143-144).

Regarding the “determinant role of IFN- α in response to SARS-CoV-2”, our intention was to point to its role as a driver of variation of transcriptional responses in our *in vitro* model (Extended Data Figure 5b, c). We have clarified now that this finding is consistent with the determinant role of IFN- α in the clinical response to SARS-CoV-2 (Bastard et al. 2020) (L153).

Regarding the statement in L233-235: the statement that “the genetic bases of leukocyte responses to SARS-CoV-2 are highly cell type-dependent” referred to the comparison of reQTL effect sizes across cell types (Figure 3c, upper diagonal), while the statement that “myeloid responses are strongly virus-specific” referred to the reQTL-sharing between IAV and SARS-CoV-2. To avoid any ambiguity, we have now rephrased this statement (L253-255).

Referee #2 (population genetics)

The manuscript by Aquino et al. characterizes the transcriptional response of PBMCs exposed to SARS-CoV-2 and Influenza A (IAV) at single-cell resolution. The authors focus on differences in the West European and Central African population samples and identify both population-specific differentially-expressed genes (popDEGs) and differential response genes (popDRGs). They further identified cell proportion differences by ancestry that seem driven by greater latent CMV infection in the Central African samples. They next identified cis-eQTLs and response eQTLs (reQTLs), with a focus on between-lineage and between-virus comparisons. They further show that variation in immune responses between populations is driven largely by cellular heterogeneity, although specific loci are affected by genetic variation. They show that popDEGs present in the European population are associated with signals of rapid adaptation, and further demonstrate that in the European population eQTLs are enriched for introgressed archaic alleles. Finally, the authors demonstrate that (r)eQTLs are enriched for COVID-19 GWAS loci. Ultimately, the authors conclude that cellular heterogeneity driven by environmental exposures explain differences in European and African population responses to SARS-CoV-2 infection.

This manuscript is exceptionally well-written and thorough in terms of experimentation, explanation, and supporting data. The authors go above-and-beyond in terms of providing data and justification for experimental and analytical choices. The authors do a superb job of incorporating genetic data with scRNA-seq and leveraging the data for multiple insightful analyses. We particularly appreciate the author's discussion of how differential environmental exposures between populations can be conflated with ancestry effects in the Discussion. Lastly, the code and data generated by this project will serve as a key resource for future work on the 'omics of response to SARS-CoV-2 infection. This manuscript is highly promising, and would serve as a useful resource for other investigators. Still some questions remain to bolster the scientific topics and clarify things throughout.

RESPONSE: We wish to thank the Reviewer for such positive and encouraging remarks.

Major comments:

- Why is it that introgressed alleles only seem to have significant effects on gene expression in Europeans? The higher rates of Neanderthal introgression in eastern Asia make it a more powered analysis in that group, and the sample sizes aren't too different. Overall, while the evidence is strongest for archaic admixture in this group the results, or lack thereof, is a bit puzzling and at a minimum deserves more clarity in the discussion.

RESPONSE: We thank the Reviewer for this interesting question. One possibility (suggested by Reviewer #3) is that the signals of adaptive introgression in East Asians have been blunted by the selection events that occurred ~25,000 years ago. Another explanation is that the enrichment

detected in Europeans is driven by selection that occurred after their split from East Asians. Such a late-onset European-specific selection could indeed explain the enrichments we observe. To distinguish between these two possibilities, we have compared the frequency of introgressed eQTLs between European and East Asian groups. We find that, among adaptively introgressed variants (top 5% frequency in Eurasians), eQTLs are strongly biased towards higher frequencies in Europeans (Wilcoxon's rank-sum p -value = 3×10^{-4} ; new Supplementary Figure 5). This supports the occurrence of European-specific selection at immune eQTLs after the split between the ancestors of Western and Eastern Eurasians. Furthermore, to formally exclude the hypothesis that the coronavirus-driven sweep identified by Souilmi *et al.* could explain the lack of signal in East Asians, we have repeated our enrichment analysis by excluding all eQTLs selected during the 770-970 generations time-window. By doing so, we find that the enrichments are maintained in Europeans (p -value = 0.001 in COV and p -value = 0.011 in IAV) and remain non-significant in East Asians (p -value > 0.11), further strengthening our conclusions.

We now mention these analyses in the main text (L344-349) and present them in detail in the new Supplementary Note 11 and Supplementary Figure 5. The corresponding Methods have been introduced (L1676-1687).

- Note, as is clear in the Covid-19 HG GWAS manuscript, the results for susceptibility and severity are highly correlated, likely due to who would get tested particularly in the USA.

RESPONSE: We thank the Reviewer for bringing this to our attention. Indeed, we also observe, in our own analyses, a strong overlap between loci associated with susceptibility and severity (OR > 200, Fisher's p -value = 4.2×10^{-40}). Yet, out of 105 eQTLs associated with susceptibility or severity (at nominal p -value < 10^{-4}), 81 are associated specifically with either susceptibility ($n = 19$) or severity ($n = 62$), hence the choice made in the manuscript to analyze them separately. Note that this is also consistent with the results presented in the COVID-19 HGI GWAS manuscript (COVID-19 Host Genetics Initiative 2021), where only 5/13 loci are associated with both susceptibility and severity (13/51 in the preprint for the release 7 that we use in our manuscript, see medRxiv, <https://doi.org/10.1101/2022.12.24.22283874>).

To avoid any ambiguity, we have now clarified our rationale to consider susceptibility and severity separately in the Methods section (L1701-1705).

- Given the richness of the models given here, and the availability of summary statistic data, there is a major opportunity to use metaxcan/fusion or similar methods to actually perform a TWAS given existing data, and compare/contrast those results to those in the Covid-19 HG manuscript. Even if in a constrained set of models (say, genes where archaic introgression is particularly relevant) and not genome-wide could be useful. This still could have potential above and beyond the performed colocalization techniques.

RESPONSE: We thank the Reviewer for highlighting this opportunity to enhance the robustness and relevance of our observations. Using the GWAS data of the COVID-19 Host Genetics Initiative (COVID-19 Host Genetics Initiative 2021), we have now applied the MetaXcan framework (Barbeira et al. 2018) to leverage our genotype-expression dataset and highlight associations between genotypes and COVID-19 traits that could be mediated by the regulation of gene expression (in each of the cell types/lineages detected in our study). Briefly, these new TWAS data support the role of gene expression regulation in mediating the reported COVID-19 genotype-phenotype associations, thus complementing our colocalization analyses. For example, for the *OAS1* variant (rs10774671) that has been associated to COVID-19 susceptibility/severity and to which we attribute an effect on *OAS1* expression across a broad range of stimulated cell types, we find strong evidence of association (p -value $< 2.4 \times 10^{-8}$) between gene expression and COVID-19 severity, with the strongest TWAS signal being observed in stimulated naïve CD8⁺ T cells.

We thank again the Reviewer for this suggestion. These new results are now presented and discussed in the main text (L384-386 and 395-396), Figure 6d and Supplementary Table 9a. The corresponding Methods have been introduced (L1717-1749).

Minor comments:

- The majority of analyses in the manuscript are performed on only male samples. This caveat should be more clearly stated and included as a potential limitation in the Discussion.

RESPONSE: Following the Reviewer's request, we have now added a whole paragraph in the Discussion section to clarify the limitations of our study design and their implications on the generalization of our findings (L456-463).

- In the text, it would be helpful to make it clear that all samples originate from the same geographic location (Ghent) as this provides important context for understanding potential shared and divergent environmental exposures in the different ancestry populations.

RESPONSE: This is indeed a very important point and we thank the Reviewer for pointing it out. Note however that only African and European samples originate from Ghent (Belgium), while East Asian samples were recruited in Hong Kong. Also note that, while recruited in Belgium, most African donors were born in Cameroon or Democratic Republic of Congo. As such, and despite they shared the same environment at the time of the study, they have been exposed to different environments during their first decades of life.

To clarify these important aspects, we now provide extensive detail of the sampling scheme and the demographic characteristics of each population in the new Supplementary Table 1.

- In the main text, please clarify that all samples were collected prior to the COVID-19 pandemic and thus had no potential to be previously exposed to SARS-CoV-2.

RESPONSE: We thank the Reviewer for this suggestion. Here again we wish to clarify that while our samples from the *EvoImmunoPop* study (Ghent) were recruited prior to the COVID-19 pandemic, this was not the case for the East Asian donors who were recruited as part of a study of COVID-19 seroprevalence in Hong Kong. Yet, COVID-19 seropositive individuals were explicitly excluded from our study, such that all East Asian participants were also naïve to SARS-CoV-2.

We have clarified this important point in the main text (L114) and provided details in the new Supplementary Table 1.

- Are the extremely-high levels of CMV infection in the western African group surprising? It doesn't seem to be a surprise that CMV infection would

RESPONSE: We thank the Reviewer for the opportunity to clarify the novelty of this finding. High CMV prevalence in our Central African group is not surprising, as this population originates from countries where CMV is highly prevalent according to epidemiological studies (Zuhair et al. 2019). Furthermore, the effect of CMV infection on cellular differentiation is well established (Bigley et al. 2016; Guma et al. 2004; Patin et al. 2018). Nevertheless, our study is the first to quantify the contribution of latent CMV infection to population differences in the response to RNA viruses and to show that this effect is a major driver of population differences in the transcriptional response of cytotoxic immune cells.

To clarify this, we have now added these references in the main text to better contextualize our findings (L202-203).

- Figure 1E: The “COV” panel is missing a “lollipop” representing the “INFECTED MONO” cells that is present in the “IAV” panel.

RESPONSE: The Reviewer is right that there is no lollipop for the “INFECTED MONO” in the COV panel. This is simply because there are no infected monocytes in the COV-stimulated condition. While IAV can efficiently replicate in monocytes, giving rise to an IAV-specific population of infected monocytes that actively transcribes viral RNAs, we do not see such a phenomenon among SARS-CoV-2-stimulated monocytes.

This has now been clarified in the legend of Figure 1e.

- Extended Data Figure 1: please use divergent colors to color the study populations relative to 1000 Genomes populations. Currently, the colors do not make it easy for readers to discern between populations

RESPONSE: We agree that Extended Data Figure 1 did not allow to clearly discern the study populations from the reference 1000 Genomes Project populations. We have now changed the dot size/shape to clearly distinguish between donors from the study population and reference populations.

- Extended Data Figure 6: Please provide significance bars for part A.

RESPONSE: All pairwise comparisons in this figure are significant at p -value $< 10^{-16}$. To avoid any ambiguity, we have now added significance bars as per the Reviewer’s request (this figure is now Extended Data Figure 5).

- Extended Data Figure 7: Please provide significance bars for part B.

RESPONSE: All pairwise comparisons in this figure are significant at p -value $< 4.7 \times 10^{-3}$. To avoid any ambiguity, we have now added significance bars as per the Reviewer’s request (this figure is now Extended Data Figure 6).

Referee #3 (human evolutionary genetics):

The manuscript by Aquino et al represents a substantial piece of work, sequencing immune cells from over 200 individuals across three ancestries following exposure to SARS-CoV2 and influenza. The manuscript identifies differences in gene expression at baseline and in response to stimulus between populations and cell types, and then attempts to pin down the drivers of that variation, be they genetic or reflective of life long environmental exposures. There is also a substantial human evolutionary component to the work, where the authors attempt to link observations at the experimental level with inferred selective events in the past. The methods are exhaustive and carefully detailed, refreshingly so, and for all that the authors make some curious choices - choosing to use lmer or the inverse normal rank transform instead of established approaches for DE in pseudobulk, I note it only as a remark, definitely not as something to change in revisions - I think that the presented work is very much robust. In addition, the figures are beautiful, consistent across plots and clearly communicate complex analyses.

RESPONSE: We thank the Reviewer for their enthusiastic and supportive comments! While we agree that the statistical methods used for DE in pseudobulk may seem unconventional relative to more established DE software (e.g., DEseq or limma), this choice was made to increase consistency among the different analyses presented (i.e., popDE, eQTL mapping, etc.).

My main comments are below:

1. The ASH samples seem to come and go, and are absent from the bulk of the paper's analyses, as far as I can tell. For instance, the CMV status of the ASH samples does not appear to have been quantified (no % given in line 191), and they are missing from all of figure 2 on cell type abundances and related analyses, and Asian samples only reappear in time for the population genetics work in the latter part of the manuscript. Why these omissions? Were the analyses performed and not reported, or not performed? Doubtlessly the analyses of the paper would be stronger if they took full advantage of all three population datasets.

RESPONSE: We entirely agree that the lack of a systematic comparison between the three populations across the manuscript can seem frustrating. This choice was motivated by our will to be as conservative as possible in our analyses and conclusions. Indeed, while African and European donors were recruited in the same centre, the same year, and samples were processed jointly (Ghent, Belgium), the East Asian donors were recruited 10 years later, and their samples were processed independently (Hong Kong). The resulting differences in sampling time and protocols of PBMC purification may introduce a bias when it comes to cross-population comparisons. Thus, although we did perform all comparisons, to be as cautious as possible, we decided to exclude East Asians when performing population comparisons, as we could not reliably disentangle true population differences (environmental or genetic) from experimental factors. Yet, East Asian samples were included for all analyses related to eQTL mapping (adjusting for the population/recruitment centre) and evolutionary questions (selection and Neanderthal introgression), as these analyses are based on intra-population comparisons that are not affected by experimental differences.

Note that we **did** assess CMV status in East Asian individuals, and these data confirm the link between CMV status and cellular proportions reported for Europeans. Yet, we chose not to include these data in the original manuscript to avoid confusing the reader, as we had excluded East Asian samples from population comparisons.

Following the Reviewer's remarks, we have now clarified the rationale for excluding East Asians from cross-population comparisons in the main text (L163-168) and explicitly presented the differences that exist between populations (new Supplementary Table 1). In addition, the analysis of the CMV status effect on cellular proportions in East Asians is now presented in the new Supplementary Note 4 and Supplementary Figure 2.

2. eQTL results, 1: Recent large scale PBMC sequencing efforts (OneK1K, Yazar et al 2022; but also Perez et al 2022) have also identified eQTLs in single cells and examined replication across individuals and cell types. Yazar et al uses a much more conservative approach to identifying eQTL sharing, and claims most eQTLs are cell type specific; Perez et al compared European and Asian samples with and without lupus. How do these findings compare with the ones in the current study? Please note that I am *not* asking for anything remotely approaching a full comparison (that is far beyond the scope of the work!), but rather for additional contextualisation of the findings of the present study in light of other population scale atlases of gene expression in PBMCs. This can well be in the discussion, but I find it is important.

RESPONSE: We thank the Reviewer for this suggestion. We have now added a systematic comparison of the list of eGenes (i.e., genes with an eQTL) detected by our study with other eQTL studies performed on both stimulated and resting PBMCs (Perez et al. 2022; Yazar et al. 2022; Randolph et al. 2021) (Supplementary Table 5e). There are numerous differences across eQTL studies that make this comparison difficult, including differences in experimental protocol, statistical methodology, populations under study, sample size, viral strains and MOI used for stimulation, or even sets of genes considered for analysis. Nevertheless, encouragingly, we find significant overlaps between our study and those on resting eQTLs (Perez et al. 2022; Yazar et al. 2022). Specifically, we observe strong enrichments of our basal eQTLs in eGenes from these studies across all lineages (OR > 4.5, Fisher's exact p -value < 1.2×10^{-85}), with 51-76% of the eGenes we detect at basal state being previously detected in at least one other study. Note, however, that we identify a total 3,544 genes with previously unreported eQTLs, including 2,658 with high confidence (FDR < 10^{-4}). For stimulated PBMCs, we find a >3.2-fold enrichment of our reQTLs among eGenes specific to the IAV-stimulated condition in the study of Randolph et al. 2021.

These new analyses are presented in the Supplementary Note 8, Supplementary Figure 3 and Supplementary Table 5e, and are briefly mentioned in the main text (L238-241).

3. eQTL results, 2: Were any of the identified eQTLs ancestry-specific? From line 1138 I gather all samples were combined to increase power during mapping, and pop was treated as a confounder, but it should still be possible to examine genotype at eQTLs to potentially identify the genetic drivers of population-specific (r)eQTLs in a systematic way that is orthogonal to the popDRG/DEG approach from L236 onwards (eg Gay et al, 2020)? The analyses in that section are very interesting, and so this might not be worthwhile, but at the very least line 250 suggests the existence of some of the ancestry-specific eQTLs, as do the PBS results in the next section. (Again this need not take the form of substantial new analyses!)

RESPONSE: All samples were indeed combined to increase power during eQTL mapping and population was treated as a confounder. We agree with the Reviewer that the inspection of eQTL frequencies across populations is highly informative as to the genetic contribution to population differences in immune responses, being complementary to the direct comparison of immune responses between populations (popDEG/popDRGs). We wish to emphasize, however, that both the mediation and PBS analyses we performed are tightly linked with the degree of genetic differentiation at eQTLs. First, mediation analysis estimates the percentage of population differences mediated by genetics, which is the ratio between the mediated effect $\beta \cdot \Delta f$ (where β is the eQTL effect size and Δf is the difference in eQTL frequency between populations) and the total difference in expression between populations (measured in our popDEG analysis). Second, PBS increases with the difference in allele frequency between the target population and the mean of frequencies in the two reference populations.

To address the Reviewer's comments, we now provide eQTL frequencies directly in Supplementary Tables 5a-d and compare the number of eQTLs/reQTLs that are specific to each ancestry (i.e., MAF>5% in one population and <1% in other populations), across the various conditions and lineages. Surprisingly, we find that there is little overlap between the ancestry-specific eQTLs identified with this definition and eQTLs identified by the PBS analysis (2-7% of high-PBS eQTLs are ancestry-specific vs. 1-11% among all eQTLs). Indeed, while ancestry-specific eQTLs are predominantly observed at relatively low frequencies (e.g., 98% of ancestry-specific eQTLs have a worldwide MAF<15%), high-PBS eQTLs are enriched in high-frequency variants (94% have worldwide MAF >15%). Yet, strikingly, we find a significant enrichment of East Asian-specific variants among SARS-CoV-2 reQTLs (OR > 4.2, Fisher's exact p -value < 2.3×10^{-6}) and myeloid reQTLs (OR > 6.3, Fisher's exact p -value < 6.2×10^{-7}), relative to other ancestry-specific eQTLs. This observation provides further support to the notion that polygenic adaptation has favoured the differentiation of immune responses to SARS-CoV-2 among individuals of East Asian ancestry. We thank the Reviewer for this suggestion that has extended our original analyses.

These new results are presented in the new Supplementary Note 6 and Supplementary Table 7d, and they are mentioned in the main text (L223-226, L287, L311-312). Furthermore, we report these ancestry-specific eQTLs/reQTLs in Supplementary Tables 5a-d and Supplementary Tables 7a, b for the interested readers.

4. I also found the selection results in general interesting, and striking, given recent work (Mostazavi et al, bioRxiv, 2022, also an earlier perspective by Umans et al) that argues that eQTLs should be depleted from strong signals of selection, given how they are mapped and identified. Do the authors expect that the strong selective effects of immune pressures explain some of the apparent discordance between their results and the theoretical model proposed?

RESPONSE: We thank the Reviewer for this interesting point. Note however that there is no real discordance between our observations and the elegant model proposed by Mostafavi et al. Indeed, both Mostafavi et al. and Umans et al. consider purifying selection at gene level (see Figure 2 from Mostafavi et al.) and report that genes under purifying selection are generally depleted in eQTLs (these eQTLs are negatively selected and maintained at low frequency, preventing them from being detected in typical eQTL studies). The limited overlap between GWAS loci and eQTLs can thus be explained by the fact that disease genes typically evolve under purifying selection and their eQTLs require very large samples sizes to be detected.

In our case, we observe that eQTLs that control immune responses have been privileged targets of positive selection. These eQTLs may have thus provided a selective advantage by increasing resistance to pathogens, favouring their rapid increase in frequency. That these regulatory variants are found at higher frequencies facilitates their detection in eQTL studies. Thus, as hinted by the Reviewer, we believe that the enrichment of immune response eQTLs among positively selected alleles can be explained by the strong selective effects related to immune pressures, and the fluctuant nature of the corresponding pathogen pressure.

5. Line 300: Is the lack of a fold enrichment for introgressed SNPs in EAS attributable to the CLUES-inferred coronavirus sweep from the Souilmi et al paper, which likely postdates introgression in the region and consequently blunted any enrichment of introgressed SNPs with immune function, or do the authors simply think the process is different in both places/there's no way of telling with the currently available data?

RESPONSE: The Reviewer raises a very interesting question. One can indeed imagine that the specific history of East Asians may have obscured a pre-existing enrichment of eQTLs among introgressed SNPs. However, it is also possible that the enrichment detected in Europeans was driven by selection that occurred after their split from East Asians, rather than immediately after introgression. Such a late-onset selection would not have impacted East Asians and could thus explain the European-specific enrichment we observe.

To distinguish between these two possibilities, we have now compared the frequency of introgressed eQTLs between European and East Asian groups. We find that, among adaptively introgressed variants (top 5% frequency in Eurasians), eQTLs are strongly biased towards higher frequencies in Europeans (Wilcoxon Rank Sum p -value = 3×10^{-4} ; new Supplementary Figure 5). This supports the occurrence of European-specific selection at immune eQTLs after the split between the ancestors of Western and Eastern Eurasians. Furthermore, to formally exclude the hypothesis raised by the Reviewer (i.e., the coronavirus-driven sweep identified by Souilmi *et al.* could explain the lack of signal in East Asians), we have repeated our enrichment analysis excluding all eQTLs selected during the 770-970 generations time-window. By doing so, we find that the enrichments are maintained in Europeans (p -value = 0.001 in COV and p -value = 0.011 in IAV) and remain non-significant in East Asians (p -value > 0.11), further strengthening our conclusions.

We have now added these analyses in the new Supplementary Note 11 and Supplementary Figure 5 and mention them in the main text (L344-349). The corresponding Methods have been introduced (L1676-1687).

Minor:

DE testing threshold changed from logFC 0.5 in line 128, for DRGs between cell types, to 0.2 in line 169 for popDEGs and popDRGs. Why?

RESPONSE: We thank the Reviewer for giving us the opportunity to clarify this point. This choice was made because the effect of *population* on gene expression tends to be much weaker than the effect of *stimulation*. Such different thresholds allow to capture, for each analysis, a subset of genes with the strongest effects. Note that this practice is not so uncommon and has been used in other papers from both our lab (Quach et al. 2016; Rotival et al. 2020) and others (Randolph et al. 2021; Nedelec et al. 2016).

Line 192: Is it a correlation if one of the variables is simply positive/negative status for CMV infection? Shouldn't it be an association?

RESPONSE: Although Pearson correlation tests typically assume normality for finite sample sizes, there is no counter-indication to compute a correlation between a binary variable x and a continuous variable y (equivalent to compute the R^2 of the linear model $y \sim a + bx + e$). To avoid confusion, we have now rephrased to “CMV seropositivity was associated with the proportions of memory-like NK and CD8⁺ EMRA T cells in Europeans” (L206-207).

Table S5 doesn't report the population pairs driving the observed differences, please add that in? I did assume from the rest of the text that it is all Africa vs Europe.

RESPONSE: All population comparisons were performed between African and European donors. This has now been added in the Supplementary Table 6 (previously Supplementary Table 5).

I couldn't find a summary of the overall composition of the data clusters (how many cells assigned to each category)

RESPONSE: We thank the Reviewer for suggesting this. We have now added a table (new Supplementary Table 3e) where we provide the number of cells assigned to each cell type/lineage per condition (and per sample), and test for differences in cell abundance between conditions.

Reference to scran (ref 60) should be "Lun ATL, McCarthy DJ, Marioni JC (2016). “A step-by-step workflow for low-level analysis of single-cell RNA-seq data with Bioconductor.” *F1000Res.*, 5, 2122" not "Amezquita, R.A. et al. Orchestrating single-cell analysis with Bioconductor. *Nat Methods* 17, 137-145 (2020)."

RESPONSE: The reference has been modified according to the Reviewer's recommendation.

Line 416: A bit speculative - can this claim be strengthened by looking up evidence of MUC20 expression in the relevant cell types? (Alternatively, should it be tempered?)

RESPONSE: This statement was indeed speculative. According to the Human Protein Atlas, *MUC20* is mostly expressed in the bronchus, the highest expression being observed for respiratory ciliated cells (Figure R1 below). Furthermore, we now replicate the *MUC20* eQTL (rs2177336) in the lung tissue from GTEx (new Supplementary Note 12 and Supplementary Table 9b), providing further support to our findings.

We have accordingly rephrased the discussion as follows: “Given the role of mucins in forming a barrier against infection in the respiratory tract, the high expression of MUC20 in ciliated epithelial cells from the bronchus (Human Protein Atlas) and the detection of the MUC20 eQTL in pulmonary tissue (Supplementary Note 12), we suggest that the greater resistance to viral infections conferred by the Neanderthal haplotype could result from a similar effect on MUC20 expression in the respiratory tract.” (L450-455).

Figure R1: Patterns of *MUC20* RNA levels body-wide, as obtained from Human Protein Atlas.

Referee #4 (lung infections).

This manuscript presents findings from single cell RNA sequencing studies of PBMCs from healthy donors of various ancestry background treated with SARS-CoV2 compared to influenza. They show that SARS-CoV2 induces a more heterogeneous ISG activity than influenza, increased pro-inflammatory signature in myeloid cells, and higher rates of CMV infection affecting the lymphoid cells in African descent individuals.

The study provides interesting insights into how ancestral background over time influences current populations' immune response to respiratory viruses. This potentially has implication in better understanding the differential clinical outcomes of individuals from COVID-19 and influenza infection, amongst other infections. More-so, this work highlights the potential differential preferences of specific interferon pathways that have evolved over time through specific ancestral lineages based on prior evolutionary exposure pressures. Additional strengths include the use of SIMOA single molecule array to quantify proteins

RESPONSE: We thank the Reviewer for their interest in the manuscript and their appreciation of the analyses performed.

Comments:

The data provided is extensive with robust statistical analytics. The work heavily relies on PBMCs from donors of various ancestral background. However, it is unclear how much admixture there is within each of the ancestral cohort being studied.

RESPONSE: We thank the Reviewer for allowing us to clarify this point. There is virtually no admixture in any of the groups studied, except for two African individuals that present 22% of ancestry from the Near-East and 25% from Europe, respectively. Note that such a moderate admixture level in < 1% of individuals, does not significantly impact the results presented in the manuscript.

We have now clarified this in the Methods section (L781-786). Furthermore, we have re-drawn the PCA presented in Extended Data Figure 1a with an updated colour scheme, which provides a clearer picture of the non-admixed nature of our study populations.

The results presented is limited based on the specific strains of virus being used. It is unclear why PR8 was used to stimulated human PBMCs since this is often and mostly used as a mouse-adapted strain of influenza. Other more clinically relevant strain could have been used. It would also been helpful if multiple human strains of COV and IAV be used to stimulate the PBMCs to determine common and differential responses.

RESPONSE: We thank the Reviewer for this remark. We agree that it would be very interesting to explore how PBMC responses vary upon stimulation with different strains of SARS-CoV-2 and IAV, with currently circulating strains in particular. Yet, given the population-level scope of our work (requiring large sample sizes to detect population differences and map their genetic basis), in the trade-off between large sample sizes and large numbers of experimental conditions, we had to prioritize the former. Because we aimed to evaluate the degree and sources of population differences in immune responses to SARS-CoV-2, we chose to use the SARS-CoV-2 strain that circulated at the time of our experiments.

Regarding the IAV strain in this study, our aim was just to compare the SARS-CoV-2 signals with those elicited by another respiratory RNA virus. In this context, PR8 was chosen based on operational considerations, rather than for its clinical relevance. Previous work from the lab and from collaborators (Duffy et al. 2014; Piasecka et al. 2018) has shown that this IAV strain triggers strong IFN responses in healthy donors. Hence, we chose this strain as a proxy for the response to H1N1 influenza, given our experience with the virus and its availability.

We have now clarified the motivations that guided the choice of the viral strains used in the Methods section (L844-855) and discuss the limitations of our study by highlighting the relevance of extending the present study to additional strains in the main text (L463-474).

For Figure 1F, the East Asian group had more ISG activity with IAV than COV- are these mostly outliers?

RESPONSE: The main purpose of Figure 1f was to illustrate that individuals presenting the strongest ISG activity upon IAV stimulation tend to also present the strongest ISG activity upon SARS-CoV-2 stimulation. Yet, we found that IAV tends to trigger a slightly stronger ISG activity than SARS-CoV-2 across all populations/lineages, with the strongest difference observed in myeloid cells (Extended Data Figure 5a). While for most individuals, this difference is rather moderate, it indeed appears much stronger for a few East Asian outliers. This can partly be explained by the lower quality of PBMCs from some East Asian donors, which leads to a reduction in the number of pDCs secreting IFN- α and exacerbates the differences in IFN- α release between SARS-CoV-2 and IAV.

Following the Reviewer's comment, to account for this, we have now resized the dots in Figure 1f, such that the 10% of samples with highest mortality upon thawing appear with smaller dots, effectively reducing the weight of such outliers in the figure.

For the studies presented in Figure 1, why was the focus only on Central African and West Europeans? Why weren't the East Asian group included. Are these convenient samples?

RESPONSE: We thank the Reviewer for raising this important point, as the original manuscript lacked clarity on this point. East Asian samples were excluded from several analyses, given our will to be as conservative as possible in our analyses and conclusions. Indeed, while African and European donors were recruited in the same centre, the same year, and samples were processed jointly (Ghent, Belgium), the East Asian donors were recruited 10 years later and their samples were processed independently (Hong Kong). The resulting differences in sampling time and protocols of PBMCs purification may introduce a bias when it comes to cross-population comparisons. Thus, to be as cautious as possible, we decided to exclude East Asians when performing population comparisons, as we could not reliably disentangle true population differences (environmental or genetic) from experimental factors. Yet, East Asian samples were included in all analyses related to eQTL mapping (adjusting for the population/recruitment centre) and evolutionary questions (natural selection and Neanderthal introgression), as these analyses are based on intra-population comparisons that are not affected by experimental factors.

We have now clarified the differences that exist between populations (new Supplementary Table 1) as well as the rationale for excluding East Asian samples from cross-population comparisons in the main text (L163-168).

Figure 2F data presented did not include CMV negative controls for the AFB cohort. Again, where were East Asian samples not included in Figure 2.

RESPONSE: The Reviewer is correct in stating that almost all African participants are CMV⁺, which prevents from directly assessing the effect of CMV on NK memory cells in this population. This is simply because there are virtually no CMV⁻ donors in our cohort (<1% of African donors), which is consistent with estimates of CMV prevalence in sub-Saharan Africa (Zuhair et al. 2019). Nevertheless, the mediation analysis we conduct circumvents this difficulty by relying on the implicit assumption that the effect of CMV status on the percentage of NK memory and EMRA CD8⁺ T cells is the same across populations. Specifically, it quantifies the differences in cellular proportions (β) between CMV⁺ and CMV⁻ individuals of European-descent and estimates the difference in cellular proportions attributable to CMV status as $\beta \cdot \Delta CMV$, where ΔCMV is the difference in prevalence of CMV between European and African-individuals. The percentage of differences mediated then corresponds to the ratio between the effect attributable to CMV status and the total difference in cellular proportions between the two populations. As such, our analysis would not be confounded by CMV-independent changes in the lymphoid fraction, if the effect of CMV status on cell type abundance is comparable between Africans and Europeans. Following the Reviewer's remark, we now further substantiate this by assessing the effect of CMV status on NK memory cells in East Asians and confirm that the effect is indeed consistent across populations.

To avoid any ambiguity with respect to these analyses and the corresponding conclusion, we have now extended and clarified them in the new Supplementary Note 4 and Supplementary Figure 2. They are also briefly mentioned in the main text (L208-210)

For studies in presented related to popDRGs, how would one interpret the data and know which cell / gene expression findings ultimately have a dominant effect in their response to the respective viruses?

RESPONSE: We thank the Reviewer for this very relevant question. Defining which genes have a dominant effect in the host response to a given virus is a complicated task. One way to approach it is through *in vitro* experiments such as siRNA or CRISPR screens, which typically inform on genes involved in susceptibility to infection at the cellular level (e.g., <https://doi.org/10.1038/s41467-019-13965-x>). Yet, these approaches may lead to conflicting results depending on the experimental conditions and/or the techniques chosen and may be inappropriate in our case: the popDRGs we report are observed in immune cells, and the bulk of viral replication typically occurs in lung epithelial cells.

Another approach, perhaps more adapted to our experimental setting, is to perform transcriptome-wide association studies (TWAS), as suggested by Reviewer #2. This approach focuses on genes whose expression is under genetic control and estimates how their expression in each cell type contributes to disease susceptibility, by aggregating GWAS summary statistics across all genetic variants that increase/decrease gene expression. Following the Reviewer's remark, we have now implemented this approach. Our TWAS data provide support for a role of gene expression in mediating the COVID-19 phenotype-genotype associations reported by the COVID-19 Host Genetics Initiative, allowing us to complement our colocalization analyses. For example, for the *OAS1* variant (rs10774671) that has been associated to COVID-19 susceptibility/severity and to which we attribute an effect on *OAS1* expression across a broad range of stimulated cell types, we find strong evidence of association (p -value $< 2.4 \times 10^{-8}$) between gene expression and COVID-19 severity, with the strongest TWAS signal being observed in stimulated naïve CD8⁺ T cells.

These new results are discussed in the main text (L384-386 and 395-396) and presented in the new Figure 6d and Supplementary Table 9a. The corresponding methods have been introduced (L1717-1749).

DCs seem to be the only source of IFN α after stimulation which is shared between the populations. the significance of this finding needs to be better clarified.

RESPONSE: The Reviewer is right in that, regardless of the population being studied, pDCs are the main source of IFN- α after stimulation by SARS-CoV-2, accounting for 88% of all IFN- α transcripts (genes *IFNA1-IFNA21*). Note, however, that, technically, they are not the only source as other cell types can produce small amounts of IFN- α (e.g., monocytes produce ~6% of all IFN- α transcripts). These results are consistent with the key role attributed to pDCs in type I IFN release (see Asselin-

Paturol and Trinchieri 2005 for a review), and the suspected implication of pDCs in COVID-19 etiology (Laurent et al. 2022).

To avoid any ambiguity, we have now rephrased this sentence in the main text (L153-156).

Are there other miRNA that are modulates type I interferon other than miR155. These need to be presented and discussed to show how miR155 compares to other know related miRNA such as MIR744, miR211.

RESPONSE: We thank the Reviewer for this question. We wish to emphasize that the rationale behind mentioning miR-155 in the original manuscript was based on the detection of a pDC-specific eQTL in its host gene, which we deemed of significance given the strong induction of miR-155 upon immune stimulation, its contribution to IFN regulation, and its association with COVID-19 severity (Giannella et al. 2022). Conversely, we did not detect any eQTLs for *MIR211* nor miR-744's host gene (*VMP1*) in our data and were thus unable to measure their impact on IFN response.

However, as suggested by the Reviewer, the importance of other miRNAs in the immune response should not be neglected. Thus, we have now considered, as a reference base, 50 miRNAs that we previously identified as overexpressed in monocytes following stimulation with the influenza A virus (at $\log_2FC > 0.2$, see Table S2 from Rotival et al. 2020), given that they are arguably relevant for the myeloid response to viral infection. We have detected 17 genetic variants associated to the expression of 8 known host genes of these miRNAs (*CYLD*, *TRIM25*, *TNFAIP6*, *NVL*, *C15orf48*, *HLA-B*, *TRRAP* and *MIR155HG*). To assess the effects of these variants on IFN- α levels and ISG transcriptional activity, at the basal state and following viral stimulation, we have mapped *trans*-(p)QTLs, in each experimental condition, using the ISG scores and SIMOA-based IFN- α measurements from stimulation supernatants. None of these variants were associated to either IFN levels or ISG activity. Note that the absence of effect for the pDC-specific *MIR155HG* eQTL on IFN- α levels in our setting does not exclude an effect at later time points or in response to specific environmental cues.

This analysis is now reported in the new Supplementary Note 7.

In page 11, Figure 6a- why was one sided analyses performed and not two sided analysis.

RESPONSE: We thank the Reviewer for allowing us to clarify this point. The main reason is that we had the strong prior that we should expect an enrichment of GWAS loci among eQTLs and reQTLs, as such enrichments have previously been reported for infectious and auto-immune traits, see for example

Fairfax et al. 2014. Furthermore, this choice was also practical, as the definition of a two-sided test from permutations is not straight-forward when the null distribution is not symmetric around 0. One way to circumvent this is to perform two one-sided tests (for either enrichment or depletion) and correct for multiple testing to obtain a single two-sided p -value. Effectively, this would lead to derive the two tailed p -value directly from the one-tailed p -values by considering $P_{2Tails}=2 \times \min(P_{1Tail}, 1 - P_{1Tail})$.

Note, however that such a change would not alter our conclusions as it would change the enrichment p -values from 10^{-4} and 3×10^{-3} to 2×10^{-4} and 6×10^{-3} , which remain significant at the nominal p -value threshold of 0.01.

The authors report there is a strong allele frequency differentiation in specific population involved in interferon immunity. Are these preferentially selected for each ancestry and what is the implication of these findings which needs to be better discussed.

RESPONSE: The Reviewer is right that we observe marked population-specific differentiation at several genes involved in IFN immunity. However, we do not see any systematic differentiation towards either increased or decreased expression of antiviral effectors. Instead, the various events we observe seem to correspond to distinct events of natural selection, occurring at different periods and exerting antagonistic effects on the expression of various antiviral proteins. For example, the interferon inducible transmembrane protein 2 (*IFITM2*) is controlled by multiple eQTLs, with Europeans harboring higher frequencies of both (i) alleles that increase *IFITM2* expression in B cells and (ii) alleles that decrease *IFITM2* expression in NK and T cells. Broadly, our results support the occurrence of repeated events of selection to maintain a balance between strong antiviral immunity and controlled responses to infection, rather than directional selection toward increased or decreased antiviral immunity in specific populations.

Following the Reviewer's remark, we now provide a new table (Supplementary Table 7c) detailing the effects of selection at eQTLs from 48 well-characterized antiviral effectors and 97 IFN- α responsive genes (25 genes being both antiviral effectors and IFN- α responsive). Furthermore, we discuss the patterns of allelic differentiation and the timing of selection at these loci in the new Supplementary Note 10 and Supplementary Figure 4 and mention these patterns in the main text (L296-298).

Overall, there is interesting implication on how ancestral genes influence viral immunity. However, the study lacked validations of their findings outside the PBMC cohort. This will enhance the significance of the work.

RESPONSE: We agree that our findings are limited to PBMCs, which otherwise provide an accessible model to (i) study the peripheral immune responses to SARS-CoV-2 in three different human populations and (ii) evaluate the links between peripheral immunity and COVID-19 clinical manifestations. While it would be highly relevant to replicate our findings in other tissues or cells (e.g., lung biopsies), obtaining such tissues together with genetic data across many individuals from diverse ancestries is extremely challenging.

However, following the Reviewer's comment, we have now used lung eQTL data (from the GTEx database, GTEx Consortium 2020) to estimate the fraction of our eQTLs that are also detected in RNA-sequencing data from lung tissue. In doing so, we observe that 38% of our eQTLs are also found in the lung, which represents a 4.7-fold enrichment relative to randomly selected SNPs. Furthermore, this percentage increases to 50% among eQTLs that colocalize with COVID-19 hits, and 72% among eQTLs that are shared across all 5 immune lineages considered in our study. We warmly thank the Reviewer for their comment, which has allowed us to extend and complement our findings.

These results are now presented in the new Supplementary Note 12 and Supplementary Table 9b. Furthermore, we have added a whole paragraph in the discussion to clarify the limitations of our study design (i.e., use of PBMCs) and their implications on the generalization of our findings (L456-474).

References cited

- Asselin-Paturel, C., and G. Trinchieri. 2005. 'Production of type I interferons: plasmacytoid dendritic cells and beyond', *J Exp Med*, 202: 461-5.
- Barbeira, A. N., S. P. Dickinson, R. Bonazzola, J. Zheng, H. E. Wheeler, J. M. Torres, E. S. Torstenson, K. P. Shah, T. Garcia, T. L. Edwards, E. A. Stahl, L. M. Huckins, G. TEx Consortium, D. L. Nicolae, N. J. Cox, and H. K. Im. 2018. 'Exploring the phenotypic consequences of tissue specific gene expression variation inferred from GWAS summary statistics', *Nat Commun*, 9: 1825.
- Bastard, P., L. B. Rosen, Q. Zhang, et al. 2020. 'Autoantibodies against type I IFNs in patients with life-threatening COVID-19', *Science*, 370: eabd4585.
- Bigley, A. B., G. Spielmann, N. Agha, D. P. O'Connor, and R. J. Simpson. 2016. 'Dichotomous effects of latent CMV infection on the phenotype and functional properties of CD8+ T-cells and NK-cells', *Cell Immunol*, 300: 26-32.
- GTEx Consortium. 2020. 'The GTEx Consortium atlas of genetic regulatory effects across human tissues', *Science*, 369: 1318-30.
- Duffy, D., V. Rouilly, V. Libri, M. Hasan, B. Beitz, M. David, A. Urrutia, A. Bisiaux, S. T. Labrie, A. Dubois, I. G. Boneca, C. Delval, S. Thomas, L. Rogge, M. Schmolz, L. Quintana-Murci, M. L. Albert, and Consortium Milieu Interieur. 2014. 'Functional analysis via standardized whole-blood stimulation systems defines the boundaries of a healthy immune response to complex stimuli', *Immunity*, 40: 436-50.
- Fairfax, B. P., P. Humburg, S. Makino, V. Naranbhai, D. Wong, E. Lau, L. Jostins, K. Plant, R. Andrews, C. McGee, and J. C. Knight. 2014. 'Innate immune activity conditions the effect of regulatory variants upon monocyte gene expression', *Science*, 343: 1246949.
- Giannella, A., S. Riccetti, A. Sinigaglia, C. Piubelli, E. Razzaboni, P. Di Battista, M. Agostini, E. Dal Molin, R. Manganelli, F. Gobbi, G. Ceolotto, and L. Barzon. 2022. 'Circulating microRNA signatures associated with disease severity and outcome in COVID-19 patients', *Front Immunol*, 13: 968991.

- Guma, M., A. Angulo, C. Vilches, N. Gomez-Lozano, N. Malats, and M. Lopez-Botet. 2004. 'Imprint of human cytomegalovirus infection on the NK cell receptor repertoire', *Blood*, 104: 3664-71.
- COVID-19 Host Genetics Initiative. 2021. 'Mapping the human genetic architecture of COVID-19', *Nature*, 600: 472-77.
- Laurent, P., C. Yang, A. F. Rendeiro, B. E. Nilsson-Payant, L. Carrau, V. Chandar, Y. Bram, B. R. tenOever, O. Elemento, L. B. Ivashkiv, R. E. Schwartz, and F. J. Barrat. 2022. 'Sensing of SARS-CoV-2 by pDCs and their subsequent production of IFN-I contribute to macrophage-induced cytokine storm during COVID-19', *Sci Immunol*, 7: eadd4906.
- Lee, J. S., S. Park, H. W. Jeong, J. Y. Ahn, S. J. Choi, H. Lee, B. Choi, S. K. Nam, M. Sa, J. S. Kwon, S. J. Jeong, H. K. Lee, S. H. Park, S. H. Park, J. Y. Choi, S. H. Kim, I. Jung, and E. C. Shin. 2020. 'Immunophenotyping of COVID-19 and influenza highlights the role of type I interferons in development of severe COVID-19', *Sci Immunol*, 5.
- Leon, J., D. A. Michelson, J. Olejnik, K. Chowdhary, H. S. Oh, A. J. Hume, S. Galvan-Pena, Y. Zhu, F. Chen, B. Vijaykumar, L. Yang, E. Crestani, L. M. Yonker, D. M. Knipe, E. Muhlberger, and C. Benoist. 2022. 'A virus-specific monocyte inflammatory phenotype is induced by SARS-CoV-2 at the immune-epithelial interface', *Proc Natl Acad Sci U S A*, 119.
- Nedelec, Y., J. Sanz, G. Baharian, Z. A. Szpiech, A. Pacis, A. Dumaine, J. C. Grenier, A. Freiman, A. J. Sams, S. Hebert, A. Page Sabourin, F. Luca, R. Blekhman, R. D. Hernandez, R. Pique-Regi, J. Tung, V. Yotova, and L. B. Barreiro. 2016. 'Genetic Ancestry and Natural Selection Drive Population Differences in Immune Responses to Pathogens', *Cell*, 167: 657-69 e21.
- Oliva, M., M. Munoz-Aguirre, S. Kim-Hellmuth, V. Wucher, A. D. H. Gewirtz, D. J. Cotter, P. Parsana, S. Kasela, B. Balliu, A. Vinuela, S. E. Castel, P. Mohammadi, F. Aguet, Y. Zou, E. A. Khramtsova, A. D. Skol, D. Garrido-Martin, F. Reverter, A. Brown, P. Evans, E. R. Gamazon, A. Payne, R. Bonazzola, A. N. Barbeira, A. R. Hamel, A. Martinez-Perez, J. M. Soria, G. TEx Consortium, B. L. Pierce, M. Stephens, E. Eskin, E. T. Dermitzakis, A. V. Segre, H. K. Im, B. E. Engelhardt, K. G. Ardlie, S. B. Montgomery, A. J. Battle, T. Lappalainen, R. Guigo, and B. E. Stranger. 2020. 'The impact of sex on gene expression across human tissues', *Science*, 369.
- Orru, V., M. Steri, G. Sole, C. Sidore, F. Viridis, M. Dei, S. Lai, M. Zoledziewska, F. Busonero, A. Mulas, M. Floris, W. I. Mentzen, S. A. Urru, S. Olla, M. Marongiu, M. G. Piras, M. Lobina, A. Maschio, M. Pitzalis, M. F. Urru, M. Marcelli, R. Cusano, F. Deidda, V. Serra, M. Oppo, R. Pilu, F. Reinier, R. Berutti, L. Pireddu, I. Zara, E. Porcu, A. Kwong, C. Brennan, B. Tarrier, R. Lyons, H. M. Kang, S. Uzzau, R. Atzeni, M. Valentini, D. Firinu, L. Leoni, G. Rotta, S. Naitza, A. Angius, M. Congia, M. B. Whalen, C. M. Jones, D. Schlessinger, G. R. Abecasis, E. Fiorillo, S. Sanna, and F. Cucca. 2013. 'Genetic variants regulating immune cell levels in health and disease', *Cell*, 155: 242-56.
- Patin, E., M. Hasan, J. Bergstedt, V. Rouilly, V. Libri, A. Urrutia, C. Alanio, P. Scepanovic, C. Hammer, F. Jonsson, B. Beitz, H. Quach, Y. W. Lim, J. Hunkapiller, M. Zepeda, C. Green, B. Piasecka, C. Leloup, L. Rogge, F. Huetz, I. Peguillet, O. Lantz, M. Fontes, J. P. Di Santo, S. Thomas, J. Fellay, D. Duffy, L. Quintana-Murci, M. L. Albert, and Consortium Milieu Interieur. 2018. 'Natural variation in the parameters of innate immune cells is preferentially driven by genetic factors', *Nat Immunol*, 19: 302-14.
- Perez, R. K., M. G. Gordon, M. Subramaniam, M. C. Kim, G. C. Hartoularos, S. Targ, Y. Sun, A. Ogorodnikov, R. Bueno, A. Lu, M. Thompson, N. Rappoport, A. Dahl, C. M. Lanata, M. Matloubian, L. Maliskova, S. S. Kwek, T. Li, M. Slyper, J. Waldman, D. Dionne, O. Rozenblatt-Rosen, L. Fong, M. Dall'Era, B. Balliu, A. Regev, J. Yazdany, L. A. Criswell, N. Zaitlen, and C. J. Ye. 2022. 'Single-cell RNA-seq reveals cell type-specific molecular and genetic associations to lupus', *Science*, 376: eabf1970.
- Piasecka, B., D. Duffy, A. Urrutia, H. Quach, E. Patin, C. Posseme, J. Bergstedt, B. Charbit, V. Rouilly, C. R. MacPherson, M. Hasan, B. Albaud, D. Gentien, J. Fellay, M. L. Albert, L. Quintana-Murci, and Consortium Milieu Interieur. 2018. 'Distinctive roles of age, sex, and genetics in shaping transcriptional variation of human immune responses to microbial challenges', *Proc Natl Acad Sci U S A*, 115: E488-E97.
- Quach, H., M. Rotival, J. Pothlichet, Y. E. Loh, M. Dannemann, N. Zidane, G. Laval, E. Patin, C. Harmant, M. Lopez, M. Deschamps, N. Naffakh, D. Duffy, A. Coen, G. Leroux-Roels, F. Clement, A. Boland, J. F. Deleuze, J. Kelso, M. L. Albert, and L. Quintana-Murci. 2016. 'Genetic Adaptation and Neandertal Admixture Shaped the Immune System of Human Populations', *Cell*, 167: 643-56 e17.

- Randolph, H. E., J. K. Fiege, B. K. Thielen, C. K. Mickelson, M. Shiratori, J. Barroso-Batista, R. A. Langlois, and L. B. Barreiro. 2021. 'Genetic ancestry effects on the response to viral infection are pervasive but cell type specific', *Science*, 374: 1127-33.
- Rotival, M., K. J. Siddle, M. Silvert, J. Pothlichet, H. Quach, and L. Quintana-Murci. 2020. 'Population variation in miRNAs and isomiRs and their impact on human immunity to infection', *Genome Biol*, 21: 187.
- Vuckovic, D., E. L. Bao, P. Akbari, et al. 2020. 'The Polygenic and Monogenic Basis of Blood Traits and Diseases', *Cell*, 182: 1214-31 e11.
- Yazar, S., J. Alquicira-Hernandez, K. Wing, A. Senabouth, M. G. Gordon, S. Andersen, Q. Lu, A. Rowson, T. R. P. Taylor, L. Clarke, K. Maccora, C. Chen, A. L. Cook, C. J. Ye, K. A. Fairfax, A. W. Hewitt, and J. E. Powell. 2022. 'Single-cell eQTL mapping identifies cell type-specific genetic control of autoimmune disease', *Science*, 376: eabf3041.
- Zuhair, M., G. S. A. Smit, G. Wallis, F. Jabbar, C. Smith, B. Devleesschauwer, and P. Griffiths. 2019. 'Estimation of the worldwide seroprevalence of cytomegalovirus: A systematic review and meta-analysis', *Rev Med Virol*, 29: e2034.

Reviewer Reports on the First Revision:

Referees' comments:

Referee #1 (Remarks to the Author):

Thank you for your thoughtful responses to the reviews. Many of my comments were satisfactorily addressed by the revision. My remaining comments are listed below.

Major points:

1. While it is encouraging that previous studies have found a relatively smaller effect of sex on eQTLs, it seems that there are still substantial differential expression and differential abundance trends between sexes, which could affect the core claims of this paper. Piasecka, et al. highlights monocytes as one of the cell types that displays sex differences, and myeloid cells appear to be important in this study as well as a mediator of SARS-CoV-2 infection.

The statement of this limitation is a good addition to the discussion, but is not quite sufficient. I recommend two additional changes: (1) making it clear earlier in the paper that the African and European cohorts are all male (at least editing Line 115 to “Central Africa, n = 80 males, West Europe, n = 80 males, East Asia, n = __ males and ___ females” with the corresponding numbers, and maybe even Lines 163-164 to “comparing males of Central African and West European ancestries”. The second statement would also make it clearer that sex isn't biasing your results in this comparison.)

2. In the equation (5) (Lines 1189-1190), population is included as a covariate. But since the goal is to adjust for age and cell mortality (and maybe number of cells, although that is not mentioned in line 1186 or 1192?), those covariates should be the only terms regressed out before the Wilcoxon rank-sum test comparing populations. It's unclear why the population term was included in this model. Even if its effects weren't subtracted (i.e., if the authors just subtracted the other covariates' effects instead of taking the overall residual from this model) it's not necessary and may lead to incorrect estimates the covariates' effects if there are partial correlations between population and other covariates.

3. My queries are largely addressed, but I have two follow-up comments: (1) For the claim in lines 411-412 “the variation of [cellular proportions] is largely due to environmental exposures” — other papers show this may depend on the cell type to some extent, e.g., Aguirre-Gamboa 2016. (2) I still disagree with the framing of this study as a general investigation of environmental drivers of population differences. If CMV is the only environmental factor studied in this paper, that is not enough to make that claim in the title and intro. (Instead, in the discussion, the authors could say that the finding of substantial variation unexplained by cell proportions/genetics in this study motivates future study of socioeconomic factors, rather than framing this study as one that focused on environmental factors too.)

I recommend removing the word “the” from the final sentence of the intro, i.e., changing it to “we delineate environmental and genetic effects” to avoid the implication that you systematically tested

many environmental effects. Also, including “Environmental” in the title makes the paper seem like it did a more comprehensive survey of environmental effects than just CMV, so I recommend rewording that as well, but I will leave the specifics of that to the editor/authors.

Minor points

2. Thank you for the comparison with previously reported eQTLs. Did you also assess the proportion of eQTLs with direction of effect consistent with prior studies?

Referee #2 (Remarks to the Author):

This is a primary revision to a manuscript we previously reviewed. The authors have gone above and beyond in addressing our comments. We appreciate the authors thorough efforts to address not just the comments of our own reviews but those of multiple domain experts.

1 comment to address: re Eurasian eQTLs, the authors claim the elevated introgressed eQTL frequency in Europe is due to selection, however it could easily be subject to ascertainment issues. The authors should justify how this is not the case if they are claiming post-introgression selection.

Referee #3 (Remarks to the Author):

I thank the authors for their response to my comments and the changes to the manuscript, there are some interesting additions worth thinking about in depth.

My only comment at this point is that the x axis in supp figs 5a & b appears mislabelled, and should surely be introgressed alleles, not eQTLs?

Referee #4 (Remarks to the Author):

The authors have been highly responsive to prior critiques and comments. They have provided clarifications in the methods section regarding admixture, use of specific influenza strain, update of figures and tables, additions of new reanalysis and new results, and additions of discussions in the manuscripts. The provided reviewers' comments and how the authors addressed them in the revised manuscript are appreciated.

Author Rebuttals to First Revision:

Point-by-point responses to Reviewers' comments:

Referee #1 (immune single-cell genomics):

Thank you for your thoughtful responses to the reviews. Many of my comments were satisfactorily addressed by the revision. My remaining comments are listed below.

RESPONSE: We thank the Reviewer for their positive comments and are happy to know that our extensively revised version addressed most of the Reviewer's concerns and comments.

Major points:

1. While it is encouraging that previous studies have found a relatively smaller effect of sex on eQTLs, it seems that there are still substantial differential expression and differential abundance trends between sexes, which could affect the core claims of this paper. Piasecka, et al. highlights monocytes as one of the cell types that displays sex differences, and myeloid cells appear to be important in this study as well as a mediator of SARS-CoV-2 infection. The statement of this limitation is a good addition to the discussion, but is not quite sufficient. I recommend two additional changes: (1) making it clear earlier in the paper that the African and European cohorts are all male (at least editing Line 115 to "Central Africa, n = 80 males, West Europe, n = 80 males, East Asia, n = __ males and ___ females" with the corresponding numbers, and maybe even Lines 163-164 to "comparing males of Central African and West European ancestries". The second statement would also make it clearer that sex isn't biasing your results in this comparison.)

RESPONSE: We agree with the Reviewer's suggestions, which make our statements more precise and cautious. We have now edited the manuscript accordingly (see L115-116 and 163).

2. In the equation (5) (Lines 1189-1190), population is included as a covariate. But since the goal is to adjust for age and cell mortality (and maybe number of cells, although that is not mentioned in line 1186 or 1192?), those covariates should be the only terms regressed out before the Wilcoxon rank-sum test comparing populations. It's unclear why the population term was included in this model. Even if its effects weren't subtracted (i.e., if the authors just subtracted the other covariates' effects instead of taking the overall residual from this model) it's not necessary and may lead to incorrect estimates the covariates' effects if there are partial correlations between population and other covariates.

RESPONSE: We thank the Reviewer for this thoughtful comment. First, we entirely agree with the Reviewer in that our description of the model in equation (5) was unclear about whether the number of cells in the sample was used as a covariate. We have now clarified that the number of cells was indeed regressed out, as well as age and cell mortality (see L1188 and 1194).

mortality on the frequency of each cell type, in order to remove their effect before testing for population effects. In this context, adding the population term in equation (5) is required to accurately estimate the effects of these covariates; failing to account for correlations between population and other covariates would lead to true population effects being incorrectly attributed to these covariates, and thus mistakenly removed when subtracting the covariates' effects. As shown in the simulation results below (**Fig. R1**), excluding the population term from the equation leads to biased estimates of age effects on cell type frequencies and a tendency to underestimate population effects.

With respect to the use of the population term in the model, as stated by the Reviewer, we used this model to estimate the respective effects of age, total cell counts in the sample, and cell

Figure R1 | Impact of the population term in equation (5) on estimates of (a) the age effect and (b) the population effect on cell proportions, estimated from age-adjusted data. Cell-type frequencies were simulated assuming independent, linear effects of age and population, for different values of α , the difference in age between populations, β , the effect of age on cell proportions, and γ , the effect of population on cell proportions. **a**, For each set of parameters (α , β , γ), the effect of age on cell proportions was estimated with $(\hat{\beta}_{Pop})$ or without $(\hat{\beta}_{Raw})$ adjustment on population. **b**, For each set of parameters (α , β , γ), the effect of population on cell proportions ($\hat{\gamma}$) was estimated after regressing out the effect of age on cell frequencies, with $(\hat{\gamma}_{Pop})$ or without $(\hat{\gamma}_{Raw})$ adjustment on population in equation (5). 1,000 simulations were performed for each set of parameters (α , β , γ).

Furthermore, we have assessed the impact of adjusting on population in our tests for differential cell-type abundance, by comparing FDRs calculated with or without including the population term in equation (5). The two models yield highly correlated FDRs across all cell types and conditions ($R^2 > 97\%$; **Fig. R2**). Adjusting (or not) for population effects has a limited impact on the rejection of the null hypothesis, with only 4 out of 128 associations passing over the

significance threshold ($FDR \geq 0.01$) when excluding population from the model. This is in line with the simulation results and suggests that including population as a covariate yields a more powerful model to detect differences in cell-type abundance between populations.

Figure R2 | Comparison of FDRs in tests for population differences in cell-type abundance, when including population as a covariate in equation (5) (x -axis) or not (y -axis). Each dot represents a cell type. On the top panels, cell-type abundance is defined based on absolute number of cells per sample (i.e., donor and condition). On the bottom panels, cell-type abundance is defined in terms of the proportion of cells in the sample. The vertical and horizontal lines indicate the threshold used ($FDR = 0.01$) to control type I error rates.

3. My queries are largely addressed, but I have two follow-up comments: (1) For the claim in lines 411-412 “the variation of [cellular proportions] is largely due to environmental exposures” — other papers show this may depend on the cell type to some extent, e.g., Aguirre-Gamboa 2016. (2) I still disagree with the framing of this study as a general investigation of environmental drivers of population differences. If CMV is the only environmental factor studied in this paper, that is not enough to make that claim in the title and intro. (Instead, in the discussion, the authors could say that the finding of substantial variation unexplained by cell proportions/genetics in this study motivates future study of socioeconomic factors, rather than framing this study as one that focused on environmental factors too.). I recommend removing the word “the” from the final sentence of the intro, i.e., changing it to “we delineate environmental and genetic effects” to avoid the implication that you systematically tested many environmental effects. Also, including “Environmental” in the title makes the paper seem like it did a more comprehensive survey of environmental effects than just CMV, so I recommend rewording that as well, but I will leave the specifics of that to the editor/authors.

RESPONSE: We agree with the Reviewer that concluding for a major role of ‘environmental factors’ based on CMV effects only was misleading and thank them for pointing this out. We have changed the title, abstract and introduction (final sentence) according to the Reviewer’s suggestion, and slightly reworded the discussion, to be very clear on this point (L413-416). We have also reworded the sentence stating that “the variation of [cellular proportions] is largely due to environmental exposures” (L407-409).

Minor points. Thank you for the comparison with previously reported eQTLs. Did you also assess the proportion of eQTLs with direction of effect consistent with prior studies?

RESPONSE: When comparing our results to other eQTL studies, we chose to quantify the overlap regardless of eQTL direction. We reasoned that this approach was the most appropriate given the marked differences between studies in experimental design and reported index SNPs. We have now compared eQTL effect directions with the results from Randolph *et al.*, as this study used an experimental setting that is the most similar to ours (i.e., PBMCs from donors of European and African ancestries, resting or stimulated with the influenza A virus). Interestingly, out of the 1,290 eQTLs detected in the same immune lineage and experimental condition by both Randolph *et al.* ($lfsr < 0.1$) and our study ($FDR < 0.01$), $> 98\%$ of effect sizes are estimated to act in the same direction, supporting further the high replicability of our eQTL results (**Fig. R3**). This additional analysis is now mentioned in the main text (L239-240 and L1442-1445) and Supplementary Note 8 (L199-204)

Figure R3 | Comparison of effect sizes of expression quantitative trait loci (eQTLs) between Randolph *et al.* and the present study. Each dot represents an eQTL with a significant effect (Randolph *et al.*: $lfsr < 0.1$; Aquino *et al.*: $FDR < 0.01$) in the same immune lineage and condition (NS: non-stimulated; IAV: influenza A virus-stimulated) in both studies. For each eQTL, we consider the index SNP reported in Randolph *et al.*

Referee #2 (population genetics)

This is a primary revision to a manuscript we previously reviewed. The authors have gone above and beyond in addressing our comments. We appreciate the authors thorough efforts to address not just the comments of our own reviews but those of multiple domain experts.

RESPONSE: We thank the Reviewer for their positive comments and are happy to know that our extensively revised version addressed most of the Reviewer's concerns and comments.

1 comment to address: re Eurasian eQTLs, the authors claim the elevated introgressed eQTL frequency in Europe is due to selection, however it could easily be subject to ascertainment issues. The authors should justify how this is not the case if they are claiming post-introgression selection.

RESPONSE: We thank the Reviewer for raising this important point. We have now repeated our eQTL mapping on a downsampled dataset composed of the same number of individuals ($n = \min_{Pop} n_{Pop}$) from each population, within each lineage/cell-type and condition. In total, we identified 10,276 eQTLs from these downsampled data, of which 87% are shared with our previous eQTL mapping (> 95% at the lineage level). After downsampling, the genome-wide enrichment in signals of archaic introgression targeting eQTLs remained significant, thus strengthening our main conclusions. Yet, we do not see significant differences in the frequency of introgressed-eQTLs between Europeans and East Asians. This suggests, as hinted by the Reviewer, that the increased genetic differentiation observed at introgressed eQTLs, relative to non-eQTLs, could be attributed to an increased power in the original dataset for the detection of eQTLs with a higher frequency in Europeans (**Fig. R4**). Moreover, when evaluating the effects of differences in sample size on the other results relating to archaic introgression, we observed that the difference between Europeans and East Asians are now subtler than in our initial assessment.

Thus, we have reduced emphasis on differences in genome-wide patterns of introgression between European and East Asian donors and focused on the main message: Neanderthal alleles altering immune responses have provided a selective advantage to Eurasians. Accordingly, the main text (see L333-337, L712-714, L1651-1656) and Supplementary Table 8 have been updated, and Supplementary Note 11 and Supplementary Fig. 5 have been removed as no longer relevant.

Figure R4 | Differences in frequency of Neanderthal-introgressed alleles between Europeans (CEU) and East Asians (CHS) as a function of their mean frequency in Eurasians. A single, randomly selected variant is represented for each 100-kb window. eQTLs are shown in red, and non-eQTLs in black. **a**, Scatter plot with overlaid regression lines and 95% confidence intervals for both eQTLs and non-eQTLs. **b**, Boxplots of differences in frequency of Neanderthal-introgressed alleles for various allele frequency bins in Eurasians. Middle line: median; notches: 95% confidence intervals (CI) of median; box limits: upper and lower quartiles; whiskers: 1.5× interquartile range; dots: outliers.

Referee #3 (human evolutionary genetics):

I thank the authors for their response to my comments and the changes to the manuscript, there are some interesting additions worth thinking about in depth.

RESPONSE: We thank the Reviewer for their positive comments and are happy to know that our extensively revised version addressed the Reviewer's concerns and comments.

My only comment at this point is that the x axis in supp figs 5a & b appears mislabelled, and should surely be introgressed alleles, not eQTLs?

RESPONSE: We thank the Reviewer for spotting this mistake, which has now been fixed.

Referee #4 (lung infections).

The authors have been highly responsive to prior critiques and comments. They have provided clarifications in the methods section regarding admixture, use of specific influenza strain, update of figures and tables, additions of new reanalysis and new results, and additions of discussions in the manuscripts. The provided reviewers' comments and how the authors addressed them in the revised manuscript are appreciated.

RESPONSE: We thank the Reviewer for their positive comments and are happy to know that our extensively revised version addressed the Reviewer's concerns and comments.

Reviewer Reports on the Second Revision:

Referees' comments:

Referee #1 (Remarks to the Author):

Thank you for the revision. All my feedback has been adequately addressed. I appreciate the authors' openness to reviewers' comments and the work they have put in to addressing them.

Referee #2 (Remarks to the Author):

All comments have been addressed.